# A numerical model for duricrust formation by water table fluctuations

Caroline Fenske[1,2], Jean Braun[1,2], François Guillocheau[3], and Cécile Robin[3]

[1]German Research Centre for Geosciences GFZ, Helmholtz Centre Potsdam, Telegrafenberg, 14473 Potsdam
[2]Institute of Geosciences, University of Potsdam, 14476 Potsdam, Germany
[3]Univ Rennes, CNRS, Géosciences Rennes – UMR 6118, F-35000 Rennes, France

**Correspondence:** Caroline Fenske (fenske@gfz-potsdam.de), Jean Braun (jbraun@gfz-potsdam.de), François Guillocheau (francois.guillocheau@univ-rennes.fr), Cécile Robin (cecile.robin@univ-rennes.fr)

**Abstract.**

Duricrusts are hard mineral layers forming in climatically contrasted environments. They form in tropical to arid environments, and can be currently observed all around the world, in areas such as Europe, Africa, South America, India, and Australia. In most cases, they cap hills and appear to protect softer layers beneath. Two main hypotheses have been proposed for the formation of duricrusts, i.e., the hydrological or transported model where the enrichment in the hardening element (iron for ferricretes, silica for silcretes or calcium carbonates for calcretes) is the product of leaching and precipitation through fluctuations of the water table during contrasted seasonal cycles, and the laterisation or in-situ model, where the formation of duricrusts is the final compacting stage of laterisation.

In this article, we present the first numerical geomorphological model for the formation of duricrusts based on the hydrological hypothesis. The model is an extension to an existing regolith formation model where the position of the water table is used to predict the formation of a hardened layer at a rate set by a characteristic time scale $\tau$ and over a depth set by the range of fluctuations of the water table, $\lambda$. Hardening causes a decrease in surface erodability, which we introduce in the model as a dimensionless factor $\kappa$ that multiplies the surface transport coefficient of the model.

Using the model we show under which circumstances duricrusts form by introducing two dimensionless numbers that combine the model parameters ($\lambda$ and $\tau$) as well as parameters representing external forcing like precipitation rate and uplift rate. We demonstrate that by using model parameter values obtained by independent constraints from field observations, hydrology and geochronology, the model predictions reproduce the observed conditions for duricrust formation. We also show that there exists a strong feedback from duricrust formation on the shape of the regolith and the position of the water table. Finally, we demonstrate that although duricrusts protect elements of the landscape, their efficiency in doing so is significantly lower than their inherent strength.

## 1  Introduction

Understanding Earth's surface evolution in cratonic areas remains difficult in parts due to its slow and therefore difficult-to-measure rates but also due to the important contribution from chemical weathering and the formation of the regolith. Although

some progress has been made recently in developing quantitative models of regolith formation and evolution on geological time scales (Lebedeva et al., 2010; Braun et al., 2016, 2017), many questions remain open, in part relating to the relative erodability (or resistance to physical erosion) of the weathering process products (Pelletier, 2010; Sacek et al., 2019). In particular, the formation of hard duricrusts is thought to protect the underlying softer weathered rock (Tardy, 1993; Taylor and Eggleton, 2001; Vasconcelos and Carmo, 2018).

The term duricrust encompasses a broad range of hardened layers such as e.g. ferricretes (Fairbridge, 2008; Tardy, 1993), calcretes, silcretes (Nash et al., 1994), dolocretes (Fairbridge, 2008), alcretes (also called, bauxite or bauxitic duricrusts (Taylor and Eggleton, 2001; Horbe and Anand, 2011; dos Santos Albuquerque et al., 2020)), or crusts made of manganese, gypsite (Taylor and Eggleton, 2001). It describes an indurated mineral layer, usually found capping hills or surfaces as seen in Figure 1, that appears to protect them from erosion (Azmon and Kedar, 1985; Twidale and Bourne, 1998; Taylor and Eggleton, 2001). Duricrusts can also be found along valley bottoms in paleodrainage systems (Radtke and Brückner, 1991; Chudasama et al., 2018). When exhumed, the system's channel beds are preserved due to the low erodability of duricrusts, while the neighbouring layers are eroded, which can lead to inverted topographies (Goudie, 1985; Twidale and Bourne, 1998; Taylor and Eggleton, 2001, 2017). Duricrusts are also recognized as essential mineral layers because of their various important functions and properties. They are used as paleoclimatic, tectonic markers (Firman, 1993, e.g.), but also as geochronological markers due to fossil abundance in some duricrusts (Pickford, 2009). Due to their hardness, duricrusts also have an important societal value, for construction and tools (Buchanan, 1807; Brown et al., 2009), and an economical value, as they are rich in important minerals e.g. iron, aluminium, uranium, phosphates (Spier et al., 2006; Bustillo et al., 2013; Chudasama et al., 2019; Hall et al., 2019).

Duricrusts can be found all over the world, under different climatic conditions, ranging from hyper-arid to tropical settings. In most cases, however, an environment with contrasting dry/wet seasons is needed for their formation (Campy and Macaire, 2003; Taylor and Eggleton, 2001; Nash et al., 1994; Tardy, 1981; Taylor and Eggleton, 2017). They form at the surface or subsurface (Stephens, 1970; Firman, 1993; Taylor and Eggleton, 2001; Fujioka et al., 2005). Duricrusts can be found in Africa (Boulangé, 1984; Tardy, 1993; Tardy and Roquin, 1998; Tardy et al., 1991; Chudasama et al., 2018; Pickford, 2009), South America (Girard et al., 2002; Tardy et al., 1991) and North America (Hall et al., 2019), Australia (Taylor and Eggleton, 2001), India (Widdowson, 2009; Ollier and Sheth, 2008), and some paleocrusts can be found in Europe (Ullyott et al., 1998; Král, 1976; Schwarz, 1997; Borger, 2000; Théveniaut et al., 2007; Strasser et al., 2009). Tardy (1993) provides specific conditions needed for the formation of duricrusts and defines a 'typical duricrust profile'. He describes duricrusts as "mostly monogenic, at least millions if not tens of millions of years old.". A climate estimate conducive to the formation of iron duricrusts as described by Tardy (1993) encompasses the following: 1) a mean annual rainfall, $P$, of around 1450 mm/yr, 2) a mean annual temperature, $T$, of ~28°C, 3) a mean relative air humidity of around 70~%, and 4) a long dry period of at least 6 months. For bauxitisation under actual conditions, annual rainfall is estimated to at least 1200 mm (Boulangé, 1984). These values are taken from contemporary regions where ferricrete and alcrete development is still occurring (Monteiro et al., 2014), though those values should not be considered as absolute conditions. Calcrete formation is described under semi-arid to arid climates, with annual precipitation, $P$, around 200 to 600 mm/yr, and mean annual temperatures, $T$, at ~18°C (Eren et al., 2008). Khalaf (2007); Moussavi-Harami et al. (2009) determine that "the suitable climate for calcrete formation include temperatures that

faciliate high evaporation rate". Estimates for environmental conditions, apart from precipitation observations, for other crusts do not exist in such details.

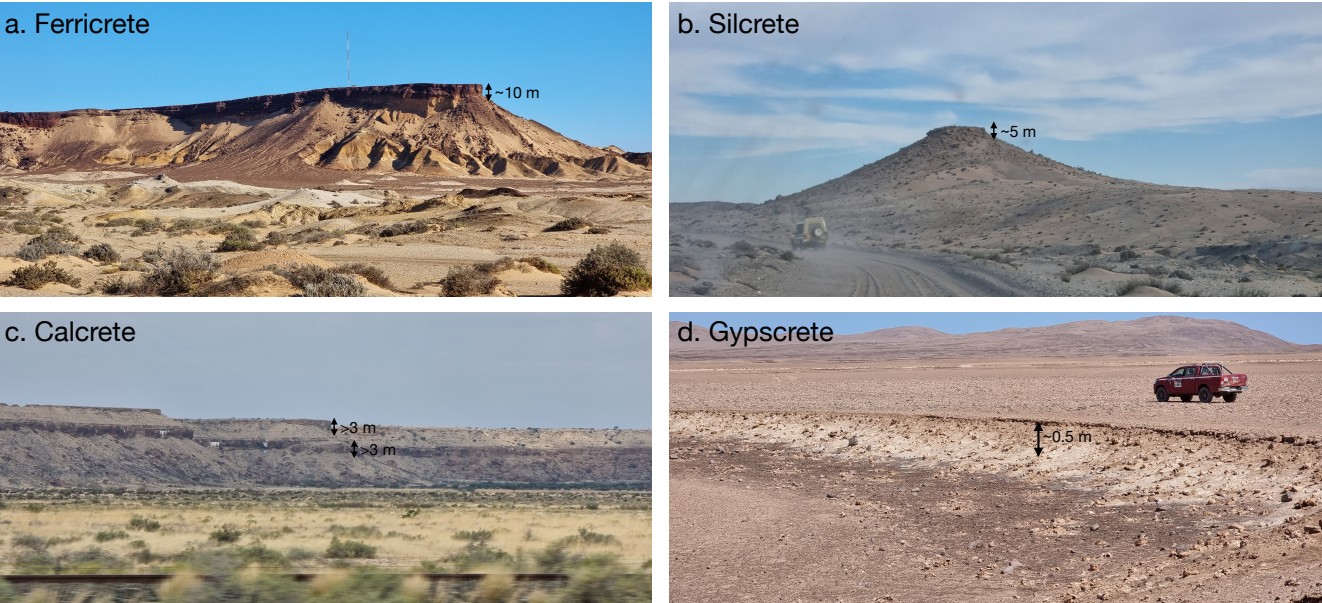

**Figure 1.** Example of duricrusts from Namibia and Chile. Four different duricrusts are illustrated, with thicknesses from multiple metres (a, b, c) to a few decimetres (d). The duricrusts are capping hills and surfaces. a. Ferricrete capping the Kakaoberg, Sperrgebiet, Namibia. b. Silcrete capping a hill, Sperrgebiet, Namibia. c. Calcretes on top of plateaus along the B1 highway, Namibia. d. Gypscrete blanketing surfaces, Atacama desert, Chile. Origin: Caroline Fenske.

Duricrust formation depends on water availability, often linked to climatic conditions, and on the minerals present in the regolith and/or the underlying protolith. The exact conditions under which duricrusts form remain rather unclear, but trends can be defined for various types of duricrust. Gypsite crusts usually form in hyper-arid areas (Watson, 1988; Twidale and Bourne, 1998; Nash, 2011) as an evaporitic blanket over the landscape, as can be seen in Figure 1, through surface/air processes in hyper arid environments. Calcretes often form in arid environments (Nash et al., 1994; Twidale and Bourne, 1998; Nash, 2011), by precipitation of dissolved groundwater calcite under dry conditions. Ferricretes form in areas where more water is available during a certain period of the year (Tardy, 1993) while alcretes appear to form under tropical conditions (Retallack, 2001). Alcretes are mainly made of aluminium oxides and hydroxides, while ferricretes are indurated layers made mostly of iron oxides and oxy-hydroxides (Paton and Williams, 1972; Nahon, 1991; Tardy, 1993; Tardy and Roquin, 1998). Silcretes are hypothesised to form in arid but also humid silica-rich environments (Butt, 1985; Nash et al., 1994; Webb and Golding, 1998; Fairbridge, 2008; Nash, 2011; Rozefelds et al., 2024).

Direct measurements of duricrust formation rates are available in some cases, but more data is clearly needed to determine what controls them. One can estimate that the time needed to create a duricrust is of the order of $10^5$ years or longer (Boulangé,

1984; Goudie, 1985; Tardy, 1993; Taylor and Eggleton, 2001; Alonso-Zarza, 2003; Phillips, 2000; Staunton et al., 2008). It is also clear that duricrusts can form through multiple episodes that last, in some cases, several tens of millions of years (Monteiro et al., 2018). The preservation of duricrusts through time depends on climate too. Depending on the climate the crusts formed under, a climate change may make them unstable (Twidale and Bourne, 1998). In semi-arid to arid areas, duricrusts are preserved and protect the regolith for longer periods of time if formed in a dry climate, while destroyed in subtropical to tropical areas, but the opposite is observed too, as described by Twidale and Bourne (1998); Taylor and Eggleton (2001); Tardy (1993). In the changed environments, geochemical stability decreases (Twidale and Bourne, 1998), leading to the physical breakdown of duricrusts by erosion. Blocks fall on lower parts of the topography and can, if they are not transported further, recompose themselves into new duricrusts or be incorporated into other formations (Twidale and Bourne, 1998; Taylor and Eggleton, 2001). Duricrusts can, however, become very old (Vasconcelos and Carmo, 2018), even after being exposed at the surface, which supports their protective function of underlying softer layers and their capping of present-day topographies (Taylor and Eggleton, 2001). A process which potentially explains duricrust longevity, at least for ferricretes and alcretes, is the rejuvenation through microbial activity inside the duricrusts (Monteiro et al., 2014; Paz et al., 2020, 2021). We will review later in this paper the existing constraints from geochronology, and other indirect estimates of duricrust formation rates and preservation.

There are currently three main hypotheses for the formation of duricrusts: (1) a transport-based process (Goudie, 1985; Wright et al., 1992; Ollier and Galloway, 1990; Taylor and Eggleton, 2001; Achyuthan, 2004; Fairbridge, 2008; Widdowson, 2009; Bonsor et al., 2014; Riffel et al., 2016; Bourman, 1985; Bourman et al., 2020), which could also be referred to as the regional or hydrological model, (2) an in-situ-weathering-based process (Goudie, 1985; Tardy, 1986; Tardy et al., 1988; Nash et al., 1994; Tardy, 1993; Théveniaut and Freyssinet, 1999; Taylor and Eggleton, 2001; Fairbridge, 2008), often referred to as the residual or laterisation model, and (3) recementation of debris coming from preexisting, dismantled duricrusts. In this third model, erosion and gravity enable duricrust blocks to be transported to lower topographies, where recementation in the form of a new duricrust takes place (Twidale and Bourne, 1998; Taylor and Eggleton, 2001; Candy et al., 2003). The first, transport model is most suitable for different types of duricrusts, e.g. calcretes, silcretes but also ferricretes in some cases (Netterberg, 1978). The second, weathering or laterisation model is better adapted to the formation of alcretes and ferricretes mainly. The third is observed with evolving topographies (Goudie, 1973; Twidale and Bourne, 1998; Taylor and Eggleton, 2001; Campy and Macaire, 2003; Taylor and Eggleton, 2017).

In some cases, a mixture of different duricrusts can be observed (Goudie, 1985), where in a same profile, e.g. calcretes and silcretes coexist, or ferricrete calcrete mixtures are found together. Silcrete to calcrete transition is the most common (Nash et al., 2004; Ullyott and Nash, 2016). In some cases, mixed crusts are found to have formed under similar conditions and processes with different sources, while in others, climatic conditions change through time changing the environments composition and behaviour accordingly, and possibly changing the duricrust formation processes, leading to combinations of different duricrusts (Firman, 1993; Nash and Shaw, 1998; Gilkes et al., 2003; Bustillo et al., 2013)).

**Hydrological hypothesis or transported model:**

In this model, duricrusts form at the water table height under a contrasting yearly climate, made of primarily wet and dry periods. Mineral accumulation is considered "absolute" (Goudie, 1985), i.e. with enrichment from external sources to the local regolith. This hypothesis has been used for and adapted to represent the formation of different duricrusts (Taylor and Eggleton, 2001; Twidale and Bourne, 1998; Paquet and Clauer, 1997), e.g. calcretes (Netterberg, 1978; Alonso-Zarza, 2003; Alonso-Zarza and Wright, 2010), silcretes (Webb and Golding, 1998; Taylor and Eggleton, 2001; Ullyott and Nash, 2016; Taylor and Eggleton, 2017) or ferricretes (Goudie, 1985; Ollier and Galloway, 1990; Wright et al., 1992; Temgoua et al., 2005; Widdowson, 2007). During wet periods, the water table height is high and minerals, such as dissolved iron or calcite, are transported from adjacent regions to a topographic low. During dry periods, the water table height drops and the transported minerals precipitate in response to changing redox (e.g. for ferricretes and alcretes), pH (e.g. for calcretes) and environmental conditions such as salinity (e.g. for silcretes), water availability and evaporation processes (e.g. for calcretes and silcretes) (Taylor and Eggleton, 2001, e.g.). Precipitation takes place in undersaturated environments. For ferricretes, it is where redox conditions are optimal, i.e., where reducing conditions become oxidising. In the upper parts of the saturated regolith, i.e., near or just above the water table height, the environment is aerobic, which enables the change in redox conditions and enhances weathering (Taylor and Eggleton, 2001) and precipitation of iron duricrusts. While the upper part of the groundwater is constantly renewed, e.g. by seasonal precipitation, the lower part stays saturated and is possibly stagnant. This can lead to depletion in $0_2$, and the deep part of the saturated zone becomes anaerobic and thus, reducing (Taylor and Eggleton, 2001) and no duricrust forms there. For calcretes, the main drivers are evaporation and evapotranspiration processes linked to water table fluctuations, and $CO_2$ degassing (Alonso-Zarza and Wright, 2010). Such processes only take place at the water table height or in the vadose zone (Moussavi-Harami et al., 2009; Alonso-Zarza and Wright, 2010). Silcrete formation processes remain poorly understood (Thiry and Milnes, 2017; Taylor and Eggleton, 2017). However, evaporation of silica-rich fluids within the regolith is suggested as one of the primary drivers (Taylor and Eggleton, 2001; Thiry and Milnes, 2017). Groundwaters are typically saturated with quartz or amorphous silica (Taylor and Eggleton, 2017).

The seasonal cycle of dissolution and precipitation repeats itself for thousands of years, with the accumulation of minerals leading to the formation of nodules, which, ultimately, cement into a duricrust. As shown by Taylor and Eggleton (2001), these mineralisation patterns can be used to identify the paleo-position of the water-table. In this case, no genetic link between the bedrock and the regolith beneath is needed nor described (Ollier and Galloway, 1990; Taylor and Eggleton, 2001) as most elements are brought from adjacent sources through transport. In this model, duricrusts form at or near the water table, which means that they usually develop several metres below the surface (Stephens, 1970; Firman, 1993), except along valley bottoms where the water table is near or in contact with the surface (Taylor and Eggleton, 2001). It is also generally accepted that, to permit accumulation of materials, the water table position needs to be stable for extended periods of time and the region needs to be tectonically inactive. Subsequent periods of uplift (or base-level drop) are likely to exhume the duricrust to the surface, where it stops evolving but presents more resistance to erosion than the surrounding weathered material (Taylor and Eggleton, 2001; Alonso-Zarza, 2003). This is why duricrusts are often observed capping hilltops (Figure 1) once they are uplifted and

exhumed (Twidale and Bourne, 1998; Taylor and Eggleton, 2001; Widdowson, 2009; Monteiro et al., 2014, 2018; Vasconcelos and Carmo, 2018). The exhumation of paleo channels, that were hardened during duricrust development at the water table height may lead to 'landscape inversion' (Nash et al., 1994; Twidale and Bourne, 1998; Taylor and Eggleton, 2001; Butt and Bristow, 2013), where former channels control the geometry of elongated hill tops.

**Laterisation hypothesis or residue model:**

In this case, duricrusts are considered the ultimate compacting stage of laterisation, bauxitisation and weathering processes leading to what are called pedogenic duricrusts (Grant and Aitchison, 1970; Paquet and Clauer, 1997), e.g. alcretes and ferricretes (Tardy and Roquin, 1992), but also pedogenic calcretes (Alonso-Zarza and Wright, 2010) or silcretes (Taylor and Eggleton, 2017). Mineral accumulation is local and relative to the bedrock, through in-situ weathering (Goudie, 1985).

Laterites are a type of tropical soil, encompassing "residual materials formed directly by in situ rock breakdown", characteristically enriched in iron and aluminium (Widdowson, 2009). Laterites evolve through leaching and vertical transport of material. Easily soluble elements like sulfates, for example, are leached out of the regolith column, whereas insoluble elements or slightly soluble elements like iron, manganese and aluminium remain. Almost all rock types can weather into laterites under the right conditions and right amount of time (Hunt et al., 1977; Widdowson, 2007; Retallack, 2010). Above the bedrock, the depleted saprolite, which can be tens of metres thick, is the thickest part of the profile. Above the saprolite, the mottled zone, is characterized by the accumulation of iron/aluminium nodules and bleached spots, giving it a mottled appearance. With time, the open pores created by leaching close by compaction and cementation of iron/aluminium nodules, leading ultimately to the formation of a ferricrete/alcrete (e.g., Tardy and Roquin (1992); Tardy (1993); Taylor and Eggleton (2001); Tardy and Roquin (1998); Nahon and Bocquier (1983); Nahon (1991)). In this case, a clear genetic link can be observed between the bedrock, the overlying regolith and the duricrust, which is likely to be reflected in their geochemical signature (Tardy, 1993). Also, to have enough material transported vertically, a constant, but slow uplift (or base-level drop) is needed to provide enough material to form a weathering profile and ultimately, a duricrust. A contrasting climate appears also to be an important condition for the formation of duricrusts through laterisation (Tardy, 1993; Taylor and Eggleton, 2001). Lateritic duricrust formation happens near the surface, or even at the surface, contrary to the hydrological formation hypothesis. We will not describe further the complexity involved in duricrust formation through pedogenic processes as it is the main aspect of a complementary research we are in the process of publishing (Fenske et al. *in prep.*).

**Reconstructed duricrusts:**

It is worth noting that duricrusts can also form through the erosion and breaking off of duricrusts that formed at higher elevations (Goudie, 1985; Twidale and Bourne, 1998; Taylor and Eggleton, 2001). The resulting debris accumulate and cement at lower topographies to form "reconstructed" duricrusts. In this case, duricrusts can be genetically linked to previous or multiple cycles of duricrusts. This has been shown through the accurate dating and geochemical analysis of individual nodules (Taylor and Eggleton, 2001).

**Modelling duricrust formation**

Our main objective is to present here a simple, yet predictive numerical model to simulate the geometry and timing of duricrust
formation on geological time scales, in order to predict their effect on surface processes and to compare them to observations.
In other words, we propose here to develop a new parametric representation of the process of duricrust formation based on
a reduced set of generic parameters that can be constrained by comparing the model predictions to observations, rather than
using a representation that would require the calibration of parameters through direct experimentation or measurements.

Apart from the conceptual model developed by Nash et al. (1994) and the highly simplified model developed by Sacek et al.
(2019) to estimate the effects of duricrust formation on erosional patterns at the continental scale, there exists, at this stage, no
numerical model predicting the formation of duricrusts in a dynamically evolving landscape. Several 1D geochemical models
have been proposed for bauxite (i.e. aluminum rich laterite (Campy and Macaire, 2003)) formation (Soler and Lasaga, 1996)
and iron evolution in copper and ferrous crusts (Lichtner and Biino, 1992). They couple a simple solute transport model in a
porous medium with a mineral dissolution and precipitation reaction model where surface erosion is regarded as an imposed
boundary condition.

To the contrary, our model is two-dimensional and fully coupled to a surface processes model and is designed to quantify
the effect that hardening associated with duricrust formation has on the distribution and timing of erosion and the potential
feedback it has on regolith formation and further duricrust formation. We will focus here on developing a model for duricrust
formation based on the hydrological hypothesis (or transported model). We are in the process of developing another model
(Fenske et al. *in prep.*) based on the in-situ hypothesis, which we plan on detailing and comparing to the model presented here
in a future publication.

## 2   Method and Results

### 2.1   Existing regolith formation model (Braun et al., 2016)

Duricrust formation takes place within the regolith, i.e., a layer at the Earth's surface that is formed by the progressive weath-
ering of the underlying basement. In the last decade, several models for regolith formation have been proposed including
Lebedeva et al. (2007), Ferrier and Kirchner (2008), Brantley and White (2009), Maher (2010), Rempe and Dietrich (2014)
Lebedeva et al. (2010), Pelletier (2010), Lebedeva and Brantley (2013), Norton et al. (2014), Pelletier et al. (2016), Braun
et al. (2016), Brantley et al. (2017) or Lebedeva and Brantley (2018). They rely on a variety of approaches combining various
physical, chemical, and hydrological processes.

Here we will use the model for regolith formation developed by Braun et al. (2016) that computes the rate of downward
migration of a weathering front in proportion to the velocity of the water in the overlying permeable regolith. This model is
highly suited for our purpose as it predicts the evolution of the regolith layer and the geometry of the water table over geological
time scales. It needs to be adapted for our purpose as it assumes that the regolith layer has uniform physical properties (hydraulic
conductivity, resistance to erosion, etc.) and cannot predict the seasonal cycles of the water table. The model was designed to

work at the scale of a 'hill' (i.e., from tens of metres to tens of kilometres) connected to an arbitrary base level e.g. a river, a lake or an ocean (see Figure 2).

Braun et al. (2016)'s model is made of three parts: a surface process component, a hydrological component and a weathering component. The surface process component assumes that the evolution of surface topography is controlled by tectonic uplift $U$ and transport of sediment assumed to be proportional to local slope, leading to the following diffusion equation:

$$\frac{\partial z}{\partial t} = U + \frac{\partial}{\partial x} K_D \frac{\partial z}{\partial x} \tag{1}$$

where $U$ is uplift rate (or base-level drop rate), $z$ the topographic height, $K_D$ a surface transport coefficient (or diffusivity), and $x$ and $t$ the spatial and temporal coordinates. Topography is assumed to be fixed at base level on one side of the model ($x = 0$) while the other side (at $x = L$) corresponds to the top of the hill where surface topography gradient is assumed to be nil. The hydrological component is based on the Dupuit-Forchheimer assumptions that flow is dominantly lateral and that discharge is

proportional to the saturated aquifer thickness, leading to the following continuity equation governing the height of the water table, $H$:

$$K(H - z + B)\frac{\partial H}{\partial x} + \int_L^x P \, dx' = 0 \tag{2}$$

where $B$ is the thickness of the regolith layer, $K$ its hydraulic conductivity and $P$ is precipitation rate. The water table is assumed to be fixed with respect to the topography at $x = 0$. Finally, the weathering component assumes that the weathering

front propagates at a velocity that is proportional to the velocity of the fluid, product of the water table gradient by the hydraulic conductivity, according to:

$$\frac{\partial B}{\partial t} = FK\frac{\partial H}{\partial x} - \frac{\partial z}{\partial t} \tag{3}$$

where $F$ is a dimensionless parameter that represents the ratio between the weathering front advance velocity and the fluid velocity and is therefore very small ($\approx 10^{-6} - 10^{-8}$).

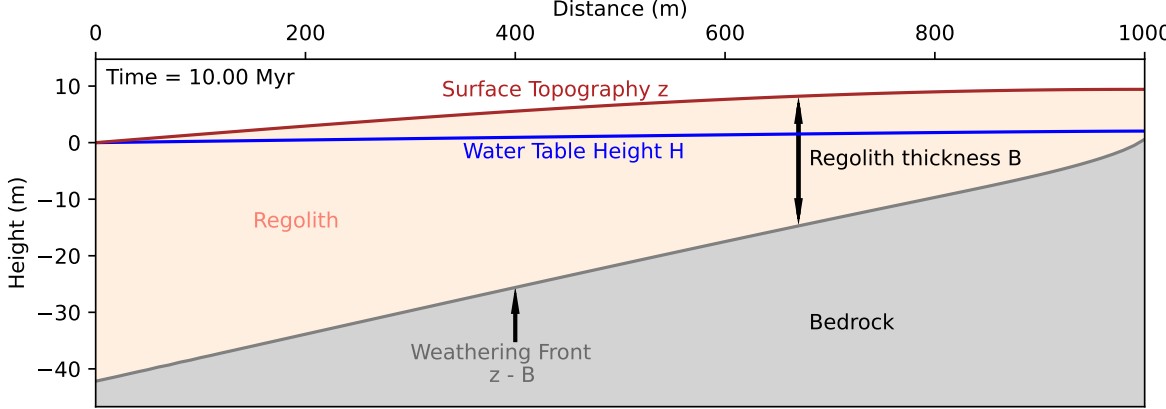

**Figure 2.** Problem geometry with quantities and variables as defined in Braun et al. (2016) and displayed on a steady-state configuration obtained by solving the set of differential equations given in the text for the weathering front velocity (weathering front in dark grey), the geometry of the water table (water table in blue) and the rate of surface erosion (topography is brick red). Result after 10 Myr.

We implemented Braun et al. (2016)'s model into the Xarray-simlab framework (Bovy et al., 2021). The hill topography, water table and regolith thickness shown in Figure 2 are the result of a 'basic' model run on a 1000 m long and initially 20 m high hill, with the water table shown in blue, the surface topography in brown and the weathering front in grey, separating the regolith layer (beige) from the underlying bedrock (dark grey). We see that, for the model parameters used, i.e., $K_D = 1$, $U = 10^{-5}$ m/yr, $K = 10^4$ m/yr, $P = 1$ m/yr and $F = 10^{-6}$, the system reaches a steady-state geometry, with regolith thickness
increasing from the top to the bottom of the hill.

Braun et al. (2016) showed that the predicted steady-state regolith geometry and water table position depends on the value of two dimensionless parameters, $\Omega$ and $\Gamma$, defined as:

$$\Omega = \frac{FKL}{2K_D} = \frac{FK\bar{S}}{U} \text{ and } \Gamma = \frac{K\bar{S}^2}{P} \tag{4}$$

where $\bar{S}$ is the mean surface slope. On the one hand, $\Omega$ controls the thickness of the regolith layer, i.e., $\Omega$ must be greater than
unity for any regolith to develop at the top of the hill and $\Omega$ must be greater than 0.5 for regolith to develop everywhere along the hill. On the other hand, $\Gamma$ controls whether the regolith is thickest at the top of the hill, i.e., when $\Gamma > \frac{\Omega^2}{\Omega-1}$, or thickest at the base of the hill, i.e., when $\Gamma < \frac{\Omega^2}{\Omega-1}$. For the model results shown in Figure 2, the values of $\Omega$ and $\Gamma$ are 5 and 0.25, respectively, which explains why the regolith is thickest at the base of the hill, as $\Gamma < \frac{\Omega^2}{\Omega-1} \approx 6$. The steady-state geometry of the water table depends also on the values of $\Omega$ and $\Gamma$. In steep topographies typical of tectonically active regions ($\Omega \approx 1$ and
$\Gamma > 1$), the water table is close to the bedrock (base of the regolith layer) as observed and assumed in Rempe and Dietrich (2014). In all settings, $\Omega$ is a direct measure of the ratio between the surface slope and the steady state slope of the water table (Braun et al., 2016). In our reference model, the value of $\Omega(\approx 6)$ implies that the water table slope is six times smaller than the surface slope. As explained in details in Braun et al. (2016), a higher water table slope could be obtained by decreasing the value of $\Omega$, by decreasing the value of the hydraulic conductivity, for example.

## 2.2 New duricrust model

To model the formation of duricrusts, we added the dimensionless quantity $\kappa$, or erodability parameter, that represents the relative strength, or more exactly the relative resistance to surface erosion of the material within the regolith layer. The parameter $\kappa$ is allowed to vary between 0 and 1, both horizontally and vertically, i.e., $\kappa = \kappa(x,y)$, where $y$ is a distance measured from the base of the regolith layer. To constrain the time evolution of $\kappa$, we use an additional, fourth equation that represents the hardening process within the range of fluctuations of the water table:

$$\frac{\partial \kappa}{\partial t} = -\frac{\kappa}{\tau}\frac{P}{P_{ref}}e^{-(y-y_w)^2/\lambda^2} - v_W \frac{\partial \kappa}{\partial y} \tag{5}$$

where $y_w$ is the height of the water table depth measured from the base of the regolith, $\lambda$ is the assumed water table fluctuation range (or WTFR) (in m), $\tau$ is the assumed characteristic time for regolith hardening (in years), $P_{ref}$ represents a reference precipitation rate (in m/yr), $v_W$ is the weathering front vertical propagation velocity given by:

$$v_W = FK\frac{\partial H}{\partial x} \tag{6}$$

as described in Braun et al. (2016). Equation 5 contains two parts. The first one represents the self limiting process of hardening that is only taking place in the vicinity of the water table, i.e., within a distance equal to the WTFR, $\lambda$, and is assumed to be proportional to precipitation rate, the prime controlling factor on flow velocity and thus on the transport and precipitation of minerals. The second one represents the advection of the regolith (and thus of the hardening parameter) with respect to the weathering front.

As portrayed in Figure 3, the variable $\kappa$ is also used in an updated version of the erosion equation (1), to account for the increased surface resistance to erosion due to the formation of a duricrust, according to:

$$\frac{\partial z}{\partial t} = U + K_{D,0}\frac{\partial}{\partial x}\kappa(x,y=0)\frac{\partial z}{\partial x} \tag{7}$$

where $K_{D,0}$ is a reference transport coefficient or diffusivity, i.e., corresponding to a regolith that has not been subjected to any hardening. Note also that equation (5) predicts a range of $\kappa$ values starting from unity for a fresh regolith, i.e., that has not been subjected to any hardening, to infinitely small values. In order to define when a duricrust has effectively been formed, we arbitrarily select a threshold value of $\kappa = \kappa_D = 0.2$, which corresponds to the formation of a layer that is five times more resistant to erosion than the surrounding regolith. This arbitrary choice is made in order for the resulting surface topography to present a clear step where the duricrust has formed.

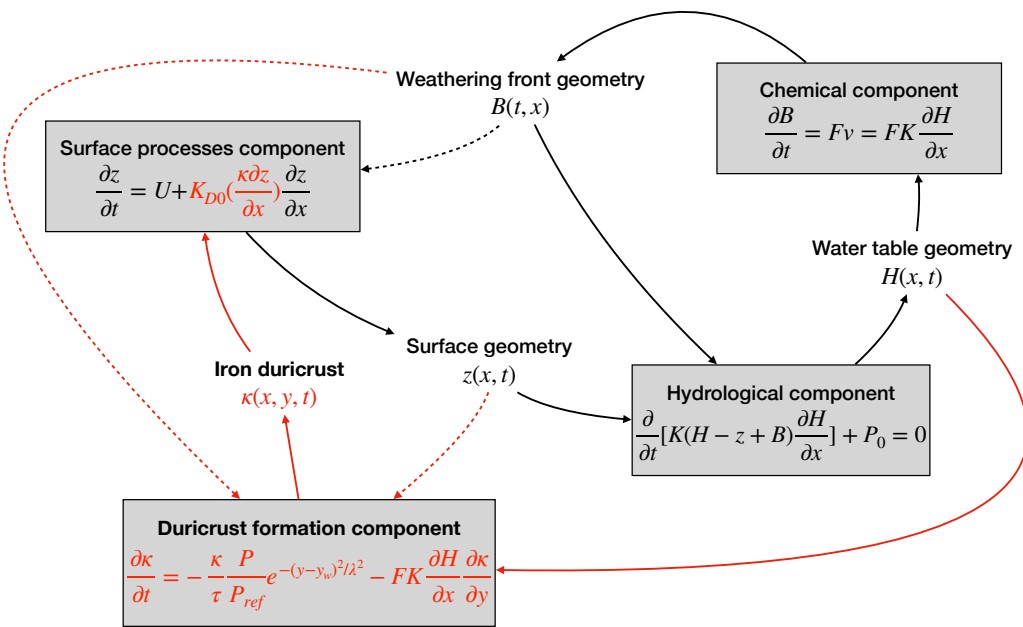

**Figure 3.** Connectivity between the four parameters in the new model, modified from Braun et al. (2016). We see the hydrological component, the surface process component and the chemical component (Braun et al., 2016). The new (highlighted in red) duricrust formation model is based on the hardening coefficient $\kappa$. It is directly connected to the hydrological component by the water table geometry H and the surface processes component by $\kappa$ (concrete arrows), and indirectly influenced by the weathering front geometry B, and surface geometry z (dashed arrows).

270      To ensure accuracy and stability of the numerical solution to this equation, we employed the total variation diminishing method (Leer, 1974) combined with a simple 1D finite volume method along the $y$-direction (Campforts and Govers, 2015). In all numerical experiments shown here, the model resolution was set to 101 points in the horizontal $x$-direction and 501 points in the vertical $y$-direction. Time stepping was controlled by the Courant condition that imposes that material cannot be advected by more than one grid spacing per time step. Note that the model is not truly two-dimensional as the evolution of

275 the surface topography, water table height and weathering front are computed in the $x$-direction only and $\kappa$ is computed in the $y$-direction only; i.e., there are no partial differential equation in the model that depends on both spatial coordinates.

     Finally, it is worth noting that, as seen in Figure 3, the model is made of four different processes, each being represented by its own differential equation and linking four unknowns: $z$, the topographic elevation, $B$, the regolith thickness, $H$, the water table height and $\kappa$, the erodability parameter. As more than one unknown appear in each equation, these equations are coupled.

280 However, in our solution scheme, we solve them sequentially.

     This heuristic approach to the parametrisation of the process of regolith hardening or armouring is based on first-order field evidence. A more mechanistic approach would have required estimates of chemical and physical rate constants that are poorly defined (such as the solubility of different substances in their various valence forms), especially under natural

conditions. Furthermore, as noted by e.g Goudie (1985); Beauvais (2009); Monteiro et al. (2014, 2018), concerning mostly alcretes and ferricretes, the formation of duricrusts is also strongly controlled by biological processes but a proper quantification and parametrisation of this effect is, so far, lacking and cannot be at this stage included in a long-term model for duricrust formation as presented here.

## 2.3 Constraining new model parameters

Our parametrisation of duricrust formation introduces two new model parameters: $\tau$ and $\lambda$. Firstly, the characteristic time for regolith hardening (or duricrust formation), $\tau$, can be constrained by various dating methods.

**The complexity of using various estimates of rates or durations to estimate $\tau$**

Much work has been devoted to dating weathering processes. Caution must be taken before using it to constrain $\tau$. First, one must distinguish between rates of primary weathering that we will define as the transformation of bedrock into regolith by the downward propagation of a weathering front versus rates of secondary weathering resulting in a chemical or physical transformation of the regolith that leads, in some cases, to the formation of a duricrust. The two can take place simultaneously, but it does not have to be so. In most situations, these weathering rates are also different from surface erosion rates. One can think of a simple steady-state situation where a weathering front propagates at a given rate, with the overlying regolith hardening at the same rate and the resulting duricrust being eroded at the same rate too. In this case, all rates are equal, showing that there can be a link between them. But, unless it is clearly documented by assessing independently the three rates, such a situation is most likely to be an idealized representation of the system. Finally, we must also note that in many cases, rates are derived from ages that are interpreted as corresponding to a process, mineral formation, sediment deposition or surface exposure.

In the following section, we have attempted to provide state-of-the-art concerning such rates and dates for systems containing a variety of duricrust types, with the purpose to extract from it estimates of $\tau$ for calcretes, silcretes, ferricretes and alcretes. In doing so, we do not focus on the most likely or proposed formation mechanism, noting, as mentioned above, that we are currently developing a model for duricrust formation by laterisation.

Early attempts to estimate duricrust formation rates have been carried out during the last century by many authors including Cooper (1936); Leneuf (1959); Tardy (1969); Netterberg (1969); Goudie (1973); Netterberg (1978, 1981); Gac (1980); Boulangé (1984); Yijian et al. (1988); Radtke and Brückner (1991); Tardy and Roquin (1992); Boulangé et al. (1997); Paquet and Clauer (1997); Théveniaut and Freyssinet (1999) working on duricrusts in Africa, Asia, Europe, South-America and Australia. Dating calcretes has a long history, although caution has been advised (Radtke et al., 1988; Wright, 1989) to interpret results that used U-Th isochron dating (Kelly et al., 2000) and ESR dating (Radtke et al., 1988; Küçükuysal et al., 2011) methods. Dating ferricretes and bauxitic duricrusts and the associated weathering processes is more recent (Retallack, 2010). Silcretes are one of the most difficult duricrusts to date, as no material datable by isotopic methods is cemented in the crust in high quantities (Taylor and Eggleton, 2001; Král, 1976; Taylor and Eggleton, 2017). Indirect dating of silcretes is possible through stratigraphic analysis and the presence of fossils. In some cases, radiocarbon dating might be used, but limited by the

short [14]C half-life (60 ka). Ferricretes are however, today, one of the best dated duricrust types (e.g. Tardy and Roquin (1992); Tardy (1993); Théveniaut and Freyssinet (1999); Ricordel-Prognon et al. (2010); Tardy and Roquin (1998); Guinoiseau et al. (2021)). Indirect methods were used, such as the latitudinal variation in oxygen isotopes (Chivas and Atlhopheng, 2010) or paleomagnetic dating (Théveniaut and Freyssinet, 1999; Taylor and Eggleton, 2001; Théveniaut et al., 2007). In recent years, iron oxide dating by (U-Th)/He geochronology (Carmo and Vasconcelos, 2006; Allard et al., 2018; Vasconcelos and Carmo, 2018; dos Santos Albuquerque et al., 2020; Heller et al., 2022) has yielded more direct constraints.

**Calcrete formation rates and ages**

Netterberg (1969) described a compilation of ages for calcretes in South Africa. He divided ages into five categories, from pre-Pliocene to recent. The same author (Netterberg, 1978) proposed formation rates ranging from 5 to 50 m/Myr for South African calcretes. Goudie (1985) compiled duricrust formation rates, with estimates fom water balance considerations ranging from 0.5 to 350 mm/ka for pedogenic calcretes (Netterberg, 1981) and 10 to 3000 mm/ka for groundwater calcretes (Netterberg, 1981). Goudie (1973) estimated 1.4 to 2.8 mm/ka for calcretes through calculations of deposition from lime-saturated $H_2O$. Yijian et al. (1988) used a combination of radiocarbon and ESR (Electron Spin Resonance) dating on calcretes from the Amadeus Basin in Australia, to produce formation ages of 20 to 40 ka. They dated multiple horizons of calcrete profiles, linking their formation time to climatic changes from drier to wetter conditions. However, Radtke et al. (1988) and Wright (1989) advised caution on the use of uranium-based dating methods, e.g. U/Th or ESR for dating calcretes, due to post-formation uranium contamination and accumulation. This may explain the wide range of estimates, i.e., from 3 kyr to over 1 Myr, obtained by these methods for the formation of calcrete layers (Wright, 1989).

More recently, Candy et al. (2003) used U-series disequilibrium dating to infer dates and rates of formation of calcrete in the Sorbas Basin in the Betic Cordillera, South-Eastern Spain. Their results suggested that the duricrust formed between 164 ka and 146 ka from which they inferred a formation rate of 6.389 to 14.375 m/Myr. In the same area, Kelly et al. (2000) obtained U/Th isochron ages from nodular and massive calcretes from fluvial terraces of the Rio Aguas drainage system, which yielded minimum ages of formation ranging from 8 ka to more than 350 ka. In the Guadix Basin on the Sierra Nevada of the Betic Cordillera, Alonso-Zarza et al. (2006) used U-Th dating methods on calcrete laminae and obtained an age of approximately 42.6 ka. Pérez-Peña et al. (2009) used the analysis and calcrete description from Alonso-Zarza (2003) to calibrate the age of glacis covered by calcretes, and obtained similar ages, i.e., 55 ka and 68 ka.

In the Negev desert (Israel), Vogel and Geyh (2008) dated colluvium calcretes using radiocarbon and uranium series methods, combined with stratigraphic ages for the region. They estimated that the calcrete formed shortly after 40 ka, under paleoclimatic conditions prone to the formation of calcretes. In the region of Ankara (Turkey), Küçükuysal et al. (2011) used ESR dating on calcretes and obtained ages of approximately 761 ka and 419 ka. Finally, Dhir et al. (2010) used isotopic stability data of calcrete nodules of ~ 2 cm in diameter to infer the age of duricrusts from the Central Thar Desert (Rajastan, India). They compared the resulting data with the regional stratigraphic record and inferred nodule formation duration to be of the order of 10 ka to 20 ka (Dhir et al., 2010). This would translate to a formation rate for the resulting calcrete of 1 to 2 m/Myr.

## Silcrete formation rates and ages

Among the more widely studied duricrust types, silcretes are the most difficult to date. Unlike ferricretes, laterites or alcretes, they do not contain minerals adapted for dating such as goethite or hematite, and they are commonly too old for methods used for dating calcretes such as radiocarbon dating or U-series methods. Until recently, the fossil record and stratigraphic constraints were the only available tools. Král (1976) estimated the age of silica duricrusts in the North-Western part of Bohemia, where silcretes are linked to laterites using stratigraphic arguments. They concluded that laterites and silcretes are linked to Upper Cretaceous and pre-Oligocene planation surfaces. In the Ida basin in New Zealand, Youngson (2005) concluded that silcretes formed episodically between the Late Cretaceous and middle Miocene, or during a single, middle Miocene event, based on stratigraphic arguments. Recently, Taylor and Eggleton (2017) provided a review of silcrete dating practices, which highlighted the difficulty of dating these duricrusts. However, in recent years, Ritter et al. (2023) successfully dated calcretes and silcretes systems in the Namib Desert, Namibia combining U-Pb dating and cosmogenic nuclide methods. The resulting ages they obtained and compared to the paleoclimatic evolution of the region suggest the occurrence of numerous weathering phases and the formation of silcretes and calcretes in stable climatic conditions that ended at the Pliocene-Plesitocene transition ($\sim 3$ Ma) in response to aridification and the incision of the nearby Kuiseb River.

## Ferricrete and alcrete formation rates and ages

Goudie (1985) described ferricrete formation rates estimated by Trendall (1962), who studied water and rock chemistry and estimated 0.4 mm/ka in Uganda, to 17 mm/ka estimated by Mikhaylov (1964) for ferricretes in the Liberian Shield, by estimating silica removal by erosion of plant ash (Mikhaylov, 1964; Goudie, 1985). For alcretes, Cooper (1936) estimated a formation rate of 2.9 mm/ka at the Gold Coast British colony (Ghana) through estimates of rainfall total, spring water chemistry and silica content of bedrock (Cooper, 1936; Goudie, 1985).

Using simple volumetric arguments, Tardy and Roquin (1992) estimated a landscape lowering rate by chemical erosion of 0.1 m/Myr in Chad and 3.3 m/Myr in Malagasy, which provide upper bounds for laterisation rates. Using paleomagnetic methods, Théveniaut and Freyssinet (1999) estimated the duration of laterisation events that led to the formation of a 14 m thick duricrust in French Guinea. From these, one can estimate a duricrust formation rate of around 1.5 m/Myr. In the same study, saprolitisation rates were estimated to be $11.3 \pm 0.5$ m/Myr. In French Guiana, Théveniaut and Freyssinet (2002) provided bauxitic and ferruginous duricrust formation ages spanning three different periods based on paleomagnetic data. They obtained ages of 60, 50 and 40 Ma for the first, highest unit, and Miocene ages for the second and third units. Théveniaut et al. (2007) described a profile called "la borne de fer" in North-Eastern France, where a multiple-metre thick iron duricrust is found just below the surface of a 450 m high hill dated by paleomagnetic means to be of at least Barremian age (Théveniaut et al., 2007). This suggests a minimum formation rate for the duricrust of ~1 m/Myr.

In more recent years, new techniques have been developed to date iron duricrusts by (U-Th)/He thermochronology (Wells et al., 2019). Using those, authors such as dos Santos Albuquerque et al. (2020) estimated formation times ranging from 10 to 15 Myrs for metre-thick iron and bauxitic duricrusts. Heller et al. (2022) have dated iron nodules from bauxitic iron duricrusts

in North-Western Brazil. There, (U-Th)/He ages span from 30 Ma to today. The ages seem to belong to three distinct formation episodes of 1 to 4 Myr duration. Considering a mean duricrust thickness of 5.5 m for the area, these ages yield a formation rate estimate ranging between 1.4 and 5.5 m/Myr. In Brazil, studies around the Quadrilàterro Ferrifero (QF), Vasconcelos and Carmo (2018) report new and previously published $^{40}$Ar/$^{39}$Ar regolith ages of up to 70 Ma old. In this region, thick cangas (i.e. ferricretes and lateritic profiles formed on BIFs) cover the topography with thicknesses of up to 50 m. Taking into account their estimated erosion rates, a minimum formation rate of cangas in the QF would be approximately 0.03 mm/Myr. While ignoring erosion and taking into account climatic changes, a maximum rate can be estimated to be more than 1 m/Myr. Chardon (2023) provides a compilation of duricrust and surface ages from Western Africa including data by e.g. Millot (1970); Vasconcelos et al. (1994); Colin et al. (2005). They show that while weathering must have been active since the early Cenozoic, ferricretes formed between 29 and 24 Ma.

**Bauxitisation rates**

Hénocque et al. (1998) studied the site of Tambao in Burkina Faso, where lateritic ore deposits are found and mined. Rates of cryptomelane precipitation in bauxitic profiles were determined through $^{40}$Ar/$^{39}$Ar dating, ranging from 1 to 5 mm/Myr. Beauvais et al. (2008) dated weathering profiles in western Africa by $^{40}$Ar/$^{39}$Ar dating, and found distinct clusters of ages. In the ferro-manganese crust at the top of the profile, ages range from 59 to 45 Ma while near the weathering front at the base of the profile , ages range from 3.4 to 2.9 Ma. They also evidenced two other weathering episodes in the Miocene, the first between 29 and 24 Ma and the second between 18 and 11.5 Ma. Chardon (2023) presents a summary of duricrust ages and surfaces in Western Africa suing data from Millot (1970); Vasconcelos et al. (1994); Colin et al. (2005). Interpreted bauxitisation periods are defined mostly between 59 and 45 Ma, with a main cluster from 50 to 45 Ma. Weathering took place between 40 and 29 Ma and accordingly, the bauxite weathering period has started during the Early Paleocene. Another phase of bauxitisation is dated to be Late Miocene, around 14 to 6 Ma for the Cameroon Rise. Campos et al. (2023) estimated bauxitisation rates by (U–Th)/He dating on goethite in ferriferous bauxite profiles in the Quadrilàtero Ferrifero, in Brazil. Ages range from 18 and 8 Ma for ochre Al-rich goethites and are younger than 4 Ma for black Al-depleted goethites. Dissolution/precipitation patterns of iron in the profile would be the cause of such young ages (Campos et al., 2023).

Bauxitisation rates can also be derived from theoretical or volumetric calculations. Based on thermodynamic considerations, Fritz and Tardy (1973) inferred bauxitisation rates of 3 m/Myr. Chen et al. (1988) estimated weathering rates of 120 m/Myr in a profile in the Tatun volcanic province in Taiwan using volumetric calculations. Similarly, Boulangé et al. (1997) report landscape lowering rates from bauxite formation on granite in the Ivory Coast of 30 m to form 15 m of bauxite and up to a 70~% volume reduction to form the pisolitic bauxite overlying the main bauxite. They report this taking from 3 to 5 Myrs. Gac (1980) calculated a bauxitisation rate of 1.35 m/Myr in the Chad Chari basin (Chad). Boulangé et al. (1997) described a 90 m lowering of Cretaceous sandstones and mudrocks in Brazil to form 10 m of sandy residuum over bauxite; this would have taken, according to their calculations, between 30 and 100 Myrs to achieve. In northwestern Brazilian Amazonia, Horbe and Anand (2011) proposed that bauxites took between 15 and 20 Myrs to form. Momo et al. (2020) described bauxitisation of volcanic rocks in Western Cameroon, and combined it with detailed field measurements. From volumetric calculations they

inferred bauxitisation rates that vary with the parent rock. On basalts, they obtain a weathering rate value of ∼0.8 m/Myr and of ∼1.6 m/Myr on trachytes..

**Kaolinisation rates**

Vasconcelos and Conroy (2003) performed [40]Ar/[39]Ar laser incremental heating analyses on samples from weathering profiles from 11 sites in the Dugald River area (Queensland, Australia), across 3 distinct elevations. They observed that the higher the sample is collected from, the older the age is, ranging from 16 to 12 Ma at high elevation, 6 to 4 Ma at mid-elevation and 2.2 to 0.8 Ma near the base of the section. With these ages, they estimated a regional weathering rate in the regolith of 3.8 m/Myr over the last 15 Ma. Taylor and Eggleton (2001) compiled multiple rates of formation of different weathering processes, but also
described laterisation and bauxitisation rate acquisition through paleomagnetic dating methods. They describe that laterites would take from 0.6 to 6 mm for 1000 years to develop.

Leneuf (1959) estimated a rate of ~5 m/Myr for kaolinite formation at Lakota and Boulangé (1984) observes 1.4 m/Myr on Mount Tato (Ivory Coast), where Paquet and Clauer (1997) described bauxitisation rates of 3 m/Myr. Tardy (1969) estimated through geochemical calculations the time necessary to transform 1 m of pure granite into kaolinite in temperate climates at
approximately 100 000 yrs. According to Tardy and Roquin (1992), "1000 mm of water percolating each year through a profile allows the formation of 1 m of kaolinite lithomarge and consequently allows a lowering of the weathering fronts of 10 m per Myr or 1000 m per 100 Myr". Benedetti et al. (1992) described basalts in Brazil "continuously evolving" since 700 000 yrs ago. These ages were computed by volumetric calculations using geochemical properties of percolating waters in basaltic rocks and their associated weathering products (Benedetti et al., 1992).

**Weathering and saprolitisation rate**

Carmo and Vasconcelos (2006) derived saprolitisation rates and weathering front lowering from [40]Ar/[39]Ar dating methods on cryptomelane from grains collected on a manganocrete profile at the Cachoeira Mine in the QF in Brazil. As described by others, e.g. Spier et al. (2006), the oldest ages are registered at the top of the weathering profile (~ 12.7 Ma), while younger ages are registered at the bottom of the profile (~ 5.2 Ma). Saprolitisation rate is inferred to be ~ 24.9 t/km$^2$/yr and weathering
front propagation ~ 8.9 m/Myr (Carmo and Vasconcelos, 2006).

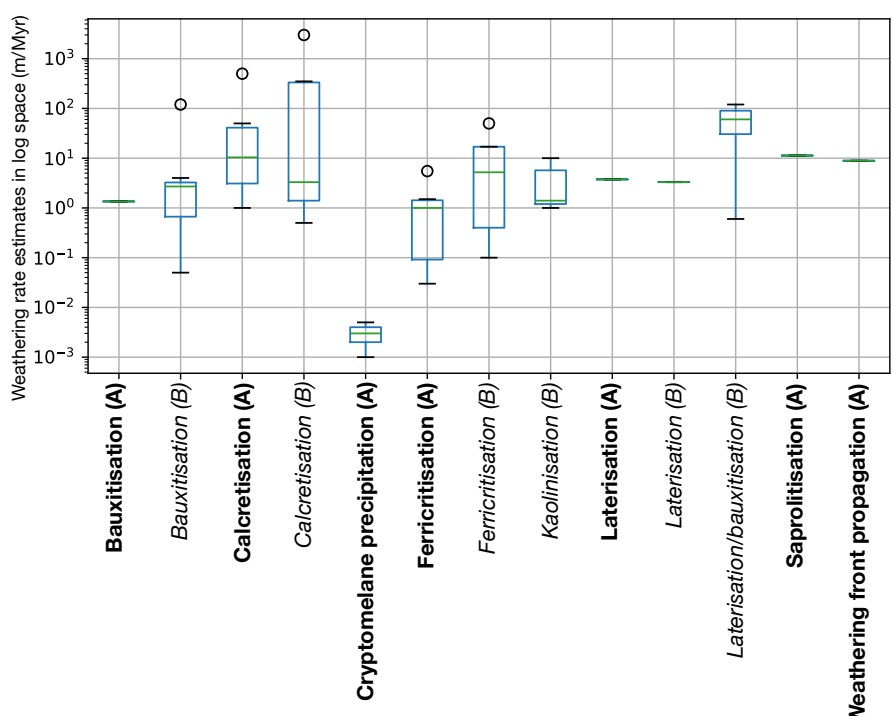

**Figure 4.** Rate estimates for nine categories of weathering processes as derived from literature. The bold (A) categories correspond to rates obtained by dating methods such as $^{40}$Ar/$^{39}$Ar, (U-Th)/He, paleomagnetic dating or radiocron dating (Gac, 1980; Hénocque et al., 1998; Théveniaut and Freyssinet, 1999; Vasconcelos and Conroy, 2003; Théveniaut et al., 2007; Vasconcelos and Carmo, 2018; dos Santos Albuquerque et al., 2020; Netterberg, 1978; Candy et al., 2003; Carmo and Vasconcelos, 2006; Dhir et al., 2010; Heller et al., 2022) and are assumed to be more reliable than the rates obtained by volumetric calculations (Leneuf, 1959; Trendall, 1962; Goudie, 1973; Wright, 1989; Boulangé, 1984; Paquet and Clauer, 1997; Boulangé et al., 1997; Tardy and Roquin, 1992; Tardy, 1969; Horbe and Anand, 2011; Momo et al., 2020; Chen et al., 1988; Taylor and Eggleton, 2001; Fritz and Tardy, 1973; Goudie, 1985) represented by italic categories (B). In this statistical box-plot representation, the green line represents the mean value, the blue box encompasses values from the lower quartile to the upper quartile, while the black whiskers limit at 1.5 of the interquartile range. The black circles represent outliers.

The rates are summarized in Figure 4 as a function of the dominant process. We see that the time span necessary for the formation of an approximately one metre thick duricrust is of the order of a few tens of thousands to millions of years. The formation rate of calcretes appears to be faster than that of ferricretes, laterites, bauxites, or silcretes, as already observed by Fairbridge (2008). Note also that values obtained from volumetric calculations lead to higher rates than those inferred from
geochronological analyses.

**Water table fluctuation range (WTFR)**

The second model parameter, $\lambda$, is the water table fluctuation range (WTFR). It can be constrained by direct observations of the change in water table height over the seasonal cycle, which ranges from a few centimetres to multiple metres, depending mostly on the seasonal variability of rainfall (Tardy, 1993; Taylor and Eggleton, 2001). Water table seasonal fluctuations have

been monitored for e.g. agriculture (Chandra et al., 2015), health, pollution (Deng et al., 2014) or water availability (Balugani et al., 2017). Groundwater studies have registered water table heights all across the globe. Here, we used studies from different climatic regions to calibrate the WTFR in our model, including, for completeness, environments where duricrusts are known to form, as well as others.

Leduc et al. (1997) describe seasonal WTFR in southern Niger in the Sahel aquifer, known for its tropical semi-arid climate,

with a long dry season and short wet season. The registered fluctuation in the 223 analysed wells ranges from 0 to 9 metres. Temgoua et al. (2005) describe a WTFR of about 2 metres in the humid tropical zone of southern Cameroon, where massive ferruginous duricrusts form on footslopes of hills. Hassan et al. (2014) worked in the Sardón Catchment near Castilla y León, Spain. The climate is Mediterranean semi-arid, and most rain events take place during Spring. The yearly WTFR is ~2 metres (Hassan et al., 2014). Hassan et al. (2014) cites a study from Ely and Kahle (2012), in Washington State (USA), where they

analysed hydrological data from the Chamokane Creek basin. Hydrographs from the area showed water level fluctuations in the range 2 to 9,9 metres, with a mean value around 5.5 metres. In South-Korea, Moon et al. (2004) analysed monitoring data from 66 wells across the whole country, where the climate is temperate. Results show maximum variability in WTFR of 0.94 to 3.68 metres. Multiple studies in India registered WTFRs across semi-arid tropical (Sreedevi et al., 2006; Kuruppath et al., 2018; Bhuiyan, 2010) environments controlled by monsoons (Chandra et al., 2015; Kuruppath et al., 2018). Sreedevi et al.

(2006) analysed aquifers in Archean granites in the Maheshwaram watershed. The seasonal WTFR was registered to be 4.2 to 5 metres between May and December (pre- and post-monsoon). Maréchal et al. (2006) have analysed wells in the same region, with WTFR of 2 to 6 metres. In the Rajasthan state, in the Aravalli range, Bhuiyan (2010) observed a mean WTFR in multiple geomorphic classes. He also measured WTFR under pediments (mean WTFR 3.61 m) and buried pediments (mean WTFR 3.09 m). Mean WTFR in this study is 3.35 metres. In another part of India, on the eastern part of the Chotanagpur plateau,

bedrock of Archean to Paleozoic ages formed pediplains (Chandra et al., 2015). WTFRs vary from ~0.2 to 5 metres in that region. Similar values were registered in the Karur district, with pre- and post-monsoon water table heights varying between 0.2 and 6.6 metres (Chandra et al., 2015). Average WTFR for the North-eastern monsoon is 2.5 metres and 1.6 metres for the South-western monsoon (Kuruppath et al., 2018). Bhuiyan (2010); Chandra et al. (2015) registered WTFR under pediments and what could be indentified as ferricretes, with mean WTFR being 3.41 m and 1.77 m respectively. In Southern China, Deng

et al. (2014) analysed groundwater monitoring data from 39 wells in the Jianghan alluvial plain. The study focused on arsenic concentration in the water and water table fluctuation patterns. The climate in that region is sub-tropical, and controlled by monsoons. Registered WTFR is about 1 to 2 metres. In Minas Gerais, Brazil, where active duricrusting and weathering is taking place today, Marques et al. (2020) registered WTFR from 2011 to 2018. A very low seasonal fluctuation is registered, with values around 1 metre. In temperate to cold humid climate zones, groundwater recharge is highly dependable on snow

cover and seasons (Nygren et al., 2020). In Finland and Sweden, Nygren et al. (2020) have analysed data from a 31 year-long monitoring dataset from 264 piezometers. Seasonal WTFRs are the lowest for this region, with values under 1 metre.

Constraining $\lambda$ can thus be done according to climatic regions. In semi-arid environments, the WTFR is of the order of several metres, with maximum values of ~10 metres. In tropical, monsoon driven climates, the WTFR is around 2 to 3 metres, sometimes less. In temperate and cold climates, WTFR is of the order of a metre. A summary of those values is given in
Figure 5.

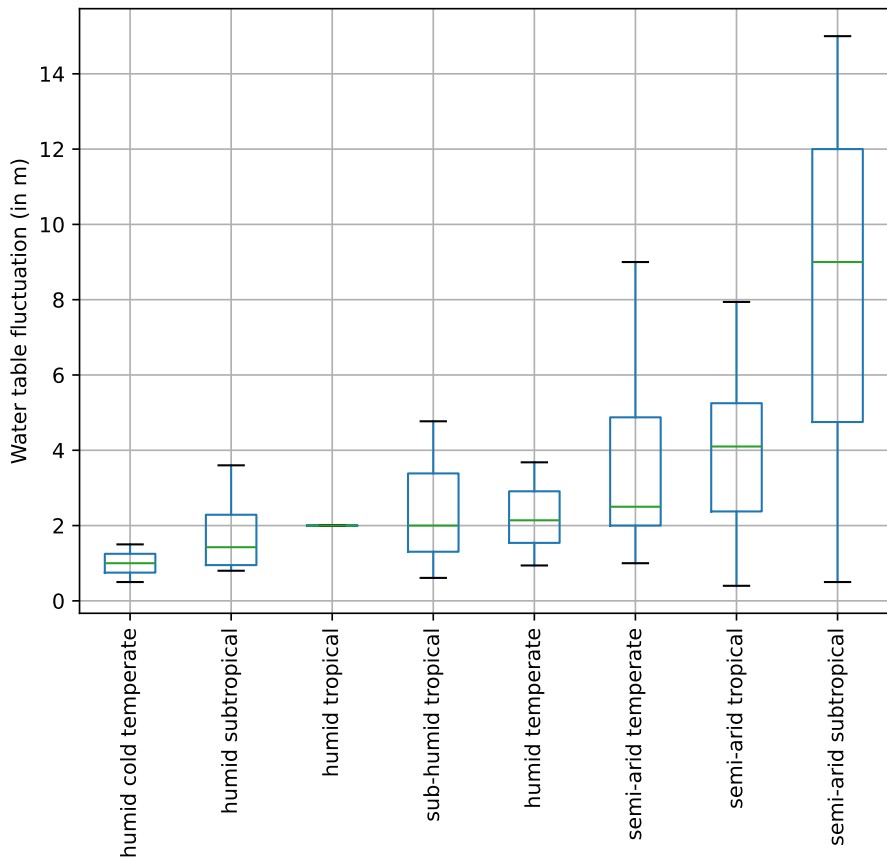

**Figure 5.** Estimates of water table fluctuation range (WTFR) as a function of climatic conditions in modern systems. In this box-plot representation, the green line represents the mean value, the blue box encompasses values from the lower quartile to the upper quartile, while the black whiskers end at the lowest and highest values of the dataset.

## 2.4 A simple model run

To illustrate the behaviour of the new model, we performed an experiment on a hill of length 1000 m. The parameter values are identical to those used in the experiment shown in Figure 2, except for the uplift, which is here a cyclic function alternating periods of uniform uplift (or base-level drop) at a rate $U_0 = 50^{-6}$ m/yr with periods of tectonic quiescence ($U = 0$). Each period

is of duration $\Phi = 500$ kyr long and the model runs for $t_f = 10$ Myrs. $P$ is constant and equal to the reference precipitation $P_{ref} = 5$ m/yr. The other two parameters introduced by the duricrust formation model are set at $\lambda = 3$ m and $\tau = 50$ kyrs.

In Figure 6 we show the resulting hill, regolith and duricrust geometries at the last time step. We see that at the level of the water table, a duricrust has formed. The colour gradient used in the regolith layer represents the values of $\kappa$, the erodability, varying between 1 (beige) and tending to 0 (dark red). When $\kappa$ values are close to 1, the regolith is unaltered. Below the
threshold of $\kappa < \kappa_D$, we consider that a hardened duricrust has formed. We also see that below the water table, the regolith, is unaltered. However, above the water table, the regolith tends to show signs of hardening, with $\kappa$ values comprised between 0.9 and 0.5. The duricrust layer itself ($\kappa < \kappa_D$) forms around the water table, and has a thickness of approximately 4 m (similar to the assumed value for $\lambda$). This behaviour results from the vertical advection of material during periods of tectonic activity in response to uplift (or base-level drop) and surface erosion. Rocks from the intact bedrock are subjected to intense weathering
as they traverse the weathering front at the base of the regolith layer. They are then exhumed towards the surface until they are eroded away. In doing so, they cross the water table and its fluctuation range where they are subjected to hardening by mineral transport and precipitation to form a duricrust if the time spent near the water table is longer than the assumed time for hardening, $\tau$. This, they do only during the period of tectonic quiescence. The partial hardening that is observed in the regolith above the duricrust corresponds to a period of active tectonics and uplift (or base-level drop) during which rocks within the
regolith traverse the water table but do not stay in its vicinity long enough for the hardening process to lead to the formation of a duricrust ($\kappa$ remains smaller than $\kappa_D$).

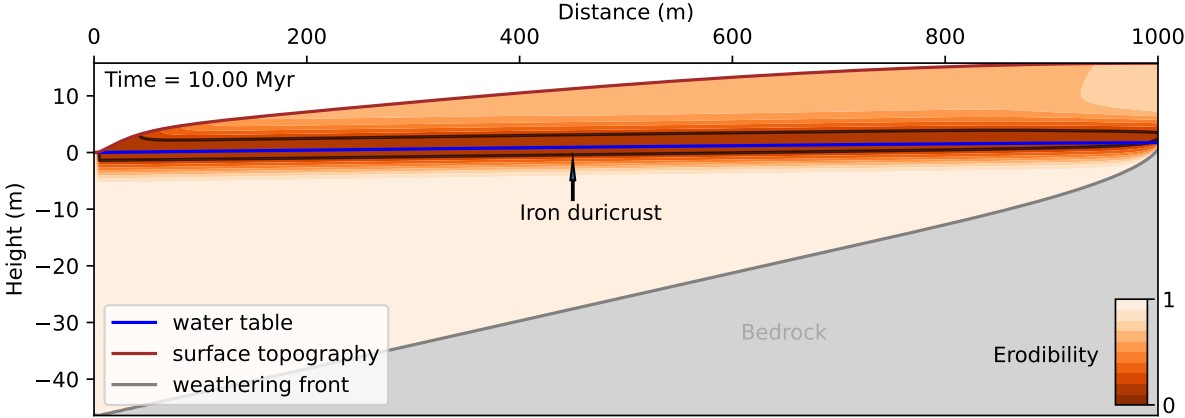

**Figure 6.** Problem geometry with quantities and variables as defined in the new model and Braun et al. (2016), with added duricrust formation, displayed on a steady-state configuration, on a 1000 m long distance. The erodability coefficient $\kappa$ concerns the regolith. The lighter the colour, the softer/more erodible is the material, and the darker the colour, the harder/less erodible the material. The duricrust, in dark red, is highlighted with an outline in black lines, corresponding to the threshold $\kappa_D$. The result shown is at the last time step, which corresponds to 10 Myrs.

Comparing the results of the model with Figure 6 and without (Figure 2) duricrust formation, we see that the solutions at the last time step are similar: the regolith has the same thickness and geometry (thicker near the base of the hill) and the surface topography geometry is similar too. However, this similarity disappears when looking at intermediary times steps, as shown in Figure 7 and in the animation A1 in the supplementary material. In Figure 7, we illustrate one complete cycle of duration $2 \times \Phi$, which includes one uplifting and one quiescent period of time (duration of a cycle). It thus covers a time span of 1 Myr, from 3.80 Myr to 4.80 Myr in the model evolution (Figures 7a and i). The first period (Figure 7a, b, c and d) is marked by no tectonic activity. From Figure 7a to c, we can observe the formation of an indurated layer at the water table level. We see that during the periods of active uplift or base-level drop (Figure 7d, e, f and g), the duricrust that has formed in the vicinity of the water table is progressively exhumed as it is subjected to surface erosion. Being characterised by values of $\kappa$ that are smaller than $\kappa_D$, the hardened layer (duricrust) is less erodible and causes the formation of a topographic step along the hill surface. This step also causes the surface slope beneath the duricrust to steepen, which leads, in turn, to an increase in the hill mean and maximum topography. During the following period of quiescence (Figure 7h and i), the exhumed duricrust temporarily protects the top of the hill from erosion (Figure 7h) until it is removed (Figure 7i), initiating a phase of rapid surface topography lowering. During this quiescence phase (Figure 7i), we also note a new duricrust forming at the water table level.

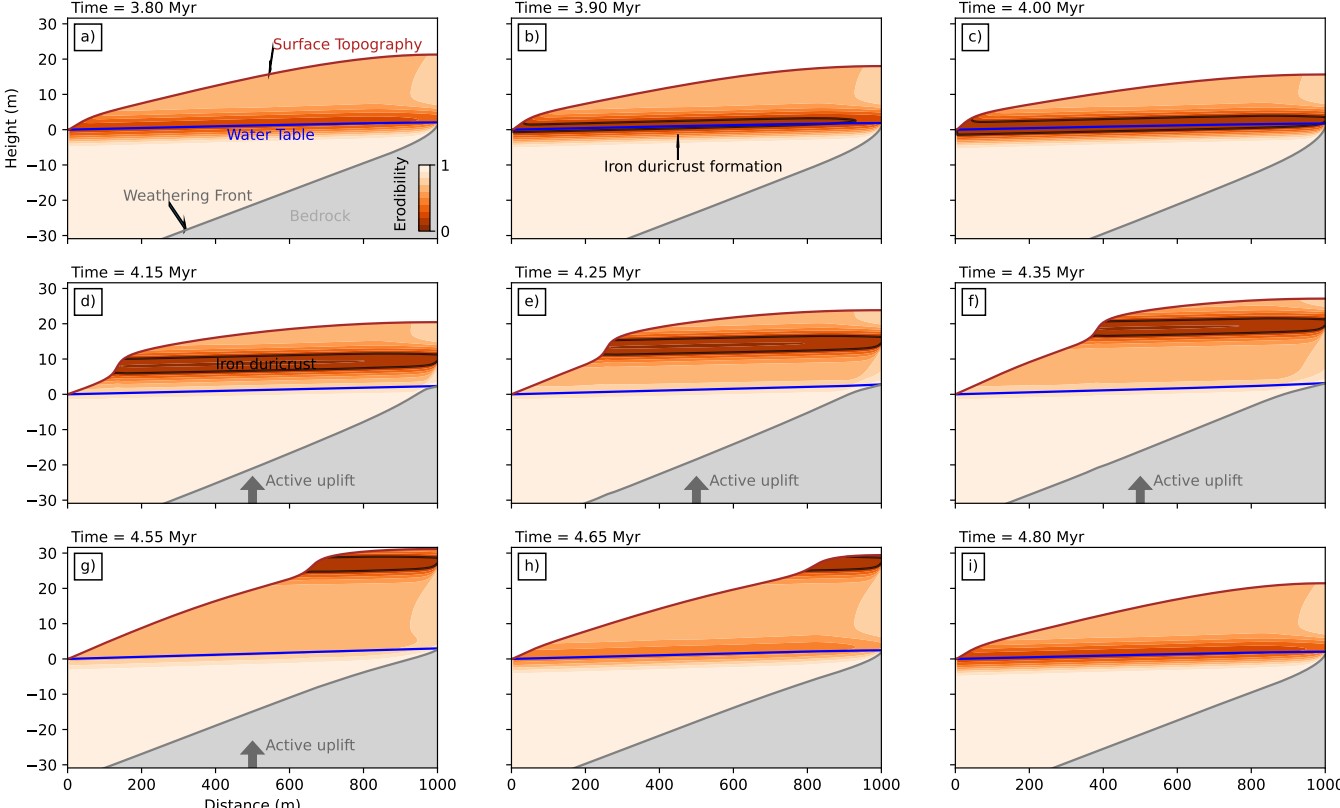

**Figure 7.** From a to i: Problem geometry across time. The chosen time frame is from 4.15 Myrs (a) to 4.80 Myrs (i). The chosen time frame corresponds to one cycle, Φ. Evolution of duricrust formation (a) from the water table to exhumation (i). Sections a to c and g to i are without uplift (or base level drop), while section d to f have active uplift, highlighted by a grey arrow.

This evolution of the topography is summarized in Figure 8 where we compare the time evolution of the maximum topography (i.e., at the top of the hill) predicted in the reference model run shown in Figure 6 with the prediction of an almost identical model experiment in which we prevented the formation of duricrust by setting $\tau$ to a very large value. We see that the experiment that includes the duricrust is characterised by higher topography but, interestingly, both experiments show a rapid decrease in maximum topography immediately following the end of the period of tectonic activity. There is almost no delay in the decay (or preservation) of the topography caused by the presence of the duricrust.

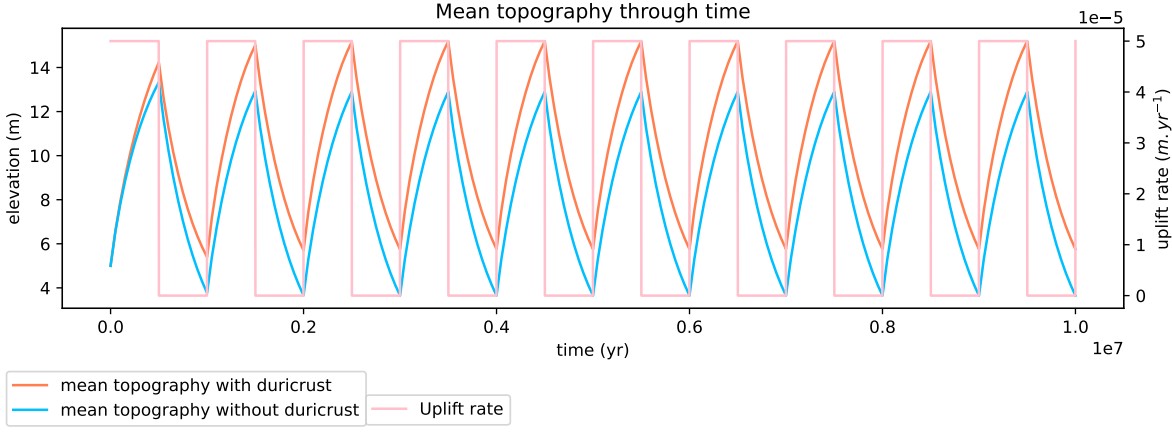

**Figure 8.** Mean topography elevation for two model experiments responding, through time, to the same periodical uplift rate. In orange: mean topography elevation through time with duricrust formation. In blue: mean topography elevation through time without duricrust formation.

The geometries that can be created by the model are varied and we cannot reproduce them exhaustively here. One case that is very relevant to many natural systems involves the presence (or preservation) of a family of duricrusts in the same landscape. Although this is likely to take place on a much greater scale than that of a single hill, we show in Figure 9a results of a model run in which three generations of duricrust have been preserved at the end of a period of quiescence. This has been achieved by lowering the value of the characteristic time $\tau$ leading to more rapid and therefore more intense lowering of the value of $\kappa$ in the vicinity of the water table. The resulting duricrusts have become stronger and have been preserved during two cycles of uplift/quiescence and erosion.

In Figure 9b we show the computed ages of the duricrust and the regolith. In the hardened layers (i.e., where $\kappa < \kappa_D$) the ages correspond to the time where the hardening took place. In the unaltered or partially altered regolith (i.e., where $\kappa > \kappa_D$) ages correspond to the time where the rocks crossed the weathering front at the boundary between the regolith layer and the underlying bedrock. In Figure 9b, we see that the youngest duricrust is still forming at the water table level, while the oldest duricrust is now at the top of the hill and is almost 2 Myrs old. The middle duricrust has an age that is the mean of the other two. While the evolution of duricrusts in the regolith enables aging of uplifting duricrusts, regolith layers also age. Regolith ages depend on the position of the weathering front. Thus, the youngest regolith is at the base of the weathering profile, while the oldest regolith is outcropping at the top of the hill. Duricrust and regolith ages do not correspond, having inter-located layers of older regolith in-between two duricrusts. However, it is interesting to note that the difference between regolith and duricrust ages is greatest near the base level and decreases beneath the hill top.

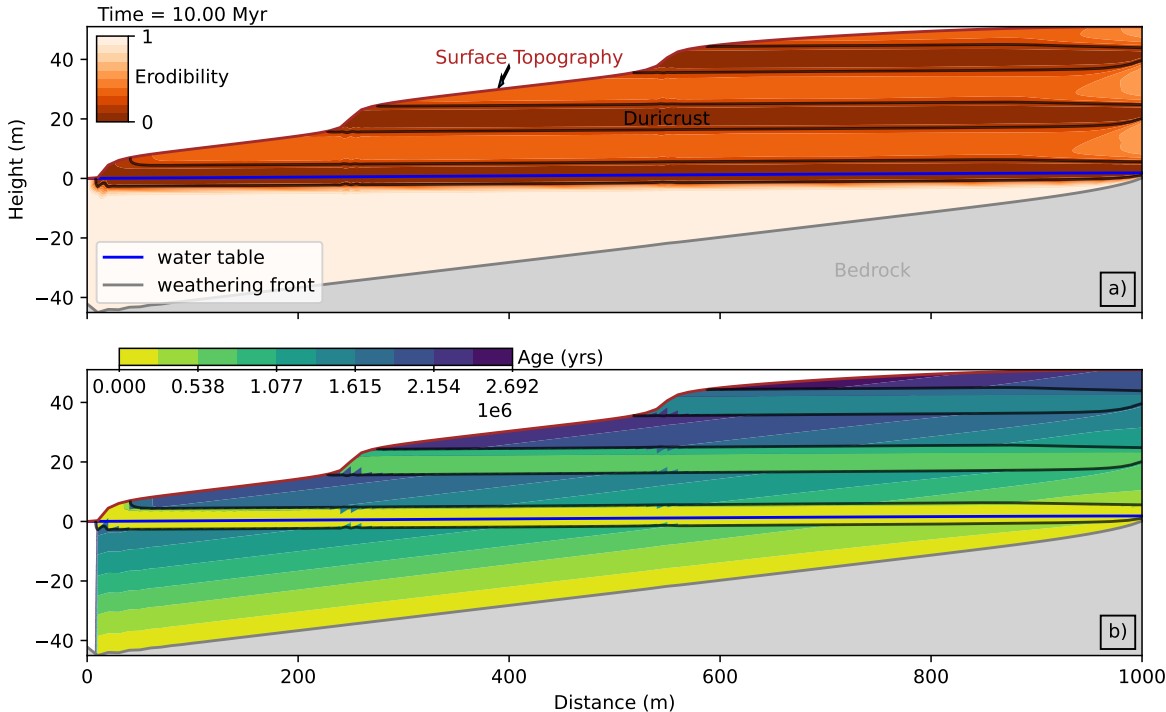

**Figure 9.** a) Model scenario with three duricrusts (outlined in black, dark red), formed in three different generations according to the equations given in Figure 3.; b) Simulation of regolith and duricrust ages derived from the formation equations ruling the new model. The younger the age, the lighter the colour of the object. The older the age, the darker the colour of the object. Note that in the duricrust layers, the age corresponds to the age of the material when it passed through the water table; in the rest of the regolith, the age corresponds to the age of the material when it passed through the weathering front.

## 2.5 Dimensionless numbers and mapping of model behaviour

545 We now derive the basic conditions necessary for the formation of a duricrust. Firstly, for a duricrust formed during the quiet period, the regolith material must spend sufficient time within the fluctuation range of the water table (WTFR) , in comparison to the characteristic time scale for hardening, $\tau \times \frac{P}{P_{ref}}$. This leads to the definition of a first dimensionless number, that we will call $W$ and define as:

$$W = \frac{\Phi}{\tau} \times \frac{P}{P_{ref}} \qquad (8)$$

550 $\Phi$ is here the duration of one of the cycles, but it must be understood in a more general context as the duration of the quiet period during which the regolith hardening takes place (regardless of the duration of the tectonic period). We see that for a duricrust to form, $W$ must be large.

Secondly, for a duricrust to form during an actively uplifting period, the same condition, i.e., that the time spent by the regolith within the WTFR, in this case equal to $U \times \tau$, is equal or larger than the characteristic time scale for hardening, leads to the definition of another dimensionless number, which we called $R_t$:

$$R_t = \frac{\lambda}{U \times \tau} \times \frac{P}{P_{ref}} \tag{9}$$

And, again for a duricrust to form requires that $R_t$ be large.

We note that the two numbers, $W$ and $R_t$, are indeed dimensionless. To assess the exact range in the $[W - R_t]$ space over which duricrusts can form, we perform a large number of model experiments varying many of the model parameters simultaneously, including, $\lambda$, $\tau$, $U$, $\Phi$ and $P$. The results are shown in Figure 10a where each of the 1296 model runs we performed has been represented by coloured circles in the $W$-$R_t$ space, where the colour is proportional to the logarithm of the maximum value of the erodability factor, $\kappa$, computed by the model and the duricrust formation threshold marked by $\kappa_D$. We see that the formation of a duricrust is favoured for high values of both $W$ and $R_t$ and that for the arbitrary threshold value of $\kappa_D$=0.2 as used in the above experiments, the conditions for duricrust formations are approximately:

$$W > 1 \quad \text{and} \quad R_t \geq 0.1 \tag{10}$$

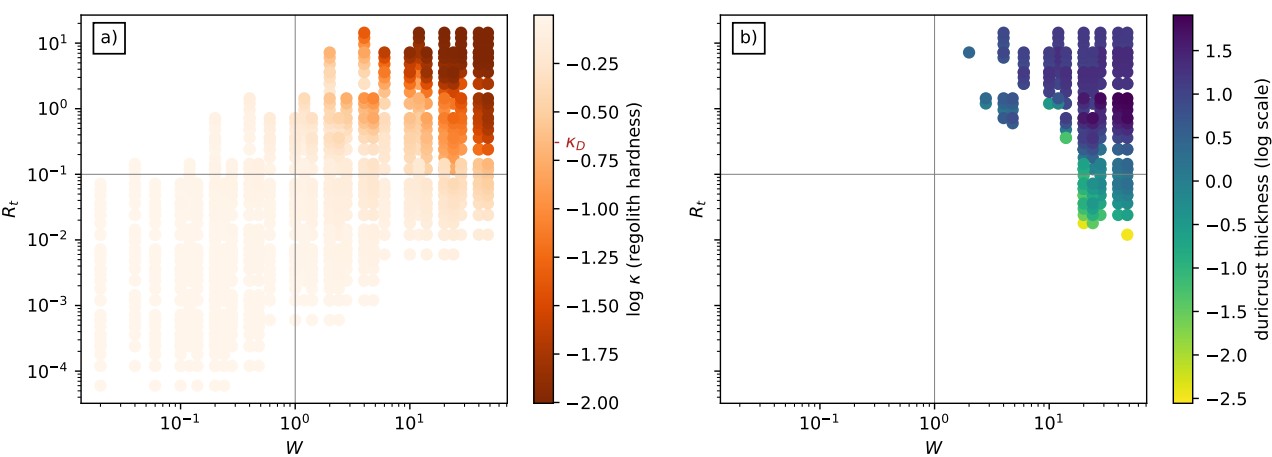

**Figure 10.** (a) Statistical study of 1296 scenarios illustrated in a log space scatter plot along dimensionless numbers $R_t$ and $W$. Dots represent occurrence or not of duricrust formation according to the input parameters. Regolith hardness, or erodability, is represented through the colour scale: the lighter the colour, the softer the regolith, the darker the colour, the harder the regolith (creation of duricrusts). Formation conditions are met when approximately $R_t > 0.1$ and $W > 1$, according to the duricrust formation threshold, defined by $\kappa_D = 0.2$ (in the top right space defined by the grey outlines); (b) Duricrust thickness illustrated in the same log space scatter plot. The lighter the colour, the thinner the duricrust. The darker the colour, the thicker the duricrust.

In Figure 10b, we show the duricrust thickness values computed from the same set of numerical experiments. We see that computed thicknesses are shown only above values for $W$ and $R_t$ approximately above 1 and 0.1, which also points to a duri-

crust formation threshold of $\kappa < \kappa_D$. Thicknesses range from centimetres to metres. The thinnest duricrusts are predominantly registered for lower $R_t$ values, while the thickest values tend to be for high values of $W$ and values of around 1 for $R_t$.

 ## 3   Discussion

### 3.1   Model behaviour

We have developed a model for the formation of duricrusts by regional transport of ions and their precipitation in the fluctuation range of the water table (WTFR). Although based on sound physical processes, our model is purely parametric in that it depends on the values of heuristic parameters including the assumed range of seasonal variations in water table height, $\lambda$, and the characteristic time for precipitation of the hardening agent, $\tau$, assumed to be inversely proportional to precipitation rate. This simple model reproduces the most widely accepted conditions for the formation of duricrusts: a wet, yet variable, climate in a relatively stable tectonic environment.

The model also shows that a period of uplift (or base level fall) is necessary to expose the duricrust to the surface. By modifying the surface erodability constant $K_D$ (here the coefficient of transport on hill slopes) in proportion to the hardening parameter, $\kappa$, used to parameterise the progressive transformation of the regolith into hardened material, the model is able to reproduce the formation of one or several duricrusts that resist erosion and therefore protect the underlying regolith. According to our model, the age of formation of a duricrust can be equated to the time since the duricrust left the WTFR. In Figure 9b, we show computed ages for the model run shown in Figure 9a. We see that regolith ages are always older than duricrust ages, and partially altered/hardened regolith is located above duricrusts and not below. This happens when the environment's activity changes and starts to uplift. If we observed inverted ages of duricrust generations, i.e. younger duricrusts on top of older ones, and altered regolith below duricrusts and not above, the recorded activity would not be characterized as uplift but rather as subsidence.

The absolute value of the hardening parameter is, however, arbitrary and determined by the ratio of the assumed characteristic time scale, $\tau$ and the duration of a period of tectonic quiescence. We hypothesise that a critical value of $\kappa = \kappa_D = 0.2$ corresponding to a decrease in surface erodability by a factor of 5 is such that it causes noticeable variations in topography that allow us to define the range of model parameters causing the formation of a duricrust as observed in the field.

### 3.2   Model parameters

The thickness of the duricrust depends strongly on the value of the parameter $\lambda$ which indicates that the seasonality of rainfall and thus the WTFR are the dominant factors that determine the thickness of the predicted duricrust. This correlates with observations in Africa (Tardy, 1993; Leduc et al., 1997), India (Sreedevi et al., 2006; Bhuiyan, 2010; Chandra et al., 2015; Kuruppath et al., 2018), Europe (Hassan et al., 2014; Balugani et al., 2017; Nygren et al., 2020), Asia (Moon et al., 2004; Deng et al., 2014) or Brazil (Allard et al., 2018). Depending on the region, long arid periods are observed with intensive fast wet seasons, sometimes monsoon driven, where duricrusts form and can attain up to more than 10 metres in thickness (Goudie

(1985); Tardy et al. (1991); Tardy (1993) and Figure 1). On the other hand, where tropical, temperate or cold environments do not allow for important arid periods, duricrusts form, but are thinner (a few cm to a few m). We note, also, that thicker duricrusts can be generated in the case of a very slow, yet continuous uplift (or base level fall)/subsidence that would cause the water table to affect a much greater range of the regolith layer. As shown in Figure 10b duricrust thickness also varies according to the dimensionless parameters $W$ and $R_t$. Duricrusts are the thickest for high values of $W$ and $R_t$, but specifically more for values around 1 for $R_t$. Precipitation prone environments like tropical areas are located on the right side of Figure 10. This result is highly interesting as it is in apparent contradiction with what we stated earlier, where the thickest duricrusts were created under semi-arid conditions as the WTFR is highest in semi-arid areas (Leduc et al., 1997). Thus, dependence of duricrust thickness on $\lambda$ may vary depending on external conditions. The WTFR could be more prevalent under drier conditions, as our models show a correlation between $\lambda$ and duricrust thickness as can be seen with Figure 6, while other processes dictate duricrust thickness under tropical conditions. Tectonically active areas (low $R_t$ values) are not prone to duricrust formation and when they do, tend to create thinner crusts (lower part of Figure 6).

As mentioned above, the value of the parameter $\tau$ is arbitrary. Constraining it by direct observations or the results of laboratory experiments on the kinetic of chemical reactions or the saturation in the brine is difficult. Geochronological methods e.g. Netterberg (1978); Hénocque et al. (1998); Carmo and Vasconcelos (2006); Küçükuysal et al. (2011); Vasconcelos and Carmo (2018); Heller et al. (2022); Campos et al. (2023); Ritter et al. (2023) on iron oxides, cryptomelane, or radiocarbon dating are a mean to provide approximate values for $\tau$, through the range of ages that can be estimated on a single nodule (Heller et al., 2022). This leads to postulate that a 'generic' value for $\tau$ could be in the range 50 - 1000 kyrs, although it is likely to be highly dependent on precipitation and duricrust type.

It is also worth mentioning that inferred rates by volumetric calculations, compared to rates derived from geochronological data are faster (Figure 4). Thus, chosen rates have to be considered according to the method used for the inferred rates. Calcrete formation rates are also faster compared to ferricrete and silcrete formation rates, which also changes the main range of $\tau$ values according to the modelled duricrust type.

Note that our model allows for a linear dependence of $\tau$ on precipitation, which we included in the model parametrisation by introducing a characteristic precipitation rate (taken to be 1 m/yr) to which the specified value of $\tau$ corresponds.

### 3.3 Erosional time scale

During phases of tectonic activity or base-level fall, duricrust layers are uplifted from the water table height to the top of hills (Figure 7). While being uplifted, we can see that the regolith layers as well as the duricrust layer are eroded with time, resulting in only a small patch of duricrust remaining at the top of the hill at the end of the uplift period (Figure 7g). When comparing duricrust hills with homogeneous regolith hills, we can see a difference in morphology and topographic height. However, as discussed earlier and shown in Figure 8, there is almost no delay in erosion due to the presence of a duricrust, which seems contra-intuitive: if a hard layer protects soft rocks, it should take longer to erode the system. Many authors have previously described duricrusts in lateritic or regolith profiles (e.g. Tardy (1993); Taylor and Eggleton (2001)) as a protecting layer that

should slow down erosion of the underlying topography/hill (Figure 1). The very high longevity of duricrusts is not observed in our model, and we now proceed to explain why.

In our model, a hill containing a duricrust layer is regarded as a layered material that is subjected to diffusion (i.e., a system where transport of material is proportional to slope). The top and bottom layers have normal diffusivity, while the central layer has a diffusivity lowered by a factor $\kappa$. We ran a series of 1D diffusion models representing this situation, starting from an initially sinusoidal hill of arbitrary height and length but varying the position, $p$, (one quarter, half and three quarters of the height of the hill), thickness, $d$, (5, 10 and 20% of the height of the hill) and hardness, $\kappa$ (in a logarithmic ratio of 1 to 0.001) of the embedded horizontal layer. The computations are performed using 1001 nodes and 20001 time steps. The diffusion equation is solved using an implicit method that requires the solution of a tri-diagonal system of equations.

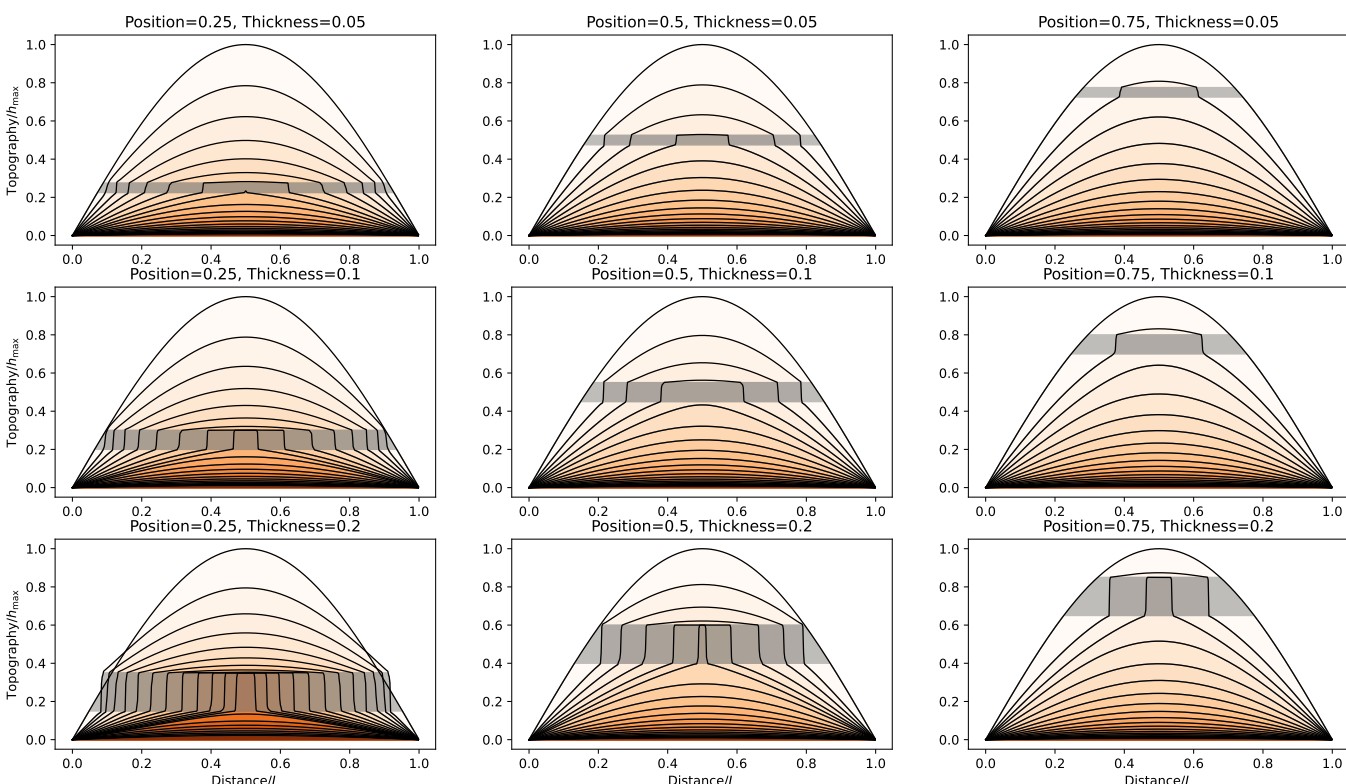

**Figure 11.** Effect of a duricrust on a sinusoidal hill. Computed hill geometries for various thickness and position of an embedded duricrust of hardness $\kappa = 0.001$. Black lines give the solution at equally spaced time intervals. Grey-shaded area gives the position and thickness of the duricrust.

In Figure 11, we show the results of nine of those experiments with a hardness parameter ($\kappa$) of 0.001 corresponding to a layer that is 1000 times harder to erode than its surroundings. In each panel, the thick black lines show the geometry of the hill at equal time intervals. The grey-shaded area corresponds to the hardened layer or duricrust. We see that the material on top of the duricrust rapidly erodes, but that the lowering of the hill is significantly postponed by the presence of the duricrust. When

the duricrust is exposed at the surface, it undergoes little to no vertical erosion but is progressively eroded by lateral erosion along its margins, also observed by Taylor and Eggleton (2001). Once the duricrust has been eroded, the hill erosion resumes at a more rapid rate. In Figure 12, we show the evolution with time of the maximum elevation of the hill in the case where the duricrust is thick (a fifth of the height of the hill) and located near the base of the hill (one quarter of the height of the hill). We see that, depending on the hardness of the duricrust, the system spends more or less time at a maximum height equal to the

position of the top of the duricrust. For each curve, we computed the erosion timescale or time it will take for the hill to reach 20% of its original size. These timescales are shown by stars on each curve of Figure 12.

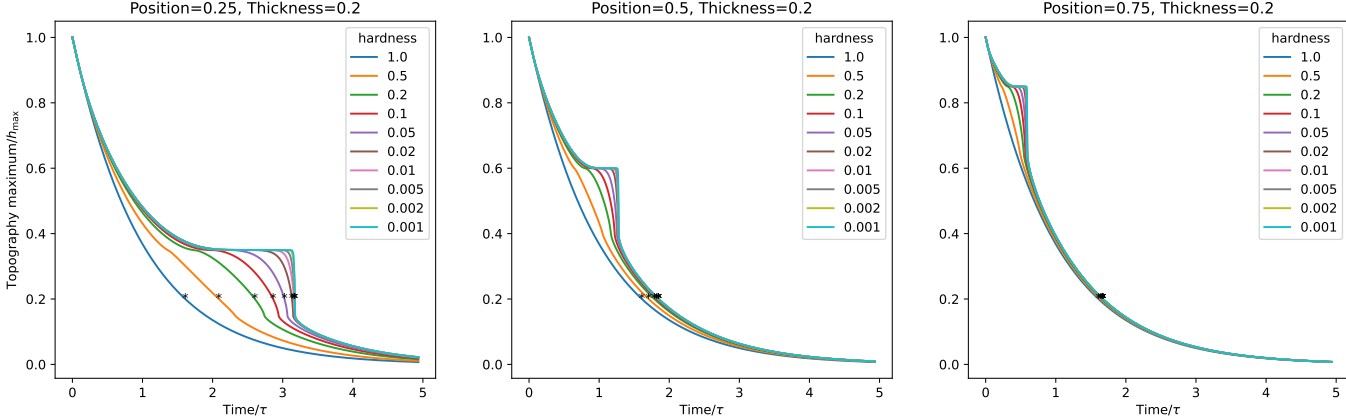

**Figure 12.** Maximum hill topography as a function of time for different hardness coefficient, $\kappa$ and position of the duricrust. The stars correspond to the erosion timescale, or time it takes for the hill to be eroded to 20% of its original maximum height.

In Figure 13, we show the erosion timescales, normalized by the erosion timescale of a hill containing no duricrust, as a function of various parameters, i.e., $d$, $p$ and $\kappa$. We see that the erosion timescale increases with the duricrust layer thickness and hardness (lower $\kappa$ value). The dependence on the hardness is, however, very non-linear. The increase in the response

timescale does not exceed a factor of 2, even for a reduction in diffusivity (or increase in hardness) of 1000. This explains why a 'normal' duricrust, i.e., that represents only a few percent of a hill total thickness, does not strongly alter the erosional timescale of the hill, confirming the results previously presented in Figure 8. It also supports our arbitrary choice of a hardness factor of 0.2 (decrease in erodability by a factor 5) to define a duricrust layer.

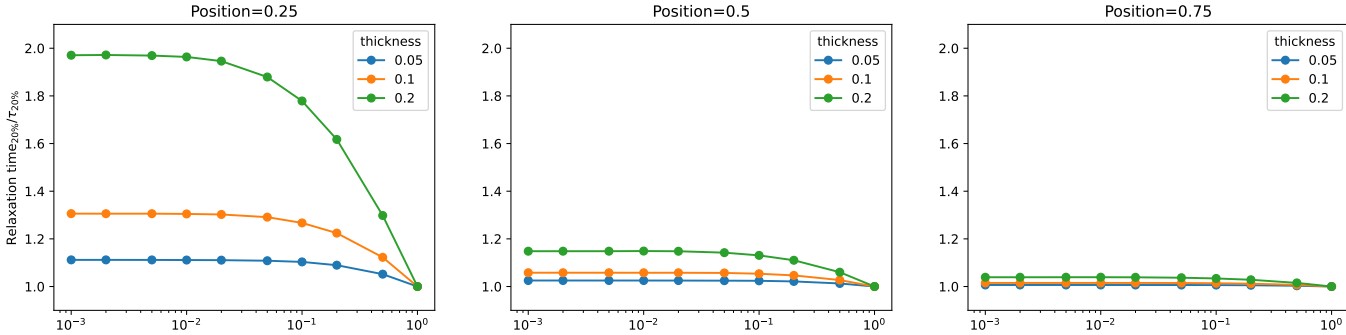

**Figure 13.** Increase in erosional timescale due to the presence of a duricrust, as a function of the assumed hardness of the duricrust (factor $\kappa$ for three different values of the duricrust thickness (green, orange and blue lines) and three positions of the duricrust layer (from left to right). The position and thickness are relative to the hill height.

We also see that the position of the duricrust has an effect on how it affects the erosional timescale of the hill. The closer it
is to the base of the hill, the more it 'protects' the hill from being eroded. This result is interesting as it supports the concept
that duricrusts protect topographic features, but not in proportion to their apparent strength: i.e., a duricrust that is 1000 times
harder than the surrounding hill can, at most, delay the erosion of the hill by a factor 2, even if it fills as much as 20% of the hill
thickness. The reason for this result is illustrated in Figure 14 in which we show the distribution of erosion rate across the model
at an arbitrary time step. It shows that the erosion rate at the top of the duricrust is nearly four orders of magnitude smaller than
the erosion rate along its edges. This explains the relatively low erosion rates (0.01 to 0.5 m/Myr) measured by cosmogenic
isotopes methods (Fujioka et al., 2010; Shuster et al., 2012; Monteiro et al., 2017, 2018; Struck et al., 2018; Bierman et al.,
2014) along the top surface of duricrusts, but their relative low exposure age (a few million years) compared to their formation
age (several tens of millions of years) (Monteiro et al., 2017, 2018), implying that duricrusts do slow down surface erosion rate
once they are exhumed to the surface but do ultimately get eroded within a few million years. This also explains the observation
that fragments are often found at low elevations in the vicinity of exposed duricrust (the area of deposition at the foot of the
ferricrete in Figure 1a and the deposition area in pink in Figure 14) and become the source of recemented duricrusts.

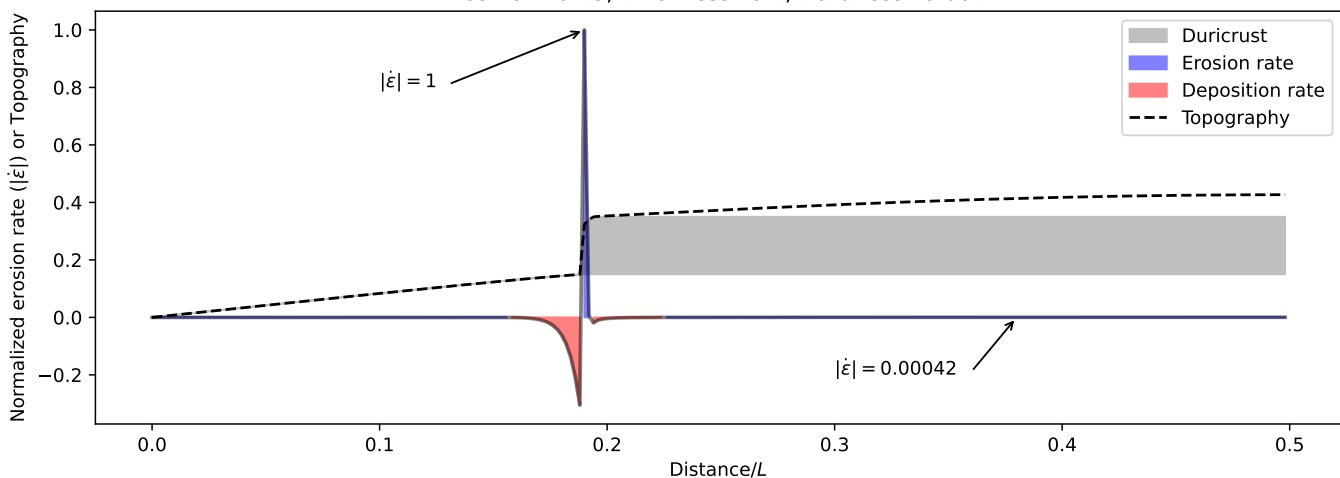

**Figure 14.** Instantaneous distribution of computed erosion rate (black solid line) across a hill containing a duricrust with a hardening coefficient values of 0.001, a thickness of 20% of the hill height and positioned at a quarter of the distance between hill bottom and top. The duricrust id highlighted in gray and the topography of the hill is shown as a dashed line. Note the very large difference in erosion rate between the top of the duricrust and its edge and the deposition of duricrust fragments on the side of the hill beneath the duricrust.

### 3.4 Important dimensionless numbers

We have shown that the model behaviour can be described in a dimensionless space containing two parameters, namely $R_t = \frac{\lambda}{U \times \tau} \times \frac{P}{P_{ref}}$ and $W = \frac{\Phi}{\tau} \times \frac{P}{P_{ref}}$. The first controls how long a section of the regolith layer stays in the vicinity of the water table

while uplifting at a rate $U$ and the other how long the material stays while not uplifting, in comparison to the characteristic time scale, $\tau$. Both dimensionless numbers also incorporate the dependence of $\tau$ on mean precipitation rate. As expected, the region of this dimensionless parameter space that is conducive to duricrust formation corresponds to conditions of tectonic stability, as previously described by several authors such as Tardy (1993); Tardy and Roquin (1998); Vasconcelos and Conroy (2003); Taylor and Eggleton (2001) and high mean precipitation rate during a set, short, period of time, in agreement with the

work of Tardy et al. (1991); Tardy (1993), which defines very detailed conditions for the creation of duricrusts, but needs to be considered with caution, as depending on the duricrust type, e.g. calcretes, a drier environment may be considered the norm (Nash, 2011).

It is, however, worth noting that this region of parameter space corresponds also to conditions which, according to Braun et al. (2016), lead to the formation of a regolith layer that is thickest near its base level (or thins towards the top of the hill). As

mentioned above, this corresponds to situations where the dimensionless number $\Gamma$ is smaller than the ratio $\Omega^2/(\Omega - 1)$. This point is highlighted in Figure 15 where we show contours of the value of the ratio $\Gamma(\Omega - 1)/\Omega^2$ (left column) and the maximum value of the hardening coefficient, $\kappa$, (right column) as a function of the assumed uplift rate, $U_0$ and precipitation rate, $P_0$, both assumed to be constant in space and time, for each of a range of $9 \times 9 \times 3 = 243$ experiments, in which those parameters were

systematically varied across natural ranges. We see that for large values of $\tau$, no duricrust form as the maximum value of $\log_{10}\kappa$ does not reach -0.7 ($= \log_{10}(0.2)$).

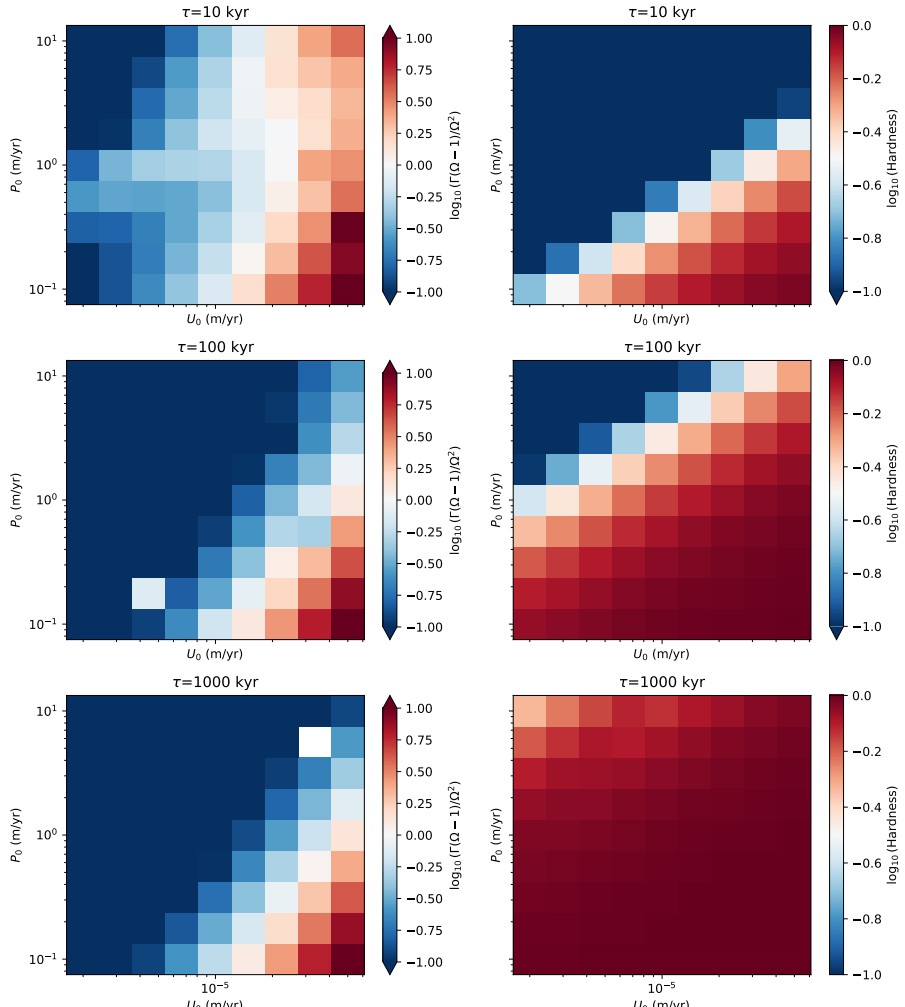

**Figure 15.** Left column: ratio $\Gamma(\Omega - 1)/\Omega^2$ controlling the shape of the regolith layer (see Braun et al. (2016) for definition) as a function of the assumed uniform and constant uplift rate, $U_0$, and precipitation rate, $P_0$, for three values of the duricrust formation characteristic time scale, $\tau$. Right column: contours of the maximum hardening coefficient, $\kappa$, for the same parameter values.

We note, however, that, for small values of $\tau$, the simple pattern of the ratio $\Gamma(\Omega - 1)/\Omega^2$ increasing with high uplift rate and low precipitation rate (leading to greater regolith thickness beneath the top of the hill than at its base level) is perturbed at mid-precipitation values. When a duricrust forms, it decreases the surface transport coefficient, which increases the surface slope. This, in turn, causes $\Omega$ to increase and $\Gamma$ to increase. However, as $\Gamma$ varies as the square of the slope, while $\Omega$ varies in direct proportion to the slope, the ratio $\Gamma(\Omega - 1)/\Omega^2$ increases leading to the formation of a duricrust. Thus, the formation of

a duricrust causes the regolith profile to thin near its base and to thicken towards the top of the hill. This feedback can be quite strong for low values of $\tau$: compare the contours of $\Gamma(\Omega-1)/\Omega^2$ for $\tau = 10$ kyr (top left panel in Figure 15) with those for $\tau = 1000$ kyr (bottom left panel in Figure 15).

### 3.5 Specification for different duricrusts

Different types of duricrusts form through hydrological processes (Taylor and Eggleton, 2001). One possible development and use of our model would consist in adjusting the value of the key rate parameter, i.e., $\tau$ (formation time), for each type of duricrust, namely ferricretes, calcretes, silcretes or other types of hardened layers that are likely to form according to the hydrological model. It is known that different types of duricrusts form in different environments, most likely in an optimum moisture/climatic gradient in time and space (Webb and Nash, 2020; Taylor and Eggleton, 2001; Khalifa et al., 2009; Momo
et al., 2020; Mather et al., 2019; Watson, 1988). Some duricrusts however, e.g. silcretes, evolve in a broad range of environments (Webb and Nash, 2020).

   Adapting the model to those different types of duricrusts would thus mean adapting the climate parameters of the model, i.e., the reference precipitation $P_{ref}$. For example, scenarios for calcretes, which are known to form in arid environments with less than 500 mm/yr precipitation (Achyuthan, 2004), have to be calibrated to such conditions, by decreasing $P_{ref}$. For silcretes,
which form in a broad range of climatic environments (Webb and Nash, 2020), climatic calibration will not provide duricrust formation boundaries. For ferricretes, which form under humid conditions, a high $P_{ref}$ value is necessary to capture their presumed formation environment (Tardy, 1993). Using the model, we could then map the tectonic and climatic scenarios under which each type of crust forms and explore potential differences in their long-term preservation.

   An obvious step in improving our model would be to incorporate variations (in space and time) of the hydraulic conductivity
in response to the formation of duricrusts. This could be done by altering the hydraulic conductivity in proportion to the relative duricrust/regolith thickness or by solving the hydraulic equation in two dimensions ($x$ and $z$) and include vertical variations in hydraulic conductivity. This later option would render the model less computationally efficient.

   Incorporating geochemical processes or combining the physical model with existing geochemical models and taking into account the different processes described in e.g. Soler and Lasaga (1996) for bauxites, Lichtner and Biino (1992) for copper-
ferricretes or Wang et al. (1993) for calcretes would be a future perspective to consider. This would improve the accuracy of our model when applied to a specific case study where variables such as the parent rock composition, vegetation type, water temperature and pH could be constrained. It could also provide better theoretical justification for the value of the heuristic parameters we introduced ($\tau$ and $P_{ref}$) or replace them with a larger set of chemical constants.

### 4   Conclusions

We have developed a new model for the formation of duricrusts by adding a fourth component to an existing model for regolith formation by Braun et al. (2016) with the purpose of investigating how duricrust formation affects surface erosion and the potential feedback it has on regolith formation and evolution. It can also be seen as a first step towards validating the transport

or hydrological duricrust formation hypothesis by investigating whether it can or cannot reproduce the major characteristics of duricrusts, including their aspect and geometry, but also the environments in which they are known to form in.

Using the model, we have shown to which degree duricrust formation can alter surface topography but also the water table position and the regolith geometry through complex feedbacks. Braun et al. (2016) demonstrated the strong control that surface slope exerts on regolith geometry, as well as uplift rate or base-level drop and precipitation rate. Some of these factors, namely uplift and precipitation rates, are also controls on duricrust formation, but the main feedback arises from the control of duricrust formation on surface slope.

We demonstrated that, under the assumption that duricrust formation is controlled by the dissolution, transport, and precipitation of minerals from distant sources (the so-called transported or hydrological hypothesis) duricrusts preferentially form under quiet tectonic conditions. Furthermore, the main control on duricrust thickness is the assumed WTFR. For the duricrust to become exposed at the surface requires a phase of tectonic uplift (or base level fall) followed by denudation. So the most likely environment to produce a series of duricrust layers must involve a sequence of tectonic/uplift events punctuated by
periods of tectonic stability.

    Our parametrisation introduces two heuristic parameters that can be calibrated using independent constraints, although the range of possible values for the parameters are relatively large. Combining these parameters with those representing external forcings (precipitation rate and uplift rate), one can define two dimensionless coefficients, namely $R_t$ and $W$, that can be used to map the conditions under which duricrust formation is most likely to take place. We have shown that when $W \geq 1$ and
$R_t \geq 0.1$ approximately, duricrust formation is possible. Below these critical values, the conditions are usually not met.

    We have shown that, although intrinsically very resistant to erosion, duricrusts protect landforms in a proportion that is lower than their strength would imply. This is because of their relative low thickness in comparison to topographic relief. We have shown that it is not only the intrinsic strength (or resistance to erosion) and thickness that control how long a duricrust will resist erosion, and therefore protect a hill, but also the position of the duricrust with respect to the summit of the hill. Our
prediction is consistent with the widespread observation that, when they are exhumed, duricrusts tend to be located near the summit of a hill. It also explains why it is relatively rare to preserve extensive duricrust layers at the surface.

    An important improvement of the model would be to adapt parameters for different types of transport forming duricrusts in including more detailed geochemical components defining different duricrusts. We could then investigate how they are affected by the same tectonic uplift and climate scenarios and, potentially, how they would differ in their preservation potential.

Another important development will be to generalise this model to two dimensions. This will enable a more realistic prediction of the extent and geometry of duricrusts and their surface expression, and, potentially, simulate the formation of inverted topographies such as seen in southern regions of Australia.

    However, the priority for us is now to develop a model that represents the competing formation hypothesis for duricrust formation, namely the weathering/laterisation or in-situ hypothesis that will allow us to uncover the main differences in duricrust
geometries and rate of formation predicted by the two models and, hopefully, determine which of the two hypotheses is most appropriate or, more likely, the conditions that will favour the formation of duricrusts through one mechanism or the other.

*Code availability.* Caroline Fenske. (2024). CarolineFenske/Duricrusts: Initial Release (v1.0). Zenodo. https://doi.org/10.5281/zenodo.10523101

*Author contributions.* Caroline Fenske prepared the manuscript with contribution of Jean Braun, Cécile Robin and François Guillocheau. Caroline Fenske and Jean Braun developed the model, Caroline Fenske performed the simulations for the new model. François Guillocheau and Cécile Robin have given important contributions regarding the geochemistry, sedimentology and regolith knowledge, development of the idea of the main hypothesis.

*Competing interests.* No competing interests are present

*Acknowledgements.* The project has received funding from the European Union's Horizon 2020 research and innovation program under the Marie Sklodowska-Curie grant agreement No 860383. The authors thank J. Webb and P. Vasconcelos for very constructive comments and suggestions on an earlier version of this manuscript.

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
