# Peer review of "A numerical model for duricrust formation by water table fluctuations"

_EGUsphere, 2024_

## Referee Comment (RC2)

**A numerical model for duricrust formation by water table fluctuations**

By Caroline Fenske, Jean Braun, François Guillocheau, and Cécile Robin

Review by Paulo Vasconcelos
Feb 14, 2024

Fenske et al. propose to develop and apply a numerical model for iron duricrust formation by water table fluctuations. They interpret their model to be based on sound physical and chemical processes and apply the model to test whether the formation of ferricretes slows erosion. Their main conclusion is that ferricretes have a limited and transient role in slowing erosion and protecting the underlying weathering profiles.

A model that properly couples chemical weathering and landscape evolution is long overdue, and such a model would allow geochemists, geomorphologists, tectonicists, geophysicists, and other geoscientists to test several hypothesis linking paleoclimates and tectonics with landscape evolution. The model advanced by Fenske et al. proposes to do so, but it comes short of its objectives.

The model fails in four aspects:

• the model is based on sound depiction of physical processes, but it does not have any chemical component;

• it is based on a misunderstanding of the mechanisms underlying the formation of iron duricrusts;

• literature review on the topic ignores relevant information demonstrating that iron duricrusts are indeed long-lived and slowly eroding components of cratonal landscapes;

• when the model is applied it produces results that are not substantiated by other models or by observations and measurements of physical reality.

An important shortcoming of the work is that it is based on a misunderstanding of the models proposed for the formation of iron duricrusts, which the authors summarize as:

"Two hypotheses have been proposed for the formation of duricrusts, i.e., the hydrological or horizontal model where the enrichment in the hardening element (iron for ferricretes) is the product of leaching and precipitation through the beating of the water table during contrasted seasonal cycles, and the laterisation or vertical model, where the formation of iron duricrusts is the final stage of laterisation."

There is indeed consensus that some duricrusts form by lateral introduction of iron into lower parts of the landscape and others form by in situ concentration of iron through time (laterization model); these are the "transported and in situ ferricretes" of Bourman (1996)

Bourman, R.P., 1996. Towards distinguishing transported and in situ ferricretes: data from southern Australia. AGSO Jour. Aust. Geol. Geophys. 16 (3), 231–241.

and discussed previously by several authors (e.g., Maignien, 1959, 1966; Ollier, 1969; McFarlane, 1976).

The ferricrete-producing process addressed by the numerical model in this work does not represent either of the accepted genetic models discussed by Bourman (1996) or Bourman et al. (2020). The

lateral model proposes that transported ferricretes form by the lateral introduction of iron-bearing minerals, rocks and solutions into lower segments of the landscape, including channels and river beds, where ferruginization occurs. In this model, there is an unconformity between the ferricrete and underlying lithologies or weathering profiles, as described in Bourman et al. (2020). As the landscape evolves, the ferricrete is more resilient than surrounding rocks, it resists erosion, and relief inversion occurs. The channel iron deposits of Western Australia would represent an extreme example of the transported ferricrete model.

The lateritic model proposes that iron is concentrated in situ, also by physical and chemical processes, through time; the duricrust evolves through the entire history of weathering and it is not "the final stage of laterisation.", as interpreted by the authors. Recurrent physical and chemical transport of iron throughout the entire history of laterization is documented by mineralogical and geochronological work (e.g., Monteiro et al., 2014, 2018a,b).

Water table ferruginization, the issue modelled by Fenske et al., does not represent either of the models above. It does occur in special circumstances, and it is particularly common at the margins of major rivers in the Amazon, along the Atlantic coast of Brazil (Monteiro et al., 2020), in Western Australia (Mann, 1983), and other localities in Africa, India, New Caledonia and elsewhere. Iron oxyhydroxide cementation occurs in places where reducing groundwaters rich in $Fe^{+2}$ ascend towards the surface or interact with $O_2$-rich meteoric water within permeable units, and iron hydrolyses and precipitates. Such ferruginized horizons occur in areas dominated by Dunne overland flow in the vicinity of BIF plateaus at Urucum, Brazil (Vasconcelos et al., 2018). But these ferruginized horizons are generally thin, of limited spatial distribution, and are not the direct precursors of the ferricretes that control regional landscape evolution. Thus, the water table "ferruginization" problem addressed by the numerical model of Fenske et al. is only marginally related to ferricretes and it does not represent either of the main ferricrete-formation models.

In addition, the physical parameters modelled by Fenske et al. are space, time, elevation (topographic height), surface slope, height of water table, precipitation rate, fluid flow velocity, hydraulic conductivity, regolith thickness, and diffusivity. The "chemical parameters" in the model are a dimensionless quantity that represents hardening or increase in "relative resistance to surface erosion", similar to that introduced by Sacek et al. (2018), and another arbitrary parameter that represents regolith hardening time. There is no chemistry in either parameter. The authors interpret these arbitrary hardening parameters as representing the formation of a ferricrete during "chemical weathering", but as it currently stands, it could be formation of silcrete, calcrete, or any other cement that decreases regolith erodibility. The lack of any chemical parameter relevant to the dissolution and reprecipitation of iron in the near surface environment makes it improper to use these hardening parameters as a model for the formation of ferricretes.

The authors also propose to "present the first numerical model for the formation of iron duricrusts based on the hydrological hypothesis". Numerical models dealing with fluid flow and chemical reactions at geological time scales, where iron dissolution and reprecipitation takes place, have been presented by Lichtner (1988), Lichtner and Waber (1992), Lichtner and Biino (1992) and many contributions since then. Lichtner and Waber (1992) consider the competing effects and the timescales of weathering and erosion, but do not directly model erosion. Thus, numerical approaches that deal with iron dissolution and reprecipitation based on robust thermodynamic, kinetic, and fluid flow models have existed for several decades.

Despite the fact that the basic model proposed by Fenske et al. does not address the accepted end-member models of ferricrete genesis and it does not contain any chemical component, the authors take the model results at face-value, without considering that the model itself may not be a correct depiction of nature. When faced with the fact that the model does not produce a result consistent

with other observations and measurements, the authors could revisit the model to assess its applicability. Instead, the authors prefer to conclude that all observations and measurements are wrong. Publication of the manuscript as is will not advance our understanding of the role of weathering in landscape evolution, and it may mislead the uninformed reader into believing that we finally have a modelling approach that couples chemical weathering and landscape evolution, which is not the case.

I will provide below a few specific comments and suggestions, line-by-line or section-by-section, that aim at improving the current version of the manuscript:

• Line 7 and throughout the manuscript.
The authors use the term "water table fluctuations" in their title, a perfect term for describing the vertical movement of the water table through time. They also use water table fluctuation to label the y-axis of their graph in Figure 4. However, in Line 7 of the abstract and throughout the manuscript they replace the perfectly understood and commonly used "water table fluctuation" by the "beating of the water table". The new term is unusual, confusing, totally unnecessary in face of the already available and perfectly understood "water table fluctuation". Beating of the water table and water table beating throughout the text should be replaced by "water table fluctuation" to make the text clearer and consistent with Figure 4.

• Lines 18-21.
The statement "Finally we demonstrate that the commonly accepted view that, because they are commonly found at the top of hills, duricrusts protect elements of the landscape is most likely an over-interpretation and that caution must be taken before using duricrusts as markers of uplift and/or base level falls." ignores the fact that the protection against erosion offered by duricrusts is not an "accepted view" but the result of numerous measurements and field relationships that cannot be simply "modelled away". If the model does not support the measurements and observations of physical reality, the model should be reconsidered, not alternate realities proposed.

• Lines 23-24.
"Understanding Earth's surface evolution in cratonic areas remains difficult in parts due to its slow and therefore difficultly measurable rates but also due to the important contribution from chemical weathering and the formation of the regolith."

• Lines 32-33.
"It describes an indurated elemental layer usually found capping hills or surfaces, that appears to protect them from erosion (Taylor and Eggleton, 2001)."

Duricrusts are not indurated by elements, they are indurated by minerals.

• Lines 36-37.
"Duricrust formation is likely to depend on water availability, often linked to climatic conditions and, for certain types of duricrusts, on the minerals present in the regolith and/or the underlying protolith."

Suggestion:

"Duricrust formation depends on water availability, often linked to climatic conditions, and on the minerals present in the regolith and/or the underlying protolith."

Duricrusts form when elements dissolve and re-precipitate, cementing parts of the regolith. Element dissolution-reprecipitation does not occur in the absence of water or the elements forming the cements.

• Lines 37-40.

"To cite some examples, gypsite crusts form in hyper-arid areas (Watson, 1988), silcretes and calcretes form in arid environments (Nash et al., 1994), whereas iron duricrusts form in areas where more water is available during a certain period of the year (Tardy, 1993), and bauxitisation happens under tropical conditions (Retallack, 2001)."

The conditions under which duricrusts form are largely undetermined and the statements above are simply working hypotheses. For example, under similar climatic conditions a gypsum-cemented (gypsite) or a calcite-cemented (calcrete) duricrust may form, depending on the relative amounts of $HCO_3^-$, $SO_4^=$, and $Ca^{++}$ dissolved in the groundwater.

• Lines 40-42.
"Although no direct measurement of their rate of formation is yet available, one can estimate that the time needed to create a duricrust is of the order of $10^5$ or more years (Tardy, 1993; Taylor and Eggleton, 2001)."

In a very simplistic view, determining rate of formation of a weathered zone or duricrust would require measuring when the weathering front arrived at a given position within the weathering profile, as illustrated below:

[Figure]

Vasconcelos, 1999

For simple weathering profiles, determining the rate of formation has been successfully carried out, as for a manganocrete in Minas Gerais, Brazil,

[Figure]

[Figure]

Fig. 4. (a) Histogram representing the modal distribution of the ages, and (b) a probability density plot illustrating the same results. (c) Plateau and pseudo plateau ages vs. depth for the 45 cryptomelane grains analyzed. The best fit line in the age vs. depth plot was obtained by using only the greatest plateau age at each depth, which records the minimum age for the arrival of oxidizing weathering solutions at each horizon. The slope of the best fit line is interpreted as the rate of propagation of the oxidation front (values for the linear fit were obtained by using the software *Ajuste 1.1* available at http://omnis.if.ufrj.br/~carlos/applets/reta/reta.html). Extrapolation of this curve to zero age intercepts the *y*-axis at ~100 m, the approximate depth of the weathering profile revealed by drill-core information.

that suggests that the weathering front advanced at a rate of 8.9 ± 1 m.Ma$^{-1}$ (Carmo and Vasconcelos, 2006)

The problem encountered when trying to apply the same approach to other duricrusts, particularly ferruginous duricrusts, is that recurrent mineral precipitation-dissolution-reprecipitation is the norm, and a simple age versus depth relationship in weathering profiles does not exist in most cases. The best that can be done with geochronology is to date as many samples as possible from various depths within a single weathering profile, such as in the example for a site in the tropical Amazon region illustrated below

[Figure]

Fig. 2. Deep weathering profiles underlay the Carajás Plateau. At Igarapé Bahia (IB), (a) the 150 m thick weathering profile is covered by a 0–15 m thick soil layer. (b) A duricrust delimits the contact (dashed line) between goethite-gibbsite-quartz-rich soils and sediments and the underlying mottled zone and gossan. (c) Major element analyses of drill core samples from the IB profile reveal that Fe is preferentially enriched within the duricrust and gossans. (U-Th)/He ages obtained for goethites selected from 5 drill cores are presented in Fig. 2c. Ages obtained for core F124 are in black, while ages of goethites from adjacent drill holes (not in the plane of this cross section) are in red. Goethites from the duricrust and saprolite are commonly older than 30 Ma, while younger goethite generations (He ages < 5 Ma) were only found at great depths. The results imply that the weathering profile had already reached its currently depths at ~60 Ma.

and use the age vs depth relationship through the oldest mineral at each depth to derive a weathering front migration rate (8.5 m.Ma$^{-1}$ in the example below):

[Figure]

Fig. 10. An age vs depth diagram for all goethites from the IB profile reveals that, on average, goethites are older near the surface, while younger generations are more frequent at depth. The results also show that the weathering profile had already reached depths of ~100 m by ~58 Ma. The calculated rate for weathering front propagation is similar to those obtained for Miocene weathering profiles in SE Brazil (Carmo and Vasconcelos, 2006), and twice as fast as similar rates obtained for weathering profiles in Australia (Vasconcelos and Conroy, 2003; Heim et al., 2006).

For Australian conditions, Heim et al. (2006) show that channel aggradation probably occurred at ~ 36 Ma, and that for the past 15 Ma the water table has been dropping and iron cementation has also progressed downward at the same rate as groundwater drawdown. Based on the (U-Th)/He results for goethite cements, water table drawdown progressed at a rate of 1.2 to 1.5 m.Ma$^{-1}$.

[Figure]

Figure 2. Goethite elevation versus (U-Th)/He ages for (A) uncorrected ages, and ages corrected for (B) 5%, (C) 10%, and (D) 20% $^4$He losses, showing that the results are statistically reproducible (2σ) when corrected for 10 and 20% $^4$He losses. Linear regression curves (solid lines) of all samples for each plot show progressively younger results with increasing profile depth (R$^2$ ≈ 0.61). Correlation coefficients are much greater (R$^2$ = 0.93) if only lower CID samples are plotted (dashed lines), suggesting the presence of two age populations (see text). Extrapolating the younger results, corrected for 20% $^4$He losses, to the original channel surface (~505 m) shows an intercept at 36 Ma, consistent with palynological ages for the initiation of channel aggradation (MacPhail and Stone, 2004). Upper CID samples are offset from this trend, suggesting partial dissolution and reprecipitation of goethite cement near the surface, possibly associated with excursion to more humid climates in the Miocene (McGowran et al., 1997).

In addition to the examples above, it is also possible to estimate rates of formation of duricrusts by dating ferruginous horizons at different states of evolution. For example, Riffel et al. (2016) shows that incipient duricrusts form at the 1-2 Ma timescales.

[Figure]

Fig. 8. (U–Th)/He ages for the goethite collected *in situ* (a) and colluvium (b, c) from Guarapuava.

[Figure]

Monteiro et al. (2018) shows that shallow iron duricrusts in the Carajás Region, Brazil, take between 1 and 10 Ma to form:

| Site | Sample name | Coordinates | Elevation (m) | Calculated age (Ma) | ±1σ (Ma) | Age +10% (Ma) | ±10% (Ma) |
|---|---|---|---|---|---|---|---|
| Itacaiunas | BOI−002 | S 06°19′47.6″ W 49° | 217 | 3.4 | 0.1 | 3.7 | 0.4 |
| Surface | BOI−002 | 47′40.7″ | | 2.4 | 0.1 | 2.7 | 0.3 |
| | BOI−002 | | | 3.9 | 0.1 | 4.3 | 0.4 |
| | BOI−002 | | | 1.8 | 0.0 | 2.0 | 0.2 |
| | BOI−002 | | | 2.4 | 0.1 | 2.6 | 0.3 |
| | BOI−004 | S 05°05′46.9″ W 49° | 285 | 0.5 | 0.0 | 0.6 | 0.1 |
| | BOI−004 | 39′13.9″ | | 1.2 | 0.0 | 1.3 | 0.1 |
| | BOI-004 | | | 0.7 | 0.0 | 0.8 | 0.1 |
| | BOI-004 | | | 0.5 | 0.0 | 0.5 | 0.1 |
| | BOI-004 | | | 0.7 | 0.0 | 0.8 | 0.1 |
| | BOI-013 | S 05°27′25.6″ W 49° | 183 | 6.9 | 0.2 | 7.6 | 0.8 |
| | BOI-013 | 28′43.5″ | | 6.8 | 0.2 | 7.4 | 0.7 |
| | BOI-013 | | | 7.7 | 0.2 | 8.5 | 0.8 |
| | BOI-014 | S 05°26′29.2″ W 49° | 179 | 8.5 | 0.2 | 9.4 | 0.9 |
| | BOI-014 | 37′4.51″ | | 6.8 | 0.2 | 7.5 | 0.8 |
| | BOI-014 | | | 8.5 | 0.2 | 9.3 | 0.9 |
| | BOI-014 | | | 7.9 | 0.2 | 8.7 | 0.9 |

The various approaches available for determining the rates of formation of duricrusts illustrated above are not ideal, but they are the ones imposed by the complexities of physical reality. They also show that the statement "Although no direct measurement of their rate of formation is yet available,… " is incorrect. Much of the literature on the topic has been ignored.

In addition, the absence of any direct geochronological evidence that ferricretes younger then ~ 1 Ma exist also shows that the statement "one can estimate that the time needed to create a duricrust is of the order of $10^5$ or more years (Tardy, 1993; Taylor and Eggleton, 2001)." is probably incorrect because it most likely underestimates the time needed for the formation of ferricretes in nature. On the other hand, young duricrusts (< 1-2 Ma) only have a minor role in slowing erosion. They form small steps on the landscape, and they are more easily destroyed by scarp retreat than older, thicker, better cemented duricrusts.

• Lines 47-49. "We will concentrate our study on ferricretes, also called ferruginous duricrusts, iron duricrusts or iron crusts, iron enriched levels or cangas (Tardy, 1993; Tardy and Roquin, 1998; Nahon, 1991; Paton and Williams, 1972; Ollier and Galloway, 1990; Vasconcelos et al., 1992; Monteiro et al., 2014; Vasconcelos and Carmo, 2018)."

Vasconcelos et al. (1992) do not address ferricretes, ferruginous duricrusts, iron duricrusts or iron crusts, iron enriched levels or cangas. However, classical work on the topic by Dorr (1964, 1969), Maignien (1966), McFarlane (1976), Ollier (1966), Tardy (1997), Samama (1986), or those using

modern tools by Vasconcelos et al. (1994), Vasconcelos (1999a,b), Shuster et al. (2005), Shuster et al. (2012), or Monteiro et al. (2018a,b,c) were mostly ignored.

• Lines 49-50. "Ferricretes are indurated layers made mostly of iron, with possible traces of other elements, e.g., titanium or manganese."

The most common elements other than Fe in ferricretes are Al and Si. Ti is common in ferricretes formed over basalts or gabbros; Cr and Al are common in ferricretes formed over ultramafic rocks.

• Lines 57-59. A climate conducive to the formation of iron duricrusts encompasses the following characteristics (Tardy et al., 1991; Tardy, 1993): 1) a mean annual rainfall, P, of around 1450 m.yr$_{-1}$, 2) a mean annual temperature, T, of ~28°C, a mean relative air humidity of around 70~%, and 4) a long dry period of at least 6 months.

The numbers above are extracted from Tardy (1997) (the English edition of Tardy 1993), fifth paragraph, page 359. Tardy was careful to give ranges for the values above, not absolute figures. Unfortunately, these estimates are derived from current climate data for sites where iron duricrusts presently occur. Evidence that lateritic weathering profiles form during prolonged periods during which climates change suggest that those numbers are simple estimates not based on any direct measurement. They are Tardy's estimations from mapping the global distribution of lateritic profiles and they should be regarded as such.

• Lines 60-63. The preservation of iron rich levels and duricrusts through time depends on climate too. In semi-arid to arid areas, they are preserved and protect the regolith for longer periods of time than in subtropical to tropical areas as described by Taylor and Eggleton (2001); Tardy (1993).

Some of the oldest preserved lateritic profiles on Earth occur in the Amazon (Carajás) and Minas Gerais (Quadrilátero Ferrífero), both tropical areas. Their ages, distribution, and the regional climatic conditions are thoroughly documented in the literature. Thus, the statement above is not based on any direct measurements of the distribution, ages and longevity of ferruginous duricrusts across the planet.

• Lines 66-69. There are currently two hypotheses for the formation of iron duricrusts: a hydrologically-based process (Taylor and Eggleton, 2001; Achyuthan, 2004; Widdowson, 2009; Bonsor et al., 2014; Riffel et al., 2016; Bourman et al., 2020) and a laterization based process (Tardy, 1986; Tardy et al., 1988; Nash et al., 1994; Tardy, 1993; Theveniaut and Freyssinet, 1999; Taylor and Eggleton, 2001), which are also referred to as horizontal and vertical models.

The authors should consult the literature again to make sure they understand the two models correctly. Their summaries in lines 70-113 could be greatly improved with a better understanding of the two models.

• Lines 71-73. In this model, iron duricrusts form under a contrasting yearly climate, made of primarily wet and dry periods. During wet periods, the water table height is high and minerals, such as Fe$^{+2}$, are transported from adjacent regions and accumulate. During dry periods, the water table height drops and minerals, such as Fe$^{+3}$, precipitate.

Availability of aqueous (Fe$^{+2}_{aq}$) iron species is indeed an important aspect in the formation of iron duricrusts, in conjunction with iron oxidation (Fe$^{+2}_{aq} \Rightarrow$ Fe$^{+3}$) into the trivalent species, which readily hydrolyses and precipitates as goethite or hematite. But Fe$^{+2}$ and Fe$^{+3}$ are chemical species in solution, they are not minerals. Goethite and hematite are the relevant minerals in the case of most duricrusts. Understanding the redox behavior of iron in the near surface environment is key in formulating a credible model for the formation of duricrusts. The redox behavior of iron in the presence of oxygenated rainwater is rather complex. In general, iron solubility requires ligands, such as organic acids, to provide the acid-reducing conditions necessary to stabilize the reduced (Fe$^{+2}_{aq}$) iron species in solution. Increase in oxidation potential and alkalinity promptly promotes iron reprecipitation. Most oxygenated water tables in the near surface environment are poor in

soluble iron because Eh-pH conditions are often within the thermodynamic stability field of goethite-hematite. Simply equating weathering with water table movement and defining that something precipitates when the water table moves up or down is too simplistic an approach, without any chemical basis, to properly model iron mobility and Fe-duricrust formation.

• Lines 73-76. This cycle repeats itself for thousands of years, with the accumulation of iron elements leading to the formation of nodules, which, ultimately, cement into a ferruginous crust. In this case, no genetic link between the bedrock and the regolith beneath is needed nor described (Ollier and Galloway, 1990; Taylor and Eggleton, 2001).

The process repeats itself for millions of years, but the only "element" that accumulates is iron (there are no other "iron elements", there are "iron isotopes"), in the form of either goethite, hematite, or lepidocrocite, depending on water availability, pH, rates of oxidation and precipitation, etc. A genetic link is indeed missing when the duricrust forms by iron cementation of detrital phases deposited unconformably on unrelated bedrock paving valley floors. But the material that is deposited in valleys and eventually becomes iron cemented is generally iron-rich, i.e., the iron is not entirely introduced by the groundwater; it is often locally remobilized.

• Lines 100-103. There exists, at this stage, no numerical model for the formation of duricrusts, apart from the conceptual model developed by Nash et al. (1994) and the highly simplified model developed by Sacek et al. (2019) to estimate the effects of duricrust formation on erosional patterns at the continental scale.

The "model" of Nash et al. (1994) for the formation of calcretes and silcretes in Africa does not provide a suitable analogue for the formation of iron duricrusts. Calcretes and silcretes depend on the amount of $Ca^{++}$ and $H_4SiO_4$ in solution, and precipitation of $CaCO_3$ (in the form of calcite) and $SiO_2$ (in the form of opal, chalcedony, or quartz) is readily achieved by supersaturation when water ascends to the surface and evaporates. Iron solubility is much more complex, it depends on pH and redox conditions, it depends on the availability of ligands, iron reprecipitation may be partially catalyzed by microorganisms, etc. Iron is most often locally sourced, mobilized, and reprecipitated. Long-range transport of iron requires special circumstances, such as those documented by Mann (1983) in Western Australia.

In addition, there are numerous numerical models for the formation of Fe-bearing leached caps (Lichtner and Biino, 1992) and of bauxites (Soler and Lasaga, 1996) that are based on fundamental thermodynamics, kinetics, and fluid flow modelling. It is true that there are no models that combine fluid flow, geochemical thermodynamics/kinetics and the physical processes – e.g., hillslope diffusion, channel transport, landslide/rock fall – controlling landscape evolution, i.e., there are no models coupling weathering to erosion. Such models are still missing.

• Lines 76-79. Also, duricrusts form at the water table, which means at multiple metres below the surface. It is generally accepted that, to permit accumulation of materials, the region needs to be tectonically inactive and that later periods of uplift (or base-level drop) are likely to exhume the duricrust to the surface where it becomes more resistant to erosion than the surrounding weathered material.

Most authors suggest that iron duricrusts form at the surface, not subsurface. For example, the duricrusts formed from Dunne overland flow at the lowlands at Urucum (Vasconcelos et al., 2018) form at the surface. If there are references that support duricrust formation by an ascending or descending water table (e.g., Mann, 1983), they should be provided. Importantly, duricrusts that form when the water table ascends to the surface, water evaporates, and minerals precipitate are more relevant to calcretes or silcretes than ferricretes.

The model presented by the authors is a modification of a regolith formation model by Braun et al. (2016):

• Lines 114-180. 2 Method and Results

 ## 2.1 Existing regolith formation model (Braun et al., 2016)

Here we will use the model for regolith formation developed by Braun et al. (2016) that computes the rate of downward migration of a weathering front in proportion to the velocity of the water in the overlying permeable regolith.

[Figure]

Braun et al. (2016)'s model is made of three components: a surface process model, a hydrological model and a weathering model.

The regolith profile produced by Braun et al. (2016)'s model shows a relationship between elevation and depth of weathering that is the opposite of that produced by alternative regolith-forming models, such as the model of Rempe and Dietrich (2014), where deeper regolith occurs at the highest elevations and the regolith shallows towards lower landscape positions

**A bottom-up control on fresh-bedrock topography under landscapes**

Daniella M. Rempe[1] and William E. Dietrich[1]

[Figure]

**Fig. 1.** Conceptual model showing the elevation of fresh bedrock, $Z_b$, under ridge and valley topography with a thin soil mantle overlying a weathered bedrock zone that extends to $Z_b$. Channel incision, at the rate $C_o$, drives hillslope erosion and drainage of fresh bedrock (flow paths illustrated with blue arrows). (*Left*) The model framework and assumptions. At the ridgetop ($x = 0$), the surface elevation is $Z_{s0}$ and the fresh-bedrock elevation is $Z_{b0}$. Groundwater flux, $q_w$, is horizontal and proportional to the water table gradient, $\nabla Z_b$. Soil transport, $q_s$, is proportional to the surface slope, $\nabla Z_s$. All soil and water leaves the hillslope at $L$ where the hillslope meets the channel. At steady state, the rate of channel incision ($C_o$) is equal to the uplift rate such that the ground surface, $Z_s$ and surface of the fresh bedrock, $Z_b$, are stationary.

or the model by Lebedeva and Brantley (2013), that shows a similar relationship between elevation and depth of weathering:

Exploring geochemical controls on weathering and
erosion of convex hillslopes: beyond the empirical
regolith production function

Marina I. Lebedeva* and Susan L. Brantley
Earth and Environmental Systems Institute, The Pennsylvania State University, University Park, PA, USA

[Figure]

**Figure 6.** The steady-state solution in the moving system of coordinates for the hill weathering in the weathering-limited regime. Parameters as in Figures 3 and 4 with the exception of $E = 4 \times 10^{-5}$ m/yr, $v = 0 \cdot 6$ m/yr, and $k_{ab} = 1 \cdot 5 \times 10^{-12}$ mol/m$^2$ s. Parameters were varied from previous figures in order to exemplify weathering limitation. The steady-state profile is the curve $\bar{H}(\bar{x}, \bar{t}) = -1.6\bar{x}^2 + 2 - 0.2\bar{t}$. This figure is available in colour online at wileyonlinelibrary.com/journal/espl

The topography vs depth of regolith produced by Braun et al. (2016)'s model also contradicts observations of weathering profiles throughout Australia, Brazil, Africa, and China, where regolith is deeper at the higher elevation sites and becomes shallower as elevation decreases. For example, in field mapping in deeply weathered terrains we teach our students to go into valleys to look for subcrops of unweathered lithologies. Thus, the starting regolith model illustrated in Figure 1 is a problematic depiction of physical reality and it raises some concerns when further applied to explain landscape evolution. If the thickness of the regolith is easily controlled by the model, as stated in Lines 158-159

Lines 158-159. On the other hand, $\Gamma$ controls whether the regolith is thickest at the top of the hill, i.e., when $\Gamma > \Omega_2$ $\Omega_{-1}$, or thickest at the base of the hill, i.e., when $\Gamma < (\Omega_2 / \Omega_{-1})$.

why not start with a regolith model that is more faithful to physical reality and to the results produced by other models?

As previously mentioned, the modified model includes physical parameters, such as space, time, elevation (topographic height), surface slope, height of water table, precipitation rate, fluid flow velocity, hydraulic conductivity, regolith thickness, and diffusivity, but the only "chemical parameters" are a dimensionless quantity that represents hardening or increase in "relative resistance to surface erosion" and another arbitrary parameter which represents regolith hardening time. These "chemical parameter" are not directly related to the dissolution and reprecipitation of iron in the near surface environment. Indeed, if the colour in Figure 5 were changed from red-brown

to grey-white, the duricrust illustrated could be a silcrete or calcrete. Thus, to equate the duricrust produced by the model with a ferricrete is an unsubstantiated extrapolation of what is essentially a physical model.

Duricrust formation in the new model is directly proportion to the arbitrary hardening parameter (k), it is proportional to precipitation rate, to the range and rate of water table fluctuation, and regolith hardening time. The direct dependence of hardening to precipitation (rainfall) rates in the case of ferricrete formation is counterintuitive. Greater precipitation rates result in greater supply of oxygenated water, which lowers iron solubility. As iron is mobile in its reduced form ($Fe^{+2}_{aq}$) in most surficial environments, rainwater by itself will not contribute to iron transport. In surficial environments, organic acids must be added to infiltrating rain to drive iron dissolution and transport. In many cases, iron becomes immobilized when rainfall is abundant, and it enters solution when rainfall decreases and water ponds in the subsurface, becoming reducing.

To constraint model parameters – "2.3 Constraining new model parameters" –, the authors search through the literature for physical studies that may help determining the length of time necessary for the formation of an iron duricrust. In this section, studies based on field observations and educated opinions are evaluated equally and interchangeably with studies based on actual measurement of time of duricrust formation. It would be useful to differentiate the literature into work that uses field observation to infer rates of processes to those that measure or at least attempt to measure rates of processes. And, as outlined above, many of those later studies were largely ignored by the authors.

The authors also evaluate the measured ranges of water table fluctuations across the planet, and show that constraining this parameter is easier.

The model run itself is interesting, even if it is only remotely associated with the formation of ferricretes. Even if the geometry of the regolith is not that commonly measured in the field and the subsurface position of induration is not what is commonly observed in actual ferricretes, the timing of formation of a duricrust and the role of duricrust in slowing erosion is a step towards better understanding these systems. However, as mentioned above, that is only relevant to "duricrust" formation, it has no specificity as which type of duricrusts. Thus, affirming that the study poses some constraint on the formation of ferricretes is an unwarranted extrapolation of the results. And it is only relevant to the formation of duricrusts by groundwaters, which is not the dominant process controlling the formation of ferricretes.

Also a concern is the fact that the model employed by the authors produces weathering profiles that are different from weathering profiles observed in nature and weathering profiles produced by other models. Thus, when the model also shows an ephemeral role of duricrusts in protecting landscapes against erosion, contrary to observations and measurements, its relevance should be taken with some caution. When facing such results, the authors could ask the question: "What is possibly wrong with the model?". However, the authors prefer to take the model results at face-value and question what nature tells us. They dismiss previous observations

"Many authors have previously described duricrusts in lateritic or regolith profiles (e.g. Tardy (1993); Taylor and Eggleton (2001)) as a protecting layer that should slow down erosion of the underlying topography/hill. We do not observe this in our model, and now proceed to explain it."

but also ignore a now large number of measurements, based on geochronology and cosmogenic isotope studies, that show that the duricrusts that sit on friable and easily erodible saprolites are indeed ancient (Vasconcelos et al. 1994; Vasconcelos, 1999a,b; Hénocque et al., 1998; Shuster et al., 2005; Beauvais et al., 2008; Monteiro et al., 2014; Vasconcelos and Carmo, 2018) and eroded very slowly (Fujioka et al.., 2010; Shuster et al., 2012; Monteiro et al., 2018a,b). Either all these measurements are wrong, and the duricrusts are not as old as determined by both $^{40}Ar/^{39}Ar$ and (U-

Th)/He geochronology, and the very low rates of long-term erosion measured by both $^3$He and $^{53}$Mn (and now $^{21}$Ne) are also incorrect, or the duricrust formation model proposed by Fenske et al. is insufficiently robust to properly model the types of landscapes that it attempts to investigate.

Some authors have also demonstrated the mechanisms ("self-healing" of Monteiro et al., 2014) that allow iron duricrusts to continuously regenerate themselves, resist erosion, and protect the material below. Some of these processes have been experimentally reproduced in the laboratory, and given the right experimental conditions, greatly accelerated (Levett et al., 2019; 2020a,b,c). Thus, the longevity, resilience, and mechanisms of formation of ferruginous duricrusts are much better understood than what is portrayed in the current manuscript. Therefore, if the model cannot reproduce what is measured in nature, the model results should be more conservatively interpreted to make the manuscript into the useful and valuable contribution that it could be.

In summary, the authors should treat their duricrust formation process as an emerging approach, accept that their duricrust is a generic indurated material and not a ferricrete, compare their model results to natural settings to see where the model could be improved, and adjust model parameters to see if they can reproduce the measurements that show that duricrusts are actually ancient and erode very slowly.

The authors should also consider introducing actual weathering geochemistry into their model by coupling robust thermodynamic and kinetic approaches available in established geochemical models with their landscape evolution approaches. Only then it will be possible to model the formation and preservation of ferricretes, silcretes, calcretes, bauxites, and other types of duricrusts and the supergene systems that they cover.

In summary, the model in this manuscript can be a valuable contribution if presented for what it is and more reservedly interpreted. As currently presented, it will mislead readers into believing that we currently have a model that couples chemical weathering and landscape evolution, when this is not the case. To be the valuable contribution that it could be, the manuscript should be substantially revised and re-written. I have not checked the references, supplementary materials, etc. I will be willing to do so if a revised version of the manuscript is produced.

References:

Anand R.R. and de Broeker P. (2005). Regolith Landscape Evolution Across Australia. CRC LEME, 354 pages.

Beauvais, A., Ruffet, G., Hénocque, O., Colin, F. (2008). Chemical and physical erosion rhythms of the West African Cenozoic morphogenesis: the 39Ar-40Ar dating of supergeneK-Mn oxides. J. Geophys. Res. 113 (F4), F04007. http://dx.doi.org/10.1029/ 2008jf000996.

Carmo, I.O., Vasconcelos, P.M., 2006. $^{40}$Ar/$^{39}$Ar geochronology constraints on Late Miocene weathering rates in Minas Gerais, Brazil. Earth Planet. Sci. Lett. 241 (1–2), 80–94. http://dx.doi.org/10.1016/j.epsl.2005.09.056.

Dorr J. V. N. (1964) Supergene iron ores of Minas Gerais, Brazil. Econ. Geol. 59(7), 1203–1240.

Dorr J. V. N. (1969) Physiographic, stratigraphic and structural development of the Quadrilatero Ferrifero, Minas Gerais, Brazil. U.S.G.S. Prof. Paper, 614-A, Washington, DC, 110 pp.

Fujioka, T., Fifield, L. K., Stone, J. O., Vasconcelos, P. M. P., Tims, S. G., & Chappell, J. (2010). In situ cosmogenic $^{53}$Mn production rate from ancient low-denudation surface in tropic Brazil. Nuclear Instruments and Methods in Physics Research B, 268, 1209–1213.

Heim, J. A., Vasconcelos, P. M. P., Farley, K. A., Shuster, D. L., & Broadbent, G. C. (2006). Dating paleochannel iron ore by (U-Th)/He analysis of supergene goethite, Hamersley Province, Australia. Geology, 34, 173–176.

Hénocque, O., Ruffet, G., Colin, F., & Féraud, G. (1998). 40Ar/39Ar dating of West Africa lateritic cryptomelanes. Geochimica et Cosmochimica Acta, 62(16), 2739–2756.

Levett, A., Gagen, E. J., Rintoul, L., Guagliardo, P., Diao, H., Vasconcelos, P. M., & Southam, G. (2020). Characterisation of iron oxide encrusted microbial fossils. Scientific Reports, 10(1), 1–11. https://doi.org/10.1038/s41598-020-66830-z.

Levett, A., Gagen, E. J., Vasconcelos, P. M., Zhao, Y., Paz, A., & Southam, G. (2020). Biogeochemical cycling of iron: Implications for biocementation and slope stabilisation. Science of the Total Environment, 707, 136128. https://doi.org/10.1016/j.scitotenv.2019. 136128.

Levett, A., Gagen, E. J., Zhao, Y., Vasconcelos, P. M., & Southam, G. (2020). Biocement stabilization of an experimental-scale artificial slope and the reformation of iron-rich crusts. Proceedings of the National Academy of Sciences of the United States of America, 117(31), 18347–18354. https://doi.org/10.1073/pnas.2001740117.

Levett, A., Vasconcelos, P. M., Gagen, E. J., Rintoul, L., Spier, C., Guagliardo, P., & Southam, G. (2020). Microbial weathering signatures in lateritic ferruginous duricrusts. Earth and Planetary Science Letters, 538, 116209. https://doi.org/10.1016/j.epsl.2020. 116209.

Levett, A., Gagen, E. J., Paz, A., Vasconcelos, P. M. & Southam, G. (2022). Strategising the bioremediation of Brazilian iron ore mines, Critical Reviews in Environmental Science and Technology, 52:15, 2749-2771, DOI: 10.1080/10643389.2021.1896346

Lichtner P.C. (1988). The quasi-stationary state approximation to coupled mass transport and fluid–rock interaction in a porous medium, Geochim. Cosmochim. Acta 52 143– 165.

Lichtner P.C. and Biino G.G. (1992). A first principles approach to supergene enrichment of a porphyry copper protore: I. Cu-Fe-S subsystem. Geochim. Cosmochim. Acta 56 pp. 3987-4013.

LICHTNER P. C. and WABER N. ( 1992) Redox front geochemistry and weathering: Theory with application to the Osamu Utsumi uranium mine, Pocos de Caldas, Brazil. J. Geochem. Exp. 45, 521-564.

Maignien R. (1966). Review of research on laterite. Natural Resources Series, 4, UNESCO, Paris.

Mann. A. (1983). Hydrogeochemistry and weathering in the Yilgarn Block, Western Australia – ferrolysis and heavy metals in continental brines. GCA 47, 181-190.

McFarlane, M. J. (1976). Laterite and landscape. London and New York: Academic Press.

Monteiro, H., Vasconcelos, P., & Farley, K. (2018). A combined (U-Th)/He and cosmogenic $^3$He record of landscape armoring by biogeochemical iron cycling. Journal of Geophysical Research: Earth Surface, 123(2), 298–323. https://doi.org/10.1002/ 2017JF004282

Monteiro, H. S., Vasconcelos, P. M., Farley, K. A., Spier, C. A., & Mello, C. L. (2014). (U–Th)/He geochronology of goethite and the origin and evolution of cangas. Geochimica et Cosmochimica Acta, 131, 267–289. https://doi.org/10.1016/j.gca.2014.01.036

Monteiro, H. S., Vasconcelos, P. M. P., Farley, K. A., & Lopes, C. A. M. (2018). Age and evolution of diachronous erosion surfaces in the Amazon: Combining (U-Th)/He and cosmogenic $^3$He records. Geochimica et Cosmochimica Acta, 229, 162–183. https://doi. org/10.1016/j.gca.2018.02.045

Ollier C. (1969). Weathering. K.M. Clyton, London, 270 pages.

Rempe D.M. and Dietrich W. (2014). A bottom-up control on fresh-bedrock topography under landscapes. PNAS 111, www.pnas.org/cgi/doi/10.1073/pnas.1404763111

Samama J.C., (1986). Ore Fields and Continental Weathering. Van Nostrand Reinhold Company, 326 pages.

Shuster D. L., Vasconcelos P. M. P., Heim J. A. and Farley K. A. (2005) Weathering geochronology by (U-Th)/He dating of goethite. Geochim. Cosmochim. Acta 69(3), 659–673.

Shuster, D. L., Farley, K. A., Vasconcelos, P. M., Balco, G., Monteiro, H. S., Waltenberg, K., & Stone, J. O. (2012). Cosmogenic 3He in hematite and goethite from Brazilian "canga" duricrust demonstrates the extreme stability of these surfaces. Earth and Planetary Science Letters, 329-330, 41–50. https://doi.org/10.1016/j.epsl.2012.02.017

Tardy Y. (1997). Petrology of Laterites and Tropical Soils. A.A.Balkema, Rotterdam, 408 pages.

Vasconcelos, P.M., 1999a. K-Ar and 40Ar/39Ar geochronology of weathering processes. Annu. Rev. Earth Planet. Sci. 27, 183–229. http://dx.doi.org/10.1146/annurev.earth.27.1.183.

Vasconcelos, P.M., 1999b. 40Ar/39Ar geochronology of supergene processes in ore deposits. Rev. Econ. Geol. 12, 73–113.

Vasconcelos, P.M., Renne, P.R., Brimhall, G.H., Becker, T.A., 1994b. Direct dating of weathering phenomena by 40Ar/39Ar and K-Ar analysis of supergene K-Mn oxides. Geochim. Cosmochim. Acta 58 (6), 1635–1665. http://dx.doi.org/10.1016/0016- 7037(94)90565-7.

---

## Author Response (AR1)

We thank the two reviewers for their comments and suggestions, which we incorporated, for the most, in the revised version of our manuscript. Many of the concerns expressed by Reviewer 2 relate to our choice to focus on ferricretes, which we had made because these are one the most studied types of duricrust. As our manuscript aims at describing a model that represents a formation mechanism (duricrust formation by water table fluctuations) rather than a particular composition, we decided to use the more general term "duricrust" to qualify the hardened layer that is predicted in the model, rather than "ferricrete".

We also restructured and expanded the introduction to help the reader better differentiate between the different hypotheses that have been proposed for the formation of duricrusts. This helps better understand the objectives of the current manuscript and that of a future manuscript that we are currently preparing on the formation of duricrusts by laterisation.

We have expanded the section providing observational constraint on the model parameters, and in particular the characteristic time scale for duricrust formation, tau. Following the reviewers' suggestions for its structure.

Finally, we expanded the section reporting the effect of duricrusts on surface erosion and their ability to "protect" landscape features. For this we added a few figures that better illustrate the model behaviour and by which process (lateral erosion rather than vertical erosion) duricrusts are likely to be eroded away, in agreement with measured rates of surface erosion by cosmogenic isotopes.

We have also incorporated other smaller changes that were suggested by the reviewers. Below is a response point-by-point of the reviewers' comments and suggestions. We strongly believe that through these numerous changes, the manuscript has improved in clarity and impact, and we thank again the reviewers for their contribution in this process.

**Review 1:** This paper showcases a newly developed model for ferricrete formation and uses the outputs of the model to comment on the role of ferricretes in landscape evolution. The paper is clearly written and illustrated, and as a result generally easy to follow. The conclusions are generally well-argued. There are, however, some points that need additional thought, and the terminology requires modification.

- The most important point concerns how iron is transported and precipitated; the authors have not correctly described this, and it affects the model concept.

For the transport of iron in groundwater with more or less neutral pH (true for the vast majority of groundwater), conditions must be reducing; the iron is present as the reduced species $Fe^{2+}$. For this to precipitate and form a ferricrete, the groundwater must be exposed to oxygen so it becomes oxidising; the $Fe^{2+}$ converts to $Fe^{3+}$ and automatically precipitates as poorly crystalline ferric hydroxide, which will crystallise as goethite or haematite over time. This is clearly explained in references like Drever (1997 The Geochemistry of Natural Waters: Surface and Groundwater Environments).

What this means is that ferricrete formation is not uniform; most ferricrete precipitates where the watertable intersects the surface and groundwater seeps occur, allowing maximum contact

between the Fe2+-containing groundwater and the oxygen in the atmosphere. Thus the situation shown in Fig. 5 is an oversimplification of the true situation.

Ferricrete precipitation extends to the right beneath the hill because the groundwater is exposed to oxidising conditions at the watertable. This will be enhanced if the watertable fluctuates substantially; as the watertable falls, the groundwater retained as a meniscus around soil particles will be exposed to the air that now fills the pore spaces and the Fe2+ in solution will be oxidised and precipitate ferric hydroxide. The greater the watertable fluctuation, the greater the ferricrete precipitation.

The authors need to explain this more clearly in the text. Note that Fe2+ cannot accumulate (line 73) if it is dissolved in groundwater.

Response: We agree with the reviewer that reducing conditions are favourable for the transport of Fe2+ and oxidising conditions for precipitation of Fe3+. However, according to several authors, e.g. Taylor et Eggleton 2001 or Tardy 1993, oxidising environments can be found near or just below the surface and not just where the water table meets the surface. To quote Taylor et Eggleton (2001), for example: "The upper part of this saturated zone, because it is moving and renewed, is generally aerobic and it is in this part of the zone that weathering is most effective. The soluble products of weathering are readily removed by its flow allowing weathering to proceed readily. Because the upper part is oxidizing, particularly near the water-table, Fe-oxyhydroxides mark the position of the water-table".

To address the reviewer's concern, we modified the paragraph entitled "Hydrological hypothesis or horizontal model" (lines 70 to 76 in original manuscript) in the following manner:

In this model, *duricrusts form at the water table height* under a contrasting yearly climate, made of primarily wet and dry periods. *This hypothesis is adapted for different duricrust, e.g. calcretes, silcretes or ferricretes (Ollier and Galloway, 1990; Wright et al., 1992; Webb and Golding, 1998; Taylor and Eggleton, 2001; Alonso-Zarza, 2003; Temgoua et al., 2005; Widdowson, 2007; Ullyott and Nash, 2016)*. During wet periods, the water table height is high and minerals, such as *dissolved iron or calcite*, are transported from adjacent regions *to a topographic low*. During dry periods, the water table height drops and *the transported minerals, precipitate under different redox, pH and environmental conditions. Precipitation is possible in undersaturated environments, where redox conditions are optimal, when reducing conditions become oxidising. In the upper parts of the saturated regolith, i.e. at the water table height, the environment is aerobic, which enables the change in redox conditions and enhances weathering (Taylor and Eggleton, 2001). As observed by Taylor and Eggleton (2001), these mineralisation patterns can mark the position of the water-table. While the upper part of the groundwater is constantly renewed, i.e. by seasonal precipitation, the lower part stays saturated and is possibly stagnant. This can lead to depletion in 02 in that regolith layer, and the deep part of the saturated zone becomes anaerobic and thus, reducing (Taylor and Eggleton, 2001).*
*The seasonal* cycle repeats itself for thousands of years, with the accumulation of *minerals* leading to the formation of nodules, which, ultimately, cement into a crust. In this case, no genetic link between the bedrock and the regolith beneath is needed nor described (Ollier and Galloway, 1990; Taylor and Eggleton, 2001). All elements are brought from adjacent sources through *transport*.

- An additional oversimplification in Fig. 5 is that the watertable is virtually never flat as shown; it is almost always a reflection of the topography with a gentler gradient. Thus, in Fig 5 it should slope gently to the left. This provides a hydraulic gradient that causes groundwater to flow to the left, helping to cause greater ferricrete precipitation where the watertable intersects the ground surface.

Response: The geometry of the water table in the model is computed by solving the equation governing the flow of water within an unconfined aquifer in a homogeneous, permeable layer assuming steady-state and following the Dupuy-Forchheimer assumption, as explained in Braun et al. (2016) and summarized in the model section of our manuscript. Consequently, the water table can take many shapes and geometries and be very close to the surface when the system is saturated, for example. As the regolith layer thickens the water table geometry is mostly controlled by the shape of the bedrock, the assumed hydraulic conductivity and the infiltration rate.

We made sure that the model description adequately addresses this point.

- The model needs to be modified to take into account points 1 and 2.

We agree with the reviewer on both points but decided not to modify the model and responded by improving its description.

- The two different hypotheses of ferricrete formation are more-or-less correctly differentiated, but need to be more carefully described. The difference is between iron that has been concentrated in situ as other elements have been removed, and iron that has been transported in groundwater and precipitated at some distance from its origin. The transport for the latter ferricretes is both lateral and vertical (the presence of vertical transport is evident from Fig 5; the ferricrete beneath the crest of the hill must have received iron transported vertically downwards). The first category of ferricretes is often called residual; the second category can be characterised as transported. Using these terms makes the distinction much clearer. Thus the word 'lateral' should be deleted in lines 76 and 413.

Response: We modified the names of the hypotheses as suggested by the reviewer and according to literature (Bourman 1985). Lines 76 and 413 were modified, while also adjusting the names for the hydrological model for consistency (lines 6, 69, 70, 111, 546 and 555).
We also modified paragraph "Hydrological hypothesis or horizontal model" (lines 70 to 82, in new manuscript, lines 104 to 133) as described above to describe it more profusely.

- The definition of laterite (lines 84-85) is incorrect because it excludes the abundance of iron oxy-hydroxides as a distinguishing feature. The term 'laterite' was originally applied to Fe-rich material in Kerala (India) by Buchanan (1807, A Journey from Madras through the countries of Mysore, Canara and Malabar. East India Company, London). Therefore it is also not correct to say that "All rock types can weather into laterites under the right conditions", because there has to be enough Fe in the rock originally to form a laterite.

Response: We corrected the definition of laterite line 84-85 to include its mineral characteristics "Laterites are a type of tropical soil, encompassing "residual materials formed directly by in situ rock breakdown", characteristically enriched in iron and aluminium (Widdowson, 2009)."

- Lines 4-5, 64-65, 486 – "In most cases today, ferricretes are observed capping and protecting hills, at the top of landscapes" – this is not true of Western Australia, where ferricretes are abundant and largely occur in valleys (e.g. Anand & M. Paine 2002 Australian Journal of Earth Sciences, 49, 3-162; Bourman et al 2020 Geomorphology 354 107017). And it is probably not true generally; this would help to confirm that ferricrete-capped hills commonly occur on only a small scale (line 487).

Response: The fact that duricrusts are mostly found capping hills is based on literature (e.g. Azmon et Kedar 1985, Taylor et Eggleton 2001, Widdowson 2009, Monteiro et al. 2014 and others) but we modified the text to include more occurrences of duricrusts (lines 30 to 35):

(lines 32 to 36 in new manuscript) *"It describes an indurated mineral layer, usually found capping hills or surfaces as seen in figure 1, that appears to protect them from erosion (Azmon and Kedar, 1985; Twidale and Bourne, 1998; Taylor and Eggleton, 2001). Duricrusts can also be found along valley bottoms in paleodrainage systems (Radtke and Brückner, 1991; Chudasama et al., 2018). When exhumed, the system's channel beds are preserved due to the low erodibility of duricrusts, while the neighbouring layers are eroded, which can lead to inverted topographies (Goudie, 1985; Twidale and Bourne, 1998; Taylor and Eggleton, 2001, 2017)."*

We also added new literature and the newly included *figure 1* "Examples of duricrusts from Namibia and Chile.", where this landscape feature is observed.

- The term 'beating' needs to be replaced by 'fluctuation'. Beating has a different meaning: pulsation or throbbing, especially of the heart.

Corrected. We modified lines 7, 11, 166, 168, 173, 286, 291, 320, 341, 387, 395, 414, 423, 435, 448 and 556.

- The word 'difficultly' is extremely rarely used. It is better replaced by 'difficult to'. So 'difficultly measurable rates' would become 'difficult-to-measure rates' and 'difficultly soluble elements' would become 'difficult-to-dissolve elements', although the latter would be better as 'slightly soluble elements'. Corrected on lines 23 and 87.
- Minor comments:
  - "silcretes and calcretes form in arid environments" (line 38). Silcretes can also form in humid environments (Rozefelds et al. 2024 Gondwana Research 130, 234–249; Webb and Golding 1998; Journal of Sedimentary Research A, 68, 981-993). Corrected. We added literature and modified the sentence line 38: *"Silcretes are hypothesised to form in arid but also humid silica-rich environments (Butt, 1985; Nash et al., 1994; Webb and Golding, 1998; Nash, 2011; Rozefelds et al., 2024)."*
  - Line 50 – 'iron' should be 'iron oxides / oxy-hydroxides'. Corrected.
  - Line 58 – should be 1450 mm.yr-1 ? Corrected.
  - Line 103 – new subheading needed. Corrected.

○ Line 482 – should be 'result'. Corrected.

**Review 2:** A numerical model for duricrust formation by water table fluctuations
By Caroline Fenske, Jean Braun, François Guillocheau, and Cécile Robin
Review by Paulo Vasconcelos, Feb 14, 2024

Fenske et al. propose to develop and apply a numerical model for iron duricrust formation by water table fluctuations. They interpret their model to be based on sound physical and chemical processes and apply the model to test whether the formation of ferricretes slows erosion. Their main conclusion is that ferricretes have a limited and transient role in slowing erosion and protecting the underlying weathering profiles.

A model that properly couples chemical weathering and landscape evolution is long overdue, and such a model would allow geochemists, geomorphologists, tectonicists, geophysicists, and other geoscientists to test several hypothesis linking paleoclimates and tectonics with landscape evolution.
The model advanced by Fenske et al. proposes to do so, but it comes short of its objectives. The model fails in four aspects:
- the model is based on sound depiction of physical processes, but it does not have any chemical component;

Response: From the reviewer's comment, it appears that our main objective, stated at the end of the introduction, i.e., "to present a simple, yet predictive numerical model to simulate the geometry and timing of duricrust formation on geological time scales, to predict their effect on surface processes and to compare them to observations", was not explicit enough. We expanded this point in the revised version of the manuscript (lines 162 to 180 in new manuscript) to avoid confusion on the objectives of our manuscript and the purpose of the numerical model.

*Modelling duricrust formation*
*Our main objective is to present here a simple, yet predictive numerical model to simulate the geometry and timing of duricrust formation on geological time scales, in order to predict their effect on surface processes and to compare them to observations. In other words, we propose here to develop a new parametric representation of the process of duricrust formation based on a reduced set of generic parameters that can be constrained by comparing the model predictions to observations, rather than using a representation that would require the calibration of parameters through direct experimentation or measurements.*

*Apart from the conceptual model developed by Nash et al. (1994) and the highly simplified model developed by Sacek et al. (2019) to estimate the effects of duricrust formation on erosional patterns at the continental scale, there exists, at this stage, no numerical model predicting the formation of duricrusts in a dynamically evolving landscape. Several 1D geochemical models have been proposed for bauxite (i.e. aluminum rich laterite (Campy and Macaire, 2003)) formation (Soler and Lasaga, 1996) and iron evolution in copper and ferrous crusts (Lichtner and Biino, 1992). They couple a simple solute transport model in a porous medium with a mineral dissolution and precipitation reaction model where surface erosion is regarded as an imposed boundary condition.*

*To the contrary, our model is two-dimensional and fully coupled to a surface processes model and is designed to quantify the effect that hardening associated with duricrust formation has on the distribution and timing of erosion and the potential feedback it has on regolith formation and further duricrust formation. We will focus here on developing a model for duricrust formation based on the hydrological hypothesis (or transported model). We are in the process of developing another model (Fenske et al. in prep) based on the in-situ hypothesis, which we plan on detailing and comparing to the model presented here in a future publication.*

- it is based on a misunderstanding of the mechanisms underlying the formation of iron duricrusts;

Response: As discussed at length in our manuscript, it is well known that there exists much debate around various hypotheses for the formation of duricrusts. We agree on this point with the reviewer. The model we propose in this manuscript is based on the hypothesis that the formation of duricrusts relates to lateral transport of an element/ion that precipitates during fluctuations of the water table height, most likely in topographic lows, and thus, is similar to the lateral ferricrete hypothesis described by Bourman (1996) and Bourman et al. (2020) and other authors. Other proposed mechanisms include the in-situ hypothesis, as named by Bourman (1996), that we will test in another manuscript that is in preparation and the other type of lateral transport hypothesis of Bourman (1996) that duricrust form from the fragments of an older, dismantled duricrust (to quote Bournam (1996) describing his lateral ferricrete mechanism: "… lateral transport of physical particles or chemical precipitates derived from lateral sources.").

We modified the introduction to clarify the description of duricrust formation and the different hypotheses. The paragraphs "Hydrological hypothesis or horizontal model" and laterisation hypothesis or vertical model" (from line 70 to 102) were modified accordingly (Modifications are listed down below at the "line by line" comments).

- literature review on the topic ignores relevant information demonstrating that iron duricrusts are indeed long-lived and slowly eroding components of cratonal landscapes;

Response: We included references to direct measurements of surface exposure and erosion rate using cosmogenic isotopes that indicate that erosion rates of ferricretes erode at a rate of a few meters per million years. We also discuss the difference between surface erosion rate and rate of removal of duricrusts by lateral erosion as demonstrated by our model.

- when the model is applied it produces results that are not substantiated by other models or by observations and measurements of physical reality.

Response: We assume that the reviewer makes a reference here to our simple computation showing that hard layers are inefficient at reducing the erosional timescale of a hill (see our point above) and to his criticism of the weathering model by Braun et al (2016).

We fully agree that model results should not be taken at face value but confronted with observational constraints. Regarding the first point, our argument that duricrusts are less efficient than stated in many studies in "protecting" landscapes is not at all dependent on the details of the model but is a simple result that can be derived from any hillslope/diffusion erosion model: introducing a thin hard layer is not an efficient way to significantly increase the

time it takes for hillslope transport to erode a hill, regardless of the process that created the hardened layer. It does not mean that duricrust are not hard, but that even if they are hard (100 or more times than the rest of the hill) and thick (20% of the height of the hill) they can only retard the erosion of the hill by a factor 2. This conclusion is not inconsistent with the fact that duricrust can be very old: a duricrust can be billions of years old, until it is brought to the surface. And even when it is brought to the surface, it can resist erosion for billions of years until it is subjected to a noticeable base level fall driving erosion. We have addressed this point in the revised version of the manuscript by greatly expanding the section that reports our finding (3.3 Erosional time scale. Old manuscript lines 459 to 489. New manuscript lines 612 to 659), i.e., by including several new figures to better describe and illustrate our points, by increasing the spatial and temporal resolution of the model to further prove its accuracy, by increasing the range of model parameters to further validate our main finding and by modifying a sentence reporting our main finding so that it better expresses it (lines 648 to 651 in new manuscript): *This result is interesting as it supports the concept that duricrusts protect topographic features, but not in proportion to their apparent strength: i.e., a duricrust that is 1000 times harder than the surrounding hill can, at most, delay the erosion of the hill by a factor 2, even if it fills as much as 20% of the hill thickness.*

Regarding the second point, we do not believe that we should "defend" the physical soundness of the model that has been previously published. We note, however, that Braun et al (2016) addresses the concerns raised by the reviewer concerning the relationship between predicted regolith thickness and surface topography. As explained in Braun et al (2016), their model reproduces the observations of Rempe and Dietrich (2014) of a thicker regolith beneath hill tops in actively uplifting/eroding landscapes. It also predicts that in these situations, the water table must be very close to the regolith-bedrock interface, which is not a prediction but an assumption of Rempe and Dietrich (2014), which regard the bedrock as fractured and thus having a finite hydraulic conductivity. It is also worth noting that Rempe and Dietrich (2014) cannot predict any regolith thickness beneath the base level because, contrary to Braun et al (2016), their model assumes that it is nil there. It is also well accepted - see for example concerns expressed in Pelletier et al (2016) global model for regolith thickness – that Rempe and Dietrich (2014)'s model only applies to "uplands", i.e., tectonically active areas. In non-actively uplifting/eroding areas, Braun et al (2016)'s model predicts indeed thinner regolith thickness beneath ridge tops. As argued in Braun et al (2016) this is true at many sites in Africa and India, where geophysical sounding evidenced that regolith thickness is thinner under hilltops or is relatively uniform beneath the topography (see data from Beauvais et al (1999) in Southern Senegal and Braun et al (2009) in India; see exact references in Braun et al (2016)). We recognize, however, that it is difficult to correlate regolith thickness and surface features in low relief slowly eroding terrains, where the position of channels may have evolved since or during the time of formation of the regolith.

An important shortcoming of the work is that it is based on a misunderstanding of the models proposed for the formation of iron duricrusts, which the authors summarize as:
"Two hypotheses have been proposed for the formation of duricrusts, i.e., the hydrological or horizontal model where the enrichment in the hardening element (iron for ferricretes) is the product of leaching and precipitation through the beating of the water table during contrasted seasonal cycles, and the laterisation or vertical model, where the formation of iron duricrusts is the final stage of laterisation."
There is indeed consensus that some duricrusts form by lateral introduction of iron into lower parts of the landscape and others form by in situ concentration of iron through time (laterization model); these are the "transported and in situ ferricretes" of Bourman (1996)

Bourman, R.P., 1996. Towards distinguishing transported and in situ ferricretes: data from southern Australia. AGSO Jour. Aust. Geol. Geophys. 16 (3), 231–241. and discussed previously by several authors (e.g., Maignien, 1959, 1966; Ollier, 1969; McFarlane, 1976).

The ferricrete-producing process addressed by the numerical model in this work does not represent either of the accepted genetic models discussed by Bourman (1996) or Bourman et al. (2020). The lateral model proposes that transported ferricretes form by the lateral introduction of iron-bearing minerals, rocks and solutions into lower segments of the landscape, including channels and river beds, where ferruginization occurs. In this model, there is an unconformity between the ferricrete and underlying lithologies or weathering profiles, as described in Bourman et al. (2020). As the landscape evolves, the ferricrete is more resilient than surrounding rocks, it resists erosion, and relief inversion occurs. The channel iron deposits of Western Australia would represent an extreme example of the transported ferricrete model.

The lateritic model proposes that iron is concentrated in situ, also by physical and chemical processes, through time; the duricrust evolves through the entire history of weathering and it is not "the final stage of laterisation.", as interpreted by the authors. Recurrent physical and chemical transport of iron throughout the entire history of laterization is documented by mineralogical and geochronological work (e.g., Monteiro et al., 2014, 2018a,b).

Water table ferruginization, the issue modelled by Fenske et al., does not represent either of the models above. It does occur in special circumstances, and it is particularly common at the margins of major rivers in the Amazon, along the Atlantic coast of Brazil (Monteiro et al., 2020), in Western Australia (Mann, 1983), and other localities in Africa, India, New Caledonia and elsewhere. Iron oxyhydroxide cementation occurs in places where reducing groundwaters rich in $Fe^{+2}$ ascend towards the surface or interact with $O_2$-rich meteoric water within permeable units, and iron hydrolyses and precipitates. Such ferruginized horizons occur in areas dominated by Dunne overland flow in the vicinity of BIF plateaus at Urucum, Brazil (Vasconcelos et al., 2018). But these ferruginized horizons are generally thin, of limited spatial distribution, and are not the direct precursors of the ferricretes that control regional landscape evolution. Thus, the water table "ferruginization" problem addressed by the numerical model of Fenske et al. is only marginally related to ferricretes and it does not represent either of the main ferricrete-formation models.

In response to the reviewer's comment, we have greatly revised the section describing the three main hypotheses concerning the formation of duricrusts (Lines 86 to 161 in the new manuscript). We have taken care to express this as a classification based on formation mechanisms, not composition. As explained in our introduction, we have removed the manuscript focus on ferricretes, which has greatly helped in organising our description of the various duricrust formation mechanisms and, hopefully, remove some of the inconsistencies of the previous version of the manuscript described by the reviewer.

In addition, the physical parameters modelled by Fenske et al. are space, time, elevation (topographic height), surface slope, height of water table, precipitation rate, fluid flow velocity, hydraulic conductivity, regolith thickness, and diffusivity. The "chemical parameters" in the model are a dimensionless quantity that represents hardening or increase in "relative resistance to surface erosion", similar to that introduced by Sacek et al. (2018), and another arbitrary parameter that represents regolith hardening time. There is no chemistry in either parameter. The authors interpret these arbitrary hardening parameters as representing the formation of a ferricrete during "chemical weathering", but as it currently stands, it could be

formation of silcrete, calcrete, or any other cement that decreases regolith erodibility. The lack of any chemical parameter relevant to the dissolution and reprecipitation of iron in the near surface environment makes it improper to use these hardening parameters as a model for the formation of ferricretes.

Response: We agree with the reviewer that our model should apply to any type of duricrust. We have made this very clear throughout the revised version of the manuscript. We have also reorganized and added much material to the section concerning the use of observational constraints on the two important model parameters, i.e., the time scale tau and the water table fluctuation range, lambda (expanded section 2.3 in revised manuscript). In doing so we acknowledge more clearly that the value of these parameters is likely to be different for various types of duricrusts.

The authors also propose to "present the first numerical model for the formation of iron duricrusts based on the hydrological hypothesis". Numerical models dealing with fluid flow and chemical reactions at geological time scales, where iron dissolution and reprecipitation takes place, have been presented by Lichtner (1988), Lichtner and Waber (1992), Lichtner and Biino (1992) and many contributions since then. Lichtner and Waber (1992) consider the competing effects and the timescales of weathering and erosion, but do not directly model erosion. Thus, numerical approaches that deal with iron dissolution and reprecipitation based on robust thermodynamic, kinetic, and fluid flow models have existed for several decades.

We have added a paragraph at the end of the introduction to give reference to some of the chemical models referred to by the reviewer (lines 170 to 174 in new manuscript). We also clarified our statement that our model is the first to be *fully coupled to a surface processes model and is designed to quantify the effect that hardening associated with duricrust formation has on the distribution and timing of erosion and the potential feedback it has on regolith formation and further duricrust formation* (lines 175 to 177 in new manuscript).

Despite the fact that the basic model proposed by Fenske et al. does not address the accepted end-member models of ferricrete genesis and it does not contain any chemical component, the authors take the model results at face-value, without considering that the model itself may not be a correct depiction of nature. When faced with the fact that the model does not produce a result consistent with other observations and measurements, the authors could revisit the model to assess its applicability. Instead, the authors prefer to conclude that all observations and measurements are wrong. Publication of the manuscript as is will not advance our understanding of the role of weathering in landscape evolution, and it may mislead the uninformed reader into believing that we finally have a modelling approach that couples chemical weathering and landscape evolution, which is not the case.

This point is already responded to at great length above in response to the reviewer's comment that "when the model is applied it produces results that are not substantiated by other models or by observations and measurements of physical reality."

- I will provide below a few specific comments and suggestions, line-by-line or section-by-section, that aim at improving the current version of the manuscript:

**Line 7 and throughout the manuscript.**

The authors use the term "water table fluctuations" in their title, a perfect term for describing the vertical movement of the water table through time. They also use water table fluctuation to label the y-axis of their graph in Figure 4. However, in Line 7 of the abstract and throughout the manuscript they replace the perfectly understood and commonly used "water table fluctuation" by the "beating of the water table". The new term is unusual, confusing, totally unnecessary in face of the already available and perfectly understood "water table fluctuation". Beating of the water table and water table beating throughout the text should be replaced by "water table fluctuation" to make the text clearer and consistent with Figure 4.

Corrected. We modified lines 7, 11, 166, 168, 173, 286, 291, 320, 341, 387, 395, 414, 423, 435, 448 and 556.

**Lines 18-21**. The statement "Finally we demonstrate that the commonly accepted view that, because they are commonly found at the top of hills, duricrusts protect elements of the landscape is most likely an over-interpretation and that caution must be taken before using duricrusts as markers of uplift and/or base level falls." ignores the fact that the protection against erosion offered by duricrusts is not an "accepted view" but the result of numerous measurements and field relationships that cannot be simply "modelled away". If the model does not support the measurements and observations of physical reality, the model should be reconsidered, not alternate realities proposed.

Response: We modified the end of the abstract to better express our finding that *"although duricrusts protect elements of the landscape, they do so in a much reduced efficiency compared to their intrinsic strength."* (lines 19 to 20 in new manuscript) We have also greatly expanded the part of the manuscript where this is demonstrated (see above).

**Lines 23-24**. "Understanding Earth's surface evolution in cratonic areas remains difficult in parts due to its slow and therefore difficultly measurable rates but also due to the important contribution from chemical weathering and the formation of the regolith."

**Lines 32-33.** "It describes an indurated elemental layer usually found capping hills or surfaces, that appears to protect them from erosion (Taylor and Eggleton, 2001)."

Duricrusts are not indurated by elements, they are indurated by minerals.

Corrected.

**Lines 36-37.**
"Duricrust formation is likely to depend on water availability, often linked to climatic conditions and, for certain types of duricrusts, on the minerals present in the regolith and/or the underlying protolith."
Suggestion:
"Duricrust formation depends on water availability, often linked to climatic conditions, and on the minerals present in the regolith and/or the underlying protolith."
Duricrusts form when elements dissolve and re-precipitate, cementing parts of the regolith. Element dissolution-reprecipitation does not occur in the absence of water or the elements forming the cements.

We incorporated the suggestion lines 36-37.

**Lines 37-40** "To cite some examples, gypsite crusts form in hyper-arid areas (Watson, 1988), silcretes and calcretes form in arid environments (Nash et al., 1994), whereas iron duricrusts form in areas where more water is available during a certain period of the year (Tardy, 1993), and bauxitisation happens under tropical conditions (Retallack, 2001)."

The conditions under which duricrusts form are largely undetermined and the statements above are simply working hypotheses. For example, under similar climatic conditions a gypsum-cemented (gypsite) or a calcite-cemented (calcrete) duricrust may form, depending on the relative amounts ofHCO3-, SO4=, and Ca++ dissolved in the groundwater.

Response: We agree that the exact climatic conditions under which duricrusts form remain unclear and we added a statement to this effect in the paragraph starting at line 58 in the new manuscript and adjusted the text accordingly in the remaining part of this paragraph.

**Lines 40-42**."Although no direct measurement of their rate of formation is yet available, one can estimate that the time needed to create a duricrust is of the order of 105 or more years (Tardy, 1993; Taylor and Eggleton, 2001)."

In a very simplistic view, determining rate of formation of a weathered zone or duricrust would require measuring when the weathering front arrived at a given position within the weathering profile, as illustrated below.

For simple weathering profiles, determining the rate of formation has been successfully carried out, as for a manganocrete in Minas Gerais, Brazil, that suggests that the weathering front advanced at a rate of 8.9 ± 1 m.Ma-1 (Carmo and Vasconcelos, 2006).

The problem encountered when trying to apply the same approach to other duricrusts, particularly ferruginous duricrusts, is that recurrent mineral precipitation-dissolution-reprecipitation is the norm, and a simple age versus depth relationship in weathering profiles does not exist in most cases. Thebest that can be done with geochronology is to date as many samples as possible from various depths within a single weathering profile, such as in the example for a site in the tropical Amazon region illustrated below and use the age vs depth relationship through the oldest mineral at each depth to derive a weathering front migration rate (8.5 m.Ma-1 in the example below):

For Australian conditions, Heim et al. (2006) show that channel aggradation probably occurred at ~36 Ma, and that for the past 15 Ma the water table has been dropping and iron cementation has also progressed downward at the same rate as groundwater drawdown. Based on the (U-Th)/He results for goethite cements, water table drawdown progressed at a rate of 1.2 to 1.5 m.Ma-1.

In addition to the examples above, it is also possible to estimate rates of formation of duricrusts by dating ferruginous horizons at different states of evolution. For example, Riffel et al. (2016) shows that incipient duricrusts form at the 1-2 Ma timescales.

The various approaches available for determining the rates of formation of duricrusts illustrated above are not ideal, but they are the ones imposed by the complexities of physical reality. They also show that the statement "Although no direct measurement of their rate of formation is yet available,… " is incorrect. Much of the literature on the topic has been ignored. In addition, the absence of any direct geochronological evidence that ferricretes younger then ~ 1Ma exist also shows that the statement "one can estimate that the time needed to create a duricrust is of the order of $10^5$ or more years (Tardy, 1993; Taylor and Eggleton, 2001)." is probably incorrect because it most likely underestimates the time needed for the formation of ferricretes in nature. On the other hand, young duricrusts (< 1-2 Ma) only have a minor role in slowing erosion. They form small steps on the landscape, and they are more easily destroyed by scarp retreat than older, thicker, better cemented duricrusts.

Response: To address the reviewer's comments, we have reorganised the section describing the observational constraints on the parameter tau (first part of section 2.3 in the revised manuscript, lines 278 to 432). We have paid much attention to separate the constraints by duricrust type and to differentiate the constraints that come from direct chronological measurements from those coming from volumetric or stratigraphic observations. We have also reorganized figure 4 to better reflect this. We have also added a short statement warning the reader to differentiate between rates and ages of weathering front propagation, secondary weathering (formation of a duricrust) and surface erosion. Only in a very idealized, steady-state situation would the three rates (or age) be equivalent.

**Lines 47-49**. "We will concentrate our study on ferricretes, also called ferruginous duricrusts, iron duricrusts or iron crusts, iron enriched levels or cangas (Tardy, 1993; Tardy and Roquin, 1998; Nahon, 1991; Paton and Williams,1972; Ollier and Galloway, 1990; Vasconcelos et al., 1992; Monteiro et al., 2014; Vasconcelos and Carmo, 2018)."

Vasconcelos et al. (1992) do not address ferricretes, ferruginous duricrusts, iron duricrusts or iron crusts, iron enriched levels or cangas (Corrected). However, classical work on the topic by Dorr (1964, 1969), Maignien (1966), McFarlane (1976), Ollier (1966), Tardy (1997), Samama (1986), or those using modern tools by Vasconcelos et al. (1994), Vasconcelos (1999a,b), Shuster et al. (2005), Shuster et al. (2012), or Monteiro et al. (2018a,b,c) were mostly ignored.

See previous comment

**Lines 49-50**. "Ferricretes are indurated layers made mostly of iron, with possible traces of other elements, e.g., titanium or manganese."

The most common elements other than Fe in ferricretes are Al and Si. Ti is common in ferricretes formed over basalts or gabbros; Cr and Al are common in ferricretes formed over ultramafic rocks.

Corrected

**Lines 57-59**. "A climate conducive to the formation of iron duricrusts encompasses the following characteristics (Tardy et al., 1991; Tardy, 1993): 1) a mean annual rainfall, P, of

around 1450 m.yr−1, 2) a mean annual temperature, T, of ~28◦C, a mean relative air humidity of around 70~%, and 4) a long dry period of at least 6 months."

The numbers above are extracted from Tardy (1997) (the English edition of Tardy 1993), fifth paragraph, page 359. Tardy was careful to give ranges for the values above, not absolute figures. Unfortunately, these estimates are derived from current climate data for sites where iron duricrusts presently occur. Evidence that lateritic weathering profiles form during prolonged periods during which climates change suggest that those numbers are simple estimates not based on any direct measurement. They are Tardy's estimations from mapping the global distribution of lateritic profiles and they should be regarded as such.

**Lines 60-63**. "The preservation of iron rich levels and duricrusts through time depends on climate too. In semi-arid to arid areas, they are preserved and protect the regolith for longer periods of time than in subtropical to tropical areas as described by Taylor and Eggleton (2001); Tardy (1993)."

Some of the oldest preserved lateritic profiles on Earth occur in the Amazon (Carajás) and Minas Gerais (Quadrilátero Ferrífero), both tropical areas. Their ages, distribution, and the regional climatic conditions are thoroughly documented in the literature. Thus, the statement above is not based on any direct measurements of the distribution, ages and longevity of ferruginous duricrusts across the planet.

Response: As we decided to change the manuscript's focus from ferricretes to cover a broader range of duricrust types, details about ferricrete formation and composition were modified and/or removed for clarity. The values cited from Tardy (1997, or the French version from 1993) are indeed hypotheses made from today's regions' climates. We also added ranges for bauxites. Also, the fact that duricrusts are more or less preserved under different conditions was made by field observations by us of duricrusts in Europe and Namibia. We incorporated literature which hypothesise similar observations due to changed environmental/pH/redox conditions.

We have reorganised the first paragraphs of the introduction and expanded parts of the introduction in a way that addresses most of the reviewer's comments.

**Lines 66-69.** "There are currently two hypotheses for the formation of iron duricrusts: a hydrologically-based process (Taylor and Eggleton, 2001; Achyuthan, 2004; Widdowson, 2009; Bonsor et al., 2014; Riffel et al., 2016; Bourman et al., 2020) and a laterization based process (Tardy, 1986; Tardy et al., 1988; Nash et al., 1994; Tardy, 1993; Theveniaut and Freyssinet, 1999; Taylor and Eggleton, 2001), which are also referred to as horizontal and vertical models."

The authors should consult the literature again to make sure they understand the two models correctly. Their summaries in lines 70-113 could be greatly improved with a better understanding of the two models.

Response: We have modified the brief statements about the three different hypotheses for duricrust formation (lines 86 to 97 in revised manuscript) following the reviewer's suggestion

*and reorganized and expanded the paragraphs detailing these hypotheses (lines 104 to 161 in revised manuscript)*

**Lines 71-73**. "In this model, iron duricrusts form under a contrasting yearly climate, made of primarily wet and dry periods. During wet periods, the water table height is high and minerals, such as Fe+2, are transported from adjacent regions and accumulate. During dry periods, the water table height drops and minerals, such as Fe+3, precipitate."

Availability of aqueous (Fe+2aq) iron species is indeed an important aspect in the formation of iron duricrusts, in conjunction with iron oxidation (Fe+2aq => Fe+3) into the trivalent species, which readily hydrolyses and precipitates as goethite or hematite. But Fe+2 and Fe+3 are chemical species in solution, they are not minerals. Goethite and hematite are the relevant minerals in the case of most duricrusts. Understanding the redox behavior of iron in the near surface environment is key in formulating a credible model for the formation of duricrusts. The redox behavior of iron in the presence of oxygenated rainwater is rather complex. In general, iron solubility requires ligands, such as organic acids, to provide the acid-reducing conditions necessary to stabilize the reduced (Fe+2aq) iron species in solution. Increase in oxidation potential and alkalinity promptly promotes iron reprecipitation. Most oxygenated water tables in the near surface environment are poor in soluble iron because Eh-pH conditions are often within the thermodynamic stability field of goethite-hematite. Simply equating weathering with water table movement and defining that something precipitates when the water table moves up or down is too simplistic an approach, without any chemical basis, to properly model iron mobility and Fe-duricrust formation.

Response: As indicated earlier, we have changed the focus of the model to represent the formation of duricrusts in general. For several types of duricrusts, including some types of ferricretes, it is well accepted that the seasonal fluctuations in water table height is a good approximation of where their formation takes place. We agree with the reviewer about his comment on the need for oxidising conditions and added a few statements to this effect in our revised version of the description of the hydrological hypothesis: *"Precipitation takes place in undersaturated environments, where redox conditions are optimal, i.e., when reducing conditions become oxidising."* (lines 113 to 114 in new manuscript).

**Lines 73-76**. "This cycle repeats itself for thousands of years, with the accumulation of iron elements leading to the formation of nodules, which, ultimately, cement into a ferruginous crust. In this case, no genetic link between the bedrock and the regolith beneath is needed nor described (Ollier and Galloway, 1990; Taylor and Eggleton, 2001)."

The process repeats itself for millions of years, but the only "element" that accumulates is iron (there are no other "iron elements", there are "iron isotopes"), in the form of either goethite, hematite, or lepidocrocite, depending on water availability, pH, rates of oxidation and precipitation, etc. A genetic link is indeed missing when the duricrust forms by iron cementation of detrital phases deposited unconformably on unrelated bedrock paving valley floors. But the material that is deposited in valleys and eventually becomes iron cemented is generally iron-rich, i.e., the iron is not entirely introduced by the groundwater; it is often locally remobilized.

Response: We agree with the reviewer that duricrusts can form by cementation of detrital phases, the third hypothesis that we cite for duricrust formation. We also state that our model

is not intended to represent this mechanism. Note that, even if the iron is remobilized, its original source is likely to be regional if the duricrust originally formed by the hydrological/transported model.

**Lines 76-79**. "Also, duricrusts form at the water table, which means at multiple metres below the surface. It is generally accepted that, to permit accumulation of materials, the region needs to be tectonically inactive and that later periods of uplift (or base-level drop) are likely to exhume the duricrust to the surface where it becomes more resistant to erosion than the surrounding weathered material."

Most authors suggest that iron duricrusts form at the surface, not subsurface. For example, the duricrusts formed from Dunne overland flow at the lowlands at Urucum (Vasconcelos et al., 2018) form at the surface. If there are references that support duricrust formation by an ascending or descending water table (e.g., Mann, 1983), they should be provided. Importantly, duricrusts that form when the water table ascends to the surface, water evaporates, and minerals precipitate are more relevant to calcretes or silcretes than ferricretes.

Response: We agree with the reviewer that the water table fluctuation hypothesis is most widely seen as applicable to calcretes and silcretes. We also agree that several authors (Ollier and Galloway, 1990; Wright et al, 1992; Widdowson, 2007), however, have argued that ferricretes also form in the region of fluctuation of the water table. We have included these and others in the revised version of the manuscript to reflect this.

**Lines 100-103**. "There exists, at this stage, no numerical model for the formation of duricrusts, apart from the conceptual model developed by Nash et al. (1994) and the highly simplified model developed by Sacek et al. (2019) to estimate the effects of duricrust formation on erosional patterns at the continental scale."

The "model" of Nash et al. (1994) for the formation of calcretes and silcretes in Africa does not provide a suitable analogue for the formation of iron duricrusts. Calcretes and silcretes depend on the amount of $Ca^{++}$ and $H_4SiO_4$ in solution, and precipitation of $CaCO_3$ (in the form of calcite) and $SiO_2$ (in the form of opal, chalcedony, or quartz) is readily achieved by supersaturation when water ascends to the surface and evaporates. Iron solubility is much more complex, it depends on pH and redox conditions, it depends on the availability of ligands, iron reprecipitation may be partially catalyzed by microorganisms, etc. Iron is most often locally sourced, mobilized, and reprecipitated. Long-range transport of iron requires special circumstances, such as those documented by Mann (1983) in Western Australia.
In addition, there are numerous numerical models for the formation of Fe-bearing leached caps (Lichtner and Biino, 1992) and of bauxites (Soler and Lasaga, 1996) that are based on fundamental thermodynamics, kinetics, and fluid flow modelling. It is true that there are no models that combine fluid flow, geochemical thermodynamics/kinetics and the physical processes – e.g., hillslope diffusion, channel transport, landslide/rock fall – controlling landscape evolution, i.e., there are no models coupling weathering to erosion. Such models are still missing.

Response: We agree that the conceptual model developed by Nash et al (1994) was developed based on observations relevant to silcretes and calcretes. We now cite it as an example of a model for duricrust formation.

The model presented by the authors is a modification of a regolith formation model by Braun et al. (2016):

**Lines 114-180. 2 Method and Results**
"115 2.1 Existing regolith formation model (Braun et al., 2016)
Here we will use the model for regolith formation developed by Braun et al. (2016) that computes the rate of downward migration of a weathering front in proportion to the velocity of the water in the overlying permeable regolith. Braun et al. (2016)'s model is made of three components: a surface process model, a hydrological model and a weathering model."

The regolith profile produced by Braun et al. (2016)'s model shows a relationship between elevation and depth of weathering that is the opposite of that produced by alternative regolith forming models, such as the model of Rempe and Dietrich (2014), where deeper regolith occurs at the highest elevations and the regolith shallows towards lower landscape positions or the model by Lebedeva and Brantley (2013), that shows a similar relationship between elevation and depth of weathering.
The topography vs depth of regolith produced by Braun et al. (2016)'s model also contradicts observations of weathering profiles throughout Australia, Brazil, Africa, and China, where regolith is deeper at the higher elevation sites and becomes shallower as elevation decreases. For example, in field mapping in deeply weathered terrains we teach our students to go into valleys to look for subcrops of unweathered lithologies. Thus, the starting regolith model illustrated in Figure 1 is a problematic depiction of physical reality and it raises some concerns when further applied to explain landscape evolution. If the thickness of the regolith is easily controlled by the model, as stated in Lines 158-159.

See our response to this reviewer's comment above where we argue about the assumption behind the Braun et al (2016) model, how it is similar to the Rempe and Dietrich (2014) model under the right circumstances and how it fits geophysical observations concerning regolith thickness in non-tectonic environments where they exist.

**Lines 158-159**. "On the other hand, Γ controls whether the regolith is thickest at the top of the hill, i.e., when $\Gamma > \Omega2/\Omega-1$ , or thickest at the base of the hill, i.e., when $\Gamma < (\Omega2 / \Omega-1)$."

why not start with a regolith model that is more faithful to physical reality and to the results produced by other models?

Response: The introduction of dimensionless numbers is a fundamental procedure to understand the behaviour of a differential equation representing a physical process. In Braun et al (2016), it is demonstrated that the dimensionless number Omega we derived from our set of equations is in fact similar (and on one case identical) to the Damköhler number used by Lebedeva et (2007), Hilley et al (2010) and Li et al (2014), for example. This demonstrates that the predictions obtained using Braun et al (2016) are not only in agreement with previous models but, in fact, generalizes them.

As previously mentioned, the modified model includes physical parameters, such as space, time, elevation (topographic height), surface slope, height of water table, precipitation rate, fluid flow velocity, hydraulic conductivity, regolith thickness, and diffusivity, but the only

"chemical parameters" are a dimensionless quantity that represents hardening or increase in "relative resistance to surface erosion" and another arbitrary parameter which represents regolith hardening time. These "chemical parameter" are not directly related to the dissolution and reprecipitation of iron in the near surface environment. Indeed, if the colour in Figure 5 were changed from red-brown to grey-white, the duricrust illustrated could be a silcrete or calcrete. Thus, to equate the duricrust produced by the model with a ferricrete is an unsubstantiated extrapolation of what is essentially a physical model.

Response: We agree with the reviewer. As stated earlier, the focus on the application of the model to ferricretes has been removed from the revised manuscript.

Duricrust formation in the new model is directly proportion to the arbitrary hardening parameter (k), it is proportional to precipitation rate, to the range and rate of water table fluctuation, and regolith hardening time. The direct dependence of hardening to precipitation (rainfall) rates in the case of ferricrete formation is counterintuitive. Greater precipitation rates result in greater supply of oxygenated water, which lowers iron solubility. As iron is mobile in its reduced form ($Fe+2aq$) in most surficial environments, rainwater by itself will not contribute to iron transport. In surficial environments, organic acids must be added to infiltrating rain to drive iron dissolution and transport. In many cases, iron becomes immobilized when rainfall is abundant, and it enters solution when rainfall decreases and water ponds in the subsurface, becoming reducing.

Response: Our precipitation dependence of the hardening process is not meant to represent small scale, rapid variations in precipitation. Rather, it is a simple reflection of the widely accepted concept that weathering and dissolution/precipitation processes that are related to the formation of duricrusts by fluctuations of the water table are more likely to take place under wet conditions. It is indeed often the case that periods of duricrust formation are associated with wet period conducive to chemical weathering (Tardy, 1993; Tardy et Roquin, 1998; Beauvais et al, 2008; Taylor et Eggleton, 2017; Heller et al, 2022)

To constraint model parameters – "2.3 Constraining new model parameters" –, the authors search through the literature for physical studies that may help determining the length of time necessary for the formation of an iron duricrust. In this section, studies based on field observations and educated opinions are evaluated equally and interchangeably with studies based on actual measurement of time of duricrust formation. It would be useful to differentiate the literature into work that uses field observation to infer rates of processes to those that measure or at least attempt to measure rates of processes. And, as outlined above, many of those later studies were largely ignored by the authors. The authors also evaluate the measured ranges of water table fluctuations across the planet, and show that constraining this parameter is easier.

Response: Following the reviewer's suggestion, we incorporated more references to constrain our model parameter for the timescale τ. We also revised how we presented the results in figure 4 to avoid confusion between the different types of estimates. Also, the data has been split in more detailed categories for different types of duricrusts. We agree also with the reviewer that constraining these parameters through existing observations remains difficult. The modifications we made are from line 275 to line 432 in the new manuscript.

The model run itself is interesting, even if it is only remotely associated with the formation of ferricretes. Even if the geometry of the regolith is not that commonly measured in the field and the subsurface position of induration is not what is commonly observed in actual ferricretes, the timing of formation of a duricrust and the role of duricrust in slowing erosion is a step towards better understanding these systems. However, as mentioned above, that is only relevant to "duricrust" formation, it has no specificity as which type of duricrusts. Thus, affirming that the study poses some constraint on the formation of ferricretes is an unwarranted extrapolation of the results. And it is only relevant to the formation of duricrusts by groundwaters, which is not the dominant process controlling the formation of ferricretes.

Response: As explained above, we followed the reviewer's suggestion to not focus on ferricretes.

Also a concern is the fact that the model employed by the authors produces weathering profiles that are different from weathering profiles observed in nature and weathering profiles produced by other models. Thus, when the model also shows an ephemeral role of duricrusts in protecting landscapes against erosion, contrary to observations and measurements, its relevance should be taken with some caution. When facing such results, the authors could ask the question: "What is possibly wrong with the model?". However, the authors prefer to take the model results at face-value and question what nature tells us. They dismiss previous observations "Many authors have previously described duricrusts in lateritic or regolith profiles (e.g. Tardy (1993); Taylor and Eggleton (2001)) as a protecting layer that should slow down erosion of the underlying topography/hill. We do not observe this in our model, and now proceed to explain it." but also ignore a now large number of measurements, based on geochronology and cosmogenic isotope studies, that show that the duricrusts that sit on friable and easily erodible saprolites are indeed ancient (Vasconcelos et al. 1994; Vasconcelos, 1999a,b; Hénocque et al., 1998; Shuster et al., 2005; Beauvais et al., 2008; Monteiro et al., 2014; Vasconcelos and Carmo, 2018) and eroded very slowly (Fujioka et al.., 2010; Shuster et al., 2012; Monteiro et al., 2018a,b). Either all these measurements are wrong, and the duricrusts are not as old as determined by both 40Ar/39Ar and (U- Th)/He geochronology, and the very low rates of long-term erosion measured by both 3He and 53Mn (and now 21Ne) are also incorrect, or the duricrust formation model proposed by Fenske et al. is insufficiently robust to properly model the types of landscapes that it attempts to investigate.

Response: Whether the model predicts (or not) that topographic features are protected by duricrusts is not dependent on the model used for the weathering process and the formation of the regolith. Our argument would also work for any mechanism that would have created the duricrust, as it depends only on the thickness, position and relative strength of the duricrust. We have thoroughly revised the section 3.3 that reports this finding to better explain why duricrust do slow down erosion of topographic features but not in proportion to their intrinsic strength. We have softened our wording of the sentence quoted by the reviewer to now state that "*The very high longevity of duricrust is not observed in our model, and we now proceed to explain why.*" (line 620/621 in new manuscript). We have also explained why despite having relatively low surface erosion rate, duricrust may not survive for very long periods of time their exposure to the surface (at least not in proportion to their strength). It is because they erode by lateral degradation rather than vertical erosion.

Some authors have also demonstrated the mechanisms ("self-healing" of Monteiro et al., 2014) that allow iron duricrusts to continuously regenerate themselves, resist erosion, and protect the material below. Some of these processes have been experimentally reproduced in the laboratory, and given the right experimental conditions, greatly accelerated (Levett et al., 2019; 2020a,b,c). Thus, the longevity, resilience, and mechanisms of formation of ferruginous duricrusts are much better understood than what is portrayed in the current manuscript. Therefore, if the model cannot reproduce what is measured in nature, the model results should be more conservatively interpreted to make the manuscript into the useful and valuable contribution that it could be.

Response: We have added a sentence in the introduction (lines 82-85) referring to the recent work by the reviewer and his co-authors on the possibility that microbial activity allows for duricrusts to regenerate themselves. That this process leads to the reconstruction of duricrust by cementing of previous duricrusts fragment is part of what we termed the third process for duricrust formation and not the focus of this manuscript, as we explained in the introduction.

In summary, the authors should treat their duricrust formation process as an emerging approach, accept that their duricrust is a generic indurated material and not a ferricrete, compare their model results to natural settings to see where the model could be improved, and adjust model parameters to see if they can reproduce the measurements that show that duricrusts are actually ancient and erode very slowly.
The authors should also consider introducing actual weathering geochemistry into their model by coupling robust thermodynamic and kinetic approaches available in established geochemical models with their landscape evolution approaches. Only then it will be possible to model the formation and preservation of ferricretes, silcretes, calcretes, bauxites, and other types of duricrusts and the supergene systems that they cover.
In summary, the model in this manuscript can be a valuable contribution if presented for what it is and more reservedly interpreted. As currently presented, it will mislead readers into believing that we currently have a model that couples chemical weathering and landscape evolution, when this is not the case. To be the valuable contribution that it could be, the manuscript should be substantially revised and re-written. I have not checked the references, supplementary materials, etc. I will be willing to do so if a revised version of the manuscript is produced.

General response:
We agree with the reviewers that the model we develop for duricrust formation by fluctuations of the water table adequately represent other types of duricrusts than just ferricretes. We modified the manuscript to ensure that our model is presented such that the ferricretes appear as one of the types of duricrusts that can form in this way. We modified the introduction accordingly, by adding more references to this effect and details about the different types of duricrusts that could form through the transport-based model. We modified lines 30 to 35, 43 to 69 accordingly, and throughout the manuscript, when "iron duricrust" or "ferricrete" was mentioned and could have been a general duricrust, was replaced by "duricrust", "crust" or "hard layer".

---

## Author Response (AR2)

We thank the reviewer John Webb, the handling editor Andreas Lang and the handling associate editor Orencio Duran Vincent for their feedback.

We address both points in the revised manuscript. First, concerning the flat water table, we thought we already addressed this point, however, we added further explanation according to Braun et al. (2016) in the manuscript:

Line 227: "The steady-state geometry of the water table depends also on the values of $\Omega$ and $\Gamma$. In steep topographies typical of tectonically active regions $\Omega\sim1$ and $\Gamma>1$, the water table is close to the bedrock (base of the regolith layer) as observed and assumed in Rempe et al. (2014). In all settings, $\Omega$ is a direct measure of the ratio between the surface slope and the steady state slope of the water table (Braun et al. 2016)." We added "In our reference model, the value of $\Omega$ (~ 6) implies that the water table slope is six times smaller than the surface slope. As explained in details in Braun et al. (2016), a higher water table slope could be obtained by decreasing the value of $\Omega$, by decreasing the value of the hydraulic conductivity, for example."

The next figure shows that with decreasing $\Omega$ the water table becomes steeper. In our reference models, we use a high value of $\Omega$, which explains the flatter geometry of the presented water tables. We do not think it is necessary to add this figure in the manuscript as this point is already discussed in Braun et al. (2014).

[Figure]

Secondly, concerning the point made about the generalisation to duricrusts or ferricretes: in the revised version seen by the reviewer, we decided to describe duricrusts in more general terms, taking into account the different processes described for the formation of calcretes or silcretes, i.e., by evaporation processes. We now added a few sentences in the introduction for a better depiction of the involved processes. We are aware that some duricrusts, e.e., calcretes and silcretes, do not form through oxidation but rather through other processes such as evaporation, evapotranspiration, or $CO_2$ degassing. The subsection "Hydrological hypothesis of transport mode" has been modified to include this. We added a few new references to address it too.

- Line 56: "Calcrete formation is described under semi-arid to arid climates, with annual precipitation, P , around 200 to 600 mm/yr, and mean annual temperatures, T , at ~18∘C (Eren et al., 2008). Khalaf (2007); Moussavi-Harami et al. (2009) determine that "the suitable climate for calcrete formation include temperatures that faciliate high evaporation rate"."
- Line 110: added a citation, Taylor et Eggleton 2001.
- Line 112 to 114: "During dry periods, the water table height drops and the transported minerals precipitate in response to changing redox (e.g. for ferricretes and alcretes), pH (e.g. for calcretes) and environmental conditions such as salinity (e.g. for silcretes), water availability and evaporation processes (e.g. for calcretes and silcretes) (Taylor and Eggleton, 2001, e.g.). Precipitation takes place in undersaturated environments. For ferricretes, it is where redox conditions are optimal, i.e., where reducing conditions become oxidising."
- Line 119: "[…]. For calcretes, the main drivers are evaporation and evapotranspiration processes linked to water table fluctuations, and CO2 degassing (Alonso-Zarza and Wright, 2010). Such processes only take place at the water table height or in the vadose zone (Moussavi-Harami et al., 2009; Alonso-Zarza and Wright, 2010). Silcrete formation processes remain poorly understood (Thiry and Milnes, 2017; Taylor and Eggleton, 2017). However, evaporation of silica-rich fluids within the regolith is suggested as one of the primary drivers (Taylor and Eggleton, 2001; Thiry and Milnes, 2017). Groundwaters are typically saturated with quartz or amorphous silica (Taylor and Eggleton, 2017).

  The seasonal cycle of dissolution and precipitation repeats itself for thousands of years, with the accumulation of minerals leading to the formation of nodules, which, ultimately, cement into a duricrust."

As noted by the reviewer, calcretes may form through vertical processes in the regolith column. We agree with this statement, and to address it, we would like to note that we are currently preparing a second article which is almost ready for submission, involving the vertical processes for duricrust formation and would better apply to some types of duricrusts, e.g. ferricretes or calcretes.

We thank the reviewer again, and hope we have answered all questions concerning the revisions in the new manuscript.